



# A leading-order viscoelastic model for crevasse propagation and calving in ice shelves

Maryam Zarrinderakht[1], Christian Schoof[1], and Thomas Zwinger[2]

[1]Department of Earth, Ocean and Atmospheric Sciences, University of British Columbia, BC, Canada
[2]CSC-IT Center for Science, Espoo, Finland

**Correspondence:** M. Zarrinderakht (mzaryam@eoas.ubc.ca)

**Abstract.** We use a leading-order viscoelastic model for crevasse evolution, in which a purely viscous model for the deformation of the domain couples with linear elastic fracture mechanics models through a viscous pre-stress. The fracture mechanics model conversely couples with the viscous model by inserting cracks into the domain, which viscous flow subsequently pulls apart. By contrast with prior work, we solve the fracture mechanics problem on the actual domain geometry using a boundary element method, coupled with a finite element solution of the Stokes equations describing the viscous flow. We study a periodic array of surface and basal crevasses on an ice shelf being stretched at a prescribed rate. We find that calving can either occur instantly for large enough stretching rates or sufficiently high surface water levels or through feedbacks between partial fracture propagation and subsequent viscous deformation and adjustment of the viscous pre-stress acting on crack faces. Our results show that purely stress-based calving laws cannot robustly describe the process of calving, since they cannot account for the gradual evolution of local crevasse and surface geometry, which can be understood at the large scale as being more akin to the evolution of a damage variable. Future work will need to coarse-grain the type of process model we describe here in order to make it applicable to ice sheet simulations.

## 1 Introduction

Alongside basal melting, calving is the main mechanism that causes mass loss from ice shelves (Depoorter et al., 2013). In turn, the resulting changes in ice shelf geometry regulate the buttressing effect of ice shelves that limits mass discharge from the grounded portion of marine ice sheets (e.g. Schoof et al., 2017). Despite the key role that calving, therefore, plays in controlling grounded ice volume and changes in global sea level, a complete physics-based model is not yet available.

Here we build on attempts to understand the calving of ice shelves as an example of linear elastic fracture mechanics (Weertman, 1973, 1980; van der Veen, 1998a,b; Jiménez and Duddu, 2018; Lai et al., 2020; Zarrinderakht et al., 2022, submitted ). A common occurrence in such models is that fractures propagate part-way through an ice shelf but then reach an equilibrium configuration in which the shelf remains intact overall. The only way that further crevasse growth can then occur and cause calving is if the forcing on the system is changed, typically in the form of increased extensional stress or surface water pressure.

While such changes in force can occur rapidly, for instance, if a large-scale downstream calving event significantly reduces buttressing and therefore increases stresses near the modelled crack, they are likely to be more gradual: as an example, a crack



is advected along an ice shelf will typically be subject to progressively reduced buttressing. The challenge for an elastic model
is that it does not describe ice response over such long-time scales. Even though ice behaves elastically at short time scales, its
long-term behaviour is purely viscous. The simplest rheology that combines a short-term elastic response with the long-term
viscous flow is a Maxwell-type viscoelastic constitute relation (Christensen, 1971), in which the change from elastic to viscous
behaviour occurs over a time scale set by the ratio of ice viscosity to Young's modulus. That ratio is known as the Maxwell
time.

Taking a conservative estimate of extensional stress of $2 \times 10^5$ in an ice shelf with Glen's law parameters appropriate for ice
at -10° C (Cuffey and Paterson, 2010), we find a viscosity around $10^{13}$ Pa s. With Young's modulus of $10^9$ Pa, the Maxwell
time is around 7 hours: long compared with the time scale for crack propagation (Olinger et al., 2022), but short compared
with the time scale over which an ice shelf flows and the forcing on any cracks in the ice therefore changes.

Krug et al. (2014) and Yu et al. (2017) avoid some of the limitations of purely elastic fracture mechanics models by com-
bining them with viscous flow descriptions. Both of their models extract viscous stress from an established ice flow model that
also computes the evolution of the ice geometry and applies that stress to force fracture opening in a linear elastic fracture
propagation model. In Krug et al. (2014), the ice flow is described by Stokes' equations, while Yu et al. (2017) additionally
uses a Herterich-Blatter-Pattyn model (Herterich, 1987; Blatter, 1995; Pattyn, 2003) or a depth-integrated shallow shelf model
(Macayeal et al., 1987; MacAyeal and Barcilon, 1988; Morland, 1987). The models of Krug et al. (2014) and Yu et al. (2017)
however also have in common that they do not solve the linear elastic fracture mechanics problem on the actual domain geom-
etry, but use a parallel-sided slab geometry as a proxy for the actual domain shape, permitting the use of interpolated Green's
functions as in van der Veen (1998a,b) and Lai et al. (2020).

Here, we explicitly formulate a model of viscoelastic ice in which fractures propagate at time scales much shorter than the
Maxwell time, and significant viscous deformation (in the sense of $O(1)$ strains) occurs over time scales much longer than a
Maxwell time. The coupling between viscous flow and linear elastic fracture mechanics then takes two forms: first, viscous
flow sets the geometry of the ice that is fractured, which affects the distribution of elastic stresses and hence controls how
far a crack will propagate. Second, the elastic stresses that are generated during fracture propagation are superimposed on the
pre-existing viscous stress at the time of fracture initiation. The sign of that pre-stress near the point of crack initiation (or
rather, the sum of that pre-stress and the water pressure at the boundary of the ice) is key to determining whether a fracture
will propagate. As in Krug et al. (2014) and version of the model in Yu et al. (2017), we extract the viscous pre-stress from
a solution of Stokes' equations: the Herterich-Blatter-Pattyn and shallow shelf models used by Yu et al. (2017) are based on
the assumption of small aspect ratios (Schoof and Hindmarsh, 2010), which does not apply in the vicinity of fractures. A key
novelty in our work is the use of a boundary element method (Zarrinderakht et al., 2022) to solve the linear elastic fracture
mechanics component of the model on the actual domain geometry that results from viscous deformation.

Our focus is on the evolution of a periodic array of vertical crevasses in an ice shelf, spaced at distances comparable to the
thickness of the ice, though the generalization of the model to other geometries should be obvious. Our goal is to characterize
how interactions between viscous flow and fracture propagation lead to calving and to map out parameter combinations for
which different styles of calving occur. The paper is organized as follows: we state the basic model in section 2, distinguishing



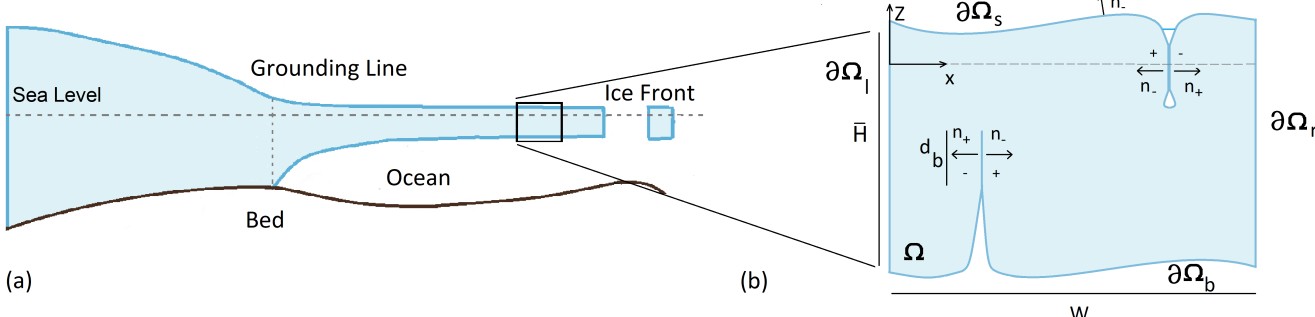

**Figure 1.** Cross section geometry of a marine ice sheet, and the geometry of the periodic problem: part of a floating ice shelf with offset basal and surface crevasses. The symbols shown are defined in the running text; $\Omega$ denotes the domain, with boundary $\partial\Omega$, $\mathbf{n}$ is the outward-pointing surface normal, while $d$ denotes crack length and $W$ is domain width.

between the viscous (section 2.2) and elastic (sections 2.3 and 2.4) behaviour that occurs on distinct time scales. The numerical procedure used is outlined in section 2.5. Section 2.6 details the simple forcing of the model that we use in the calculations reported in the paper, involving a prescribed stretching rate and surface water table height. We give illustrative examples of how crevasses evolve through the interplay of viscous deformation and crack propagation in sections 3.1–3.2, focusing on the role of a quantity we term the "effective pre-stress" in controlling when an existing crevasse will extend further. We investigate systematically how different levels of forcing lead to different styles of calving in section 3.4 and discuss the implications of our results as well as limitations and future improvements of the model in section 4.

## 2 Model

### 2.1 Evolving geometry, conservation laws and boundary conditions

Assume a two-dimensional Cartesian coordinate system $(x_1, x_2) = (x, z)$ in which the $x$-axis is horizontal and $z = 0$ defines sea level. Consider ice occupying a time-dependent domain $\Omega(t)$ that is periodic in $x$, with a time-dependent spatial period $W(t)$. The coordinate system is intended to be a local system that follows a piece of ice, of a comparable horizontal extent to its thickness, as it travels along the ice shelf, so the origin in $x$ moves at the average speed of the ice at that location. Denote the velocity field in the ice relative to that average by $\mathbf{v} = (v_1, v_2)$. For simplicity, assume that there are no net surface mass balance terms. At times when there is no propagation of cracks, the upper and lower surfaces $\partial\Omega_\mathrm{s}$ and $\partial\Omega_\mathrm{b}$ are therefore material surfaces. We can then parameterize the position of a point on these material surfaces in the form $(x_1, x_2) = \mathbf{X}(\mu, t)$, where $\mu$ parameterizes position along the surface and the function $\mathbf{X}$ satisfies a kinematic boundary condition

$$\dot{\mathbf{X}} = \mathbf{v}(\mathbf{X}, t), \tag{1}$$

where the overdot denotes differentiation with respect to time $t$, here at constant $\mu$: $\mu$ is simply a Lagrangian tracer that labels points on the surface.



Equation (1) differs from more commonly seen kinematic boundary conditions that specify the rate of change of surface elevation $x_2$, but is necessary here to account for surface elevation potentially not being single-valued once there is significant deformation, or for vertical portions of the surface.

Both of these situations can plausibly occur as the result of fracture propagation, which we will discuss shortly. For now, we emphasize that the growth of a fracture does not correspond to the motion of a material surface, and therefore does not satisfy

a kinematic boundary condition of the form (1): it corresponds to the introduction of a new portion of the surface in the curve we obtain by treating $\mathbf{X}(\mu, t)$ as a function of $\mu$ at fixed $t$: $\mathbf{X}$ then becomes discontinuous in $\mu$ at the location of the new crack.

Denote the stress field in the ice by $\sigma_{ij}$. We neglect inertial effects and impose a balance of forces in the form

$$\frac{\partial \sigma_{ij}}{\partial x_j} + \rho_{\mathrm{i}} g_i = 0 \tag{2}$$

where $\rho_{\mathrm{i}}$ is the density of ice, $\mathbf{g} = (0, -g)$ is the acceleration due to gravity, and we have used standard subscript notation

including the summation convention over $j \in \{1, 2\}$. The density $\rho_{\mathrm{i}}$ satisfies a mass conservation equation of the formation

$$\frac{\partial \rho_{\mathrm{i}}}{\partial t} + \nabla \cdot (\rho_{\mathrm{i}} \mathbf{v}) = 0, \tag{3}$$

$\nabla$ being the usual gradient operator $(\partial/\partial x, \partial/\partial z)$. Provided there is no contact of the ice surface with itself, we assume that there is an imposed normal stress on the surfaces $\partial \Omega_{\mathrm{s}}$ and $\partial \Omega_{\mathrm{b}}$, given by the fluid in contact with the surface, which can be ocean water, surface water or air. In addition, we assume vanishing shear stress, so that

$$\sigma_{ij} n_j = -p_{\mathrm{f}} n_i, \tag{4}$$

where $\mathbf{n} = (n_1, n_2)$ is the outward-pointing unit normal and $p_{\mathrm{f}}$ is the fluid pressure.

This simple condition fails where the ice re-establishes contact with itself, that is, where $\mathbf{X}(\mu_1, t) = \mathbf{X}(\mu_2, t)$ for distinct values $\mu_1$ and $\mu_2$. In many fluid dynamical situations, such a situation would be modelled by assuming that the fluid on both sides of the contact merges (Crowdy, 1999), as is also assumed in some glacier flow models (Jouvet et al., 2008). Our focus on

fracture propagation in ice that behaves partly as a solid leads us to consider a different possibility: we assume that a fracture-like surface remains. Using superscripts $+$ and $-$ to denote limits taken from either side of the contact, an appropriate set of conditions at such a surface (by no means the only conceivable ones) are then (see also Stubblefield et al., 2021; de Diego et al., 2022; Zarrinderakht et al., 2022)

either    $-[v_i n_i]^+_- > 0$   and   $-\sigma_{ij} n_i n_j = p_{\mathrm{f}},$ $\hspace{2cm}$ (5a)

or    $-[v_i n_i]^+_- = 0,$   $[\sigma_{ij} n_j]^+_- = 0,$   and   $-\sigma_{ij} n_i n_j \geq p_{\mathrm{f}},$ $\hspace{2cm}$ (5b)

where $[f]^+_- = f^+ + f^-$ is the sum of the limiting values of the bracketed quantity, $n_i^{\pm}$ being the outward-pointing normal to the side labeled '+' or '−'; lack of a superscript indicates that the equation holds regardless of which side of the contact the limit is taken from. In addition,

$$(\delta_{ij} - n_i n_j) \sigma_{jk} n_k = 0, \tag{5c}$$



where $\delta_{ij}$ is the usual Kronecker delta. The conditions (5b) state that normal stress in the contact areas is still given by equation (4) when the surfaces are about to move apart (the sum of the outward-pointing normal components of velocity $v_i^+ n_i^+ + v_i^- n_i^-$ measures how fast the two sides of the contact area move *towards* each other), with normal compress stress potentially exceeding the fluid if the surfaces are not moving apart. The second condition (5c) imposes vanishing shear stress as done previously in Zarrinderakht et al. (2022), neglecting the possibility of ice-on-ice friction.

We denote the left and right lateral boundaries by $\partial\Omega_l$ and $\partial\Omega_r$, respectively. In assuming a periodic domain of width $W$, we require every point $(x, z)$ on $\partial\Omega_l$ to have a counterpart $(x + W, z)$ on $\partial\Omega_r$, and vice versa. Denote by $[\cdot]_l^r$ the difference in the bracketed quantity between a point on $\partial\Omega_r$ and its counterpart on $\partial\Omega_l$. We impose periodic boundary conditions on stress as

$$[\sigma_{ij}]_l^r n_j = 0, \tag{6}$$

$n_j$ being the outward-pointing at one of the lateral boundaries. The assumption of a domain that is periodic in $x$ but able to

stretch with a time-dependent periodicity $W(t)$ means that we do not impose periodic conditions on the velocity field: instead, the horizontal velocity component on the right-hand end of the domain can differ from that on the left by uniform amount, which we can quantify in terms of a stretching rate. In other words,

$$[v_1]_l^r = V_X(t)W(t), \qquad [v_2]_l^r = 0,, \tag{7}$$

where the stretching rate $V_X$ is independent of position.

Even though the lateral domain boundaries are not boundaries between ice and air or water but separate ice in the domain from the rest of the ice shelf, we also treat them as material surfaces, satisfying equation (1). From the quasi-periodic velocity boundary conditions, it then follows that, if points on $\partial\Omega_r$ and $\partial\Omega_l$ can be matched initially through a horizontal translation $x \mapsto x + W(0)$, they remain matched through a translation $x \mapsto x + W(t)$, with the rate at which the domain stretches given by $V_X$ as

$$\dot{W} = V_X W. \tag{8}$$

    A full statement of boundary conditions requires the fluid pressure $p_f$ to be specified. On the bottom surface $\partial\Omega_b$, we assume a hydrostatic increase in water pressure below sea level,

$$p_f = p_{f,b}(z) = \rho_w g \max(-z, 0), \tag{9}$$

where $\rho_w$ is the density of water. On the upper surface $\partial\Omega_s$, we assume that there is a prescribed water level that we choose to

express as a depth to the water table $h_w(t)$ relative to an approximate mean upper surface elevation $\bar{s}$

$$p_f = p_{f,s}(z, t) = \max(\rho_w g(\bar{s}(t) - h_w - z), 0). \tag{10}$$

The precise definition of $\bar{s}$ used in our computations is given in section 2.6; the formulation for $p_f$ in terms of two parameters $h_w$ and $\bar{s}$ (one of which ought to be redundant) is motivated by wanting to align our choice of parameters with that in the previous work by van der Veen (1998a) and Zarrinderakht et al. (2022, submitted ), who prescribe water level as depth below the upper

ice surface (which is trivial in their case since that upper surface is flat, while ours is not once there has been significant viscous deformation of the ice).





## 2.2 Rheology: viscous time scales

We assume that ice is an elastically compressible but viscously incompressible, upper-convected Maxwell fluid with a power-law viscosity (Zarrinderakht et al., 2022, appendix A). That does not, however, mean that we solve the problem consisting of
equations (1)-(10) with full, viscoelastic rheology. Instead, we appeal to a separation of time scales to split the problem into two parts. In the first, the ice behaves entirely viscously, and its flow deforms the domain $\Omega$ significantly as described by equation (1). In the second part, the ice behaves elastically and alters the domain purely through the rapid growth of cracks that can subsequently be opened by viscous flow. From the perspective of the slow, viscous problem, the growth of cracks described by the second, elastic fracture mechanics problem (which is technically an "internal layer" in time, Holmes (1995)) then happens
instantly.

In this section, we consider the purely viscous flow problem. For any realistic combination of ice shelf thickness, stress, elastic moduli, and viscosity parameters, significant deformation of the domain geometry over time scales much longer than a single Maxwell time. The ice then behaves purely as an incompressible viscous fluid. Stress therefore satisfies

$$\sigma_{ij} - \frac{1}{2}\sigma_{kk}\delta_{ij} = BD^{1/n-1}D_{ij}, \tag{11}$$

$$D_{kk} = \nabla \cdot \mathbf{v} = 0, \tag{12}$$

where the summation is over $k \in \{1,2\}$ a d

$$D_{ij} = \frac{1}{2}\left(\frac{\partial v_i}{\partial x_j} + \frac{\partial v_j}{\partial x_i}\right), \qquad D = \sqrt{D_{ij}D_{ij}/2}, \tag{13}$$

are strain rate and its second invariant, respectively. $B$ and $n$ are the usual parameters in Glen's law (Cuffey and Paterson, 2010). The incompressibility condition (12) ensures that we can treat $\rho_{\mathrm{i}}$ in equations (3) and therefore in (2) as a constant.
The problem (1)–(13) is a relatively straightforward Stokes flow free boundary problem (with the addition of slightly unusual contact conditions, which actually turn out to be problematic for our current numerical solver as described in section 2.5 below), The model describes the viscous deformation of the ice at all times when there is no crack propagation. The only missing component is the introduction of new crack surfaces.

Note however that the stress boundary conditions on $\partial\Omega_{\mathrm{b}}$ and $\partial\Omega_{\mathrm{s}}$ combined with the periodic boundary conditions imply
that the existence of solutions is contingent on a global force balance condition (the depth at which ice sits in the ocean is set by Archimedes' principle, after accounting for the extra load imposed by surface water) and that an arbitrary constant vertical component can apparently be added to a valid velocity solution $\mathbf{v}$ and still solve the Stokes flow problem (2)–(13). This observation is closely associated with the force balance problem described previously in Berg and Bassis (2020): the apparent indeterminacy in the velocity field is easily resolved by insisting that global force balance must hold at all times, so
the net vertical motion must maintain the balance in buoyant forces. We give details in appendix C.

## 2.3 Elastic time scales

Suppose that crack growth is initiated at some distinct time $t = t_{\mathrm{c}}$, and denote the top and bottom parts of the domain boundary just prior to $t_{\mathrm{c}}$ by $\partial\Omega_{\mathrm{b}}^-$ and $\partial\Omega_{\mathrm{s}}^-$. We assume that cracks propagate much faster than a single Maxwell time. The previous



assumption in section 2.2, that significant changes in the domain $\Omega$ due to flow only occur at time scales much larger than a
single Maxwell time, implies that strains remain small over time scales comparable to or shorter than a Maxwell time: to be
specific, the strain that accumulates at the Maxwell time or faster scales as the ratio of Maxwell time to the viscous deformation
time scale. As a result, rapid crack propagation corresponds to small strains. At leading order, the two sides of the crack can
therefore be treated as parts of $\partial\Omega$ that coincide with each other, bordering $\Omega$ on both sides in the usual sense of linear elastic
fracture mechanics. Importantly, the domain remains otherwise unchanged during crack propagation.

In addition, during fracture propagation over time scales much less than a Maxwell time (which leads to an abrupt change
in boundary conditions along the new crack face, and hence to an equally rapid change in stress field), the total Cauchy stress
is the sum of a pre-stress $\sigma_{ij}^{\mathrm{v}}$ (equal to the stress field just prior to $t = t_{\mathrm{c}}$), and additional elastic stress $\sigma_{ij}^{\mathrm{e}}$ (Zarrinderakht et al.,
2022, appendix A):

$$\sigma_{ij} = \sigma_{ij}^{\mathrm{v}} + \sigma_{ij}^{\mathrm{e}}, \tag{14}$$

where we will assume that the pre-stress is given by the purely viscous stress field immediately prior to $t_c$,

$$\sigma_{ij}^{\mathrm{v}} = \lim_{t \to t_c^-} \sigma_{ij}, \tag{15}$$

$\sigma_{ij}$ being determined by the solution of equations (2)–(13), with the domain geometry being taken just prior to $t_c$. This turns
out to be an assumption that requires some care, as we discuss in section 4.

The additional elastic stress can be related through a standard linear elastic model to the strain $\varepsilon_{ij}$ accrued since the beginning
of crack propagation:

$$\varepsilon_{ij} = \frac{1+\nu}{E}\left(\sigma_{ij}^{\mathrm{e}} - \nu\sigma_{kk}^{\mathrm{e}}\delta_{ij}\right), \tag{16}$$

where $E$ is Young's modulus, $\nu$ is Poisson's ratio and $\delta_{ij}$. We have assumed plane strain conditions, with vanishing out-of-
plane displacements, and Latin alphabet subscripts continue to range over $\{1, 2\}$. Strain $\varepsilon_{ij}$ is given in terms of displacements
$u_i$ that have occurred since time $t_i$

$$\varepsilon_{ij} = \frac{1}{2}\left(\frac{\partial u_i}{\partial x_j} + \frac{\partial u_j}{\partial x_i}\right), \tag{17}$$

where $u_i(x, z, t) = \int_{t_{\mathrm{c}}}^{t} v_i(x, z, t')\mathrm{d}t'$ (or equivalently, $\partial u_i/\partial t = v_i$ with $u_i = 0$ at $t = t_{\mathrm{c}}$) and $t - t_{\mathrm{c}}$ is much less than a Maxwell
time.

Note that, unlike the viscous model, the elastic model (16) is compressible (unless we choose the extreme case of a Poisson's
ratio $\nu = 1/2$). With elastic strains remaining small, this implies that there are very small but non-zero fractional changes in
density comparable in size to the elastic strain. These density changes can be computed from equations (3), but have no
significance for the computation of elastic stresses as the gravitational body force $\rho_{\mathrm{i}} g_i$ is unchanged at leading order. In the
domain, we assume that inertial effects can continue to be neglected, so the force balance relation (2) still holds. Since $\sigma_{ij}^{\mathrm{v}}$
satisfies equation (2), the elastic stress $\sigma_{ij}^{\mathrm{e}}$ more simply satisfies

$$\frac{\partial \sigma_{ij}^{\mathrm{e}}}{\partial x_j} = 0. \tag{18}$$





In addition, $\sigma_{ij}^{\mathrm{v}}$ satisfies the imposed boundary condition (4) on any parts of the pre-existing surface $\partial\Omega_{\mathrm{b}}^{-} \cup \partial\Omega_{\mathrm{b}}^{+}$ where there is no contact immediately prior to $t_{\mathrm{c}}$. Since $\sigma_{ij}n_j = (\sigma_{ij}^{\mathrm{e}} + \sigma_{ij}^{\mathrm{v}})n_j = -p_{\mathrm{f}}n_i$ also satisfies equation (4), it follows that

$$\sigma_{ij}^{\mathrm{e}}n_j = 0, \tag{19}$$

on these parts of $\partial\Omega_{\mathrm{b}}^{-}$ and $\partial\Omega_{\mathrm{s}}^{-}$.

On the newly introduced crack surfaces or any part of $\partial\Omega_{\mathrm{b}}^{-} \cup \partial\Omega_{\mathrm{s}}^{-}$ that was a contact surface immediately prior to $t = t_{\mathrm{c}}$ (referred to below as a pre-existing contact area), it can be shown that we obtain boundary conditions of the same form as (5), but with velocity $v_i$ replaced by displacement $u_i$ throughout, and fluid pressure $p_{\mathrm{f}}$ replaced by a quantity we denote by

$$\sigma_{nn}^{\mathrm{eff}} := \sigma_{ij}^{\mathrm{v}}n_i n_j + p_{\mathrm{f}}. \tag{20}$$

Demonstrating that this set of boundary conditions holds using the original conditions (5) as a starting point involves some subtleties that we defer to appendix B. Similarly, boundary conditions at the lateral boundaries mirror (6) and (7), now in the form

$$[u_i]_1^{\mathrm{r}} = [\sigma_{ij}^{\mathrm{e}}]_1^{\mathrm{r}} n_j = -0. \tag{21}$$

The stretching rate does not appear in equation $(21)_1$ because there is insignificant stretching (compared to elastic strains) over time scales much less than a Maxwell time.

Note that the term $\sigma_{nn}^{\mathrm{eff}} = \sigma_{ij}^{\mathrm{v}}n_i n_j + p_{\mathrm{f}}$ as defined in equation (20) appears in the elastic problem as an imposed normal stress on crack faces and pre-existing contact areas (see the conditions (B1) in appendix B). It can be interpreted as the normal component of an "effective pre-stress", defined analogously to the usual definition of effective stress as the sum of pre-stress and fluid pressure, $\sigma_{ij}^{\mathrm{eff}} = \sigma_{ij}^{\mathrm{v}} + p_{\mathrm{f}}\delta_{ij}$, in which case $\sigma_{ij}^{\mathrm{v}}n_i n_j + p_{\mathrm{f}} = \sigma_{ij}^{\mathrm{eff}}n_i n_j = \sigma_{nn}^{\mathrm{eff}}$. For future reference, we will call the effective pre-stress tensile if $\sigma_{nn}^{\mathrm{eff}} > 0$ (the sum of the compressive-negative normal component of pre-stress $\sigma_{nn}^{\mathrm{v}}$ "pulling" on the surface from within the ice and the fluid pressure $p_{\mathrm{f}}$ pushing from the outside is positive), and compressive if $\sigma_{nn}^{\mathrm{eff}} < 0$.

Given an effective pre-stress $\sigma_{ij}^{\mathrm{eff}}$ and a current geometry, including crack configuration, equations (16)–(21) combined with the conditions (B1) are sufficient to determine the elastic stress field $\sigma_{ij}^{\mathrm{e}}$. The elastostatic problem defined by these equations is essentially the same as that considered in Zarrinderakht et al. (2022, submitted), albeit in a more general geometry. The key observation for coupling the viscous flow problem of section 2.2 to the elastic problem here is that coupling occurs purely through the effective pre-stress, acting solely on contact areas and new crack surfaces.

## 2.4 Crack propagation

A description of how cracks actually propagate in response to the stress field around the crack tip is still missing. The formulation up to this point has been quite general. In what follows, we will consider only cracks that grow vertically in two predefined locations, one at $x = x_{\mathrm{s}}(t)$ for cracks extending from the upper surface $\partial\Omega_{\mathrm{s}}$ and the other at $x = x_{\mathrm{b}}(t)$ from the lower surface $\partial\Omega_{\mathrm{b}}$. We assume an initially rectangluar domain shape between $x_1 = -W(0)/2$ and $x_1 = W(0)/2$, and assume that the crack locations start at $x_{\mathrm{b}}(0) = -W(0)/2$ and $x_{\mathrm{s}}(0) = W/4$, as also considered (with a uniform shift in $x_1$) in Zarrinderakht et al.



(submitted ). The initial domain geometry is symmetrical about these initial locations and cracks are subsequently advected at the local ice flow velocity as in Figure 5a. It is then easy to show from the quasi-periodic boundary conditions (6) and (7) that the stress field remains symmetrical about the crack locations and vertical crack propagation is self-consistent, if contrived. By not considering arbitrary crack locations and geometries, we are able to take a simpler computational approach that leads

to significant qualitative insight, building on the companion paper Zarrinderakht et al. (submitted ). We anticipate that future work will deal with much more general crack geometries.

As in Zarrinderakht et al. (2022), we assume that crack propagation can be modelled in terms of the difference between the static stress intensity factor at the crack tip and fracture toughness for the ice (Freund, 1990)

$$\dot{d} = \max\left(\frac{K_{\mathrm{I,stat}} - K_{\mathrm{Ic}}}{K_{\mathrm{Ic}}|K'(0)|}, 0\right),\tag{22}$$

where $d$ is the length of the crack that has grown since $t_c$, the dot again denotes differentiation with respect to time, $K_{Ic}$ is fracture toughness of ice, and $K'$ is the derivative of the "universal function" in Freund (1990). $K_{I,stat}$ is the static stress intensity factor, given by the near-field around the crack tip of the elastostatic stress $\sigma_{ij}^{\mathrm{e}}$ computed from (16)–(21): using a local $(r, \theta)$ coordinate system centered on the crack tip, $\theta = 0$ aligned with the crack,

$$K_{\mathrm{I,stat}} = \lim_{r \to 0} \sqrt{2\pi r} \sigma_{\theta\theta}(r, \pi).\tag{23}$$

Note that, in general, the pre-stress will be non-singular at the crack tip (which is an interior point of the domain on which the pre-stress was computed), in which case $\sigma_{\theta\theta}$ can be replaced at $\sigma_{\theta\theta}^{\mathrm{e}}$.

With a finite fracture toughness $K_{\mathrm{Ic}}$, a finite initial crack length is generally required for $K_{\mathrm{I,stat}}$ to exceed $K_{\mathrm{Ic}}$, leading to the non-trivial question of how that initial crack can be generated (e.g. Krug et al., 2014). Here we sidestep that issue by noting that estimated fracture toughness values for ice $K_{\mathrm{Ic}} \approx 0.1$ MPa m$^{1/2}$ (Rist et al., 1996) are typically much smaller than

the stress intensity factors $K_{\mathrm{I,stat}} \sim (\rho_{\mathrm{w}} - \rho_{\mathrm{i}})gH^{3/2} \approx 10$ MPa m$^{3/2}$ that we can expect from the boundary values of stress imposed on the elastic problem with an ice thickness $H \sim 500$ m, $\rho_{\mathrm{i}} = 917$ kg m$^{-3}$, $\rho_{\mathrm{w}} = 1028$ kg m$^{-3}$ and $g = 9.81$ m s$^{-2}$. Consequently, we assume that $K_{Ic}$ is small enough to be negligible compared with $K_{I,stat}$, replacing equation (22) with

$$\dot{d} = \max\left(\frac{K_{\mathrm{I,stat}}}{K_{\mathrm{Ic}}|K'(0)|}, 0\right).\tag{24}$$

Note that this is equivalent to treating the small fracture toughness parameter $\kappa$ in Zarrinderakht et al. (2022, submitted ) as

zero.

Given (24), even very short cracks with small but positive stress intensity factors will propagate. We assume that such short cracks are readily available as material flaws in the ice shelf (although we consider them only at the predefined locations $x_{\mathrm{s}}(t)$ and $x_{\mathrm{b}}(t)$ as discussed above). We, therefore, allow crack propagation as described by equations (16)–(21), (23), and (24) to occur at any time $t = t_c$ in the solution of the viscous flow problem given by equations (1)–(13) at which an arbitrarily short,

vertical "seeding crack" at $x_{\mathrm{s}}(t)$ or $x_{\mathrm{b}}(t)$ is able to propagate. With vanishing fracture toughness, it is straightforward to show using the local analysis of short cracks in Weertman (1973) that we expect seed cracks to propagate if the effective pre-stress $\sigma_{nn}^{\mathrm{eff}}$ on the seed crack is tensile, while compressive effective pre-stress suppresses seed crack growth.



We refer to each instance at which one or more of these seeding cracks propagates further as a linear elastic fracture propagation event. Once a crack propagation event has run to completion, we revert to the viscous flow problem, with the new domain

boundary consisting of the pre-existing surface $\partial\Omega^-$ combined with the newly-propagated fracture or fractures. As above, since we work within the limit of small elastic displacements, the two sides of that crack are initially in the same location. Strictly speaking, the fracture, therefore, constitutes a contact area in the viscous flow model when we restart the solution of equations (1)–(13): any opening of the crack is infinitesimal due to the assumption of small strains, and therefore indistinguishable from a contact area at the order of approximation to which the viscous flow model is valid.

In general, after crack propagation is complete, the viscous flow model restarts again at the same time $t_c$ at which crack growth was initiated: even though equation (24) is a differential equation with respect to time, the amount of time elapsed during crack propagation is insignificant in the context of the viscous flow problem. It is however possible that the recomputation of the viscous pre-stress $\sigma_{ij}^{\mathrm{v}}$ after the crack is inserted into the domain and we revert to solving the viscous problem for velocity and stress (equations (2)–(13)) immediately leads to crack reactivation. That is, once the cracks have grown to a new steady

state given the initial pre-stress, the domain for the viscous flow problem will change simply because of the presence of that crack. As a result, the viscous stress field solution also changes, even before allowing further domain deformation according to the kinematic boundary condition (1). That new viscous stress field may be such that a new seed crack introduced at the tip of one of the previously grown cracks will again grow. From the perspective of the viscous flow problem, crack reactivation is due only to the adjustment of the viscous pre-stress, and happens instantaneously.

When this occurs, we assume that crack growth and recomputation of the viscous stress field alternate without adjustment of the domain geometry until new seed cracks no longer propagate (accepting that this could in fact be an iterative process that continues but leads to a finite total crack length introduced in the limit of many iterations between the recomputation of the viscous pre-stess and crack growth in the elastic fracture mechanics model).

By way of terminology, we refer to each collection of multiple fracture propagation events that occur due to the immediate

reactivation of a crack after the viscous pre-stress is recomputed as a fracture propagation episode. Each episode starts when a new seeding crack is able to propagate after a finite interval of purely viscous flow and terminates when the updated viscous pre-stress no longer leads to reactivation of the crack. To avoid semantic inconsistency, we also refer to a single fracture propagation event after which there is no immediate crack reactivation as a fracture propagation episode.

## 2.5   Numerical solution

The viscous component of the model is solved using the Elmer/Ice finite element package (Gagliardini et al., 2013; Todd et al., 2018; Amundson et al., 2022) with a prescribed stretching rate $V_X(t)$ and surface water pressure function $p_{\mathrm{fs}}(z,t)$; in practice, we use the prescriptions for $V_X$ and $p_{\mathrm{f,s}}$ given in section 2.6. To handle the high-stress concentrations near the newly-introduced crack tips, we used local mesh refinement. The normal stress condition (4) is handled using the regularization scheme of Durand et al. (2009, section 3.4), more commonly known as the "sea spring" scheme. As a corollary of that scheme, the solvability

condition (C2) need not be imposed: instead, the equivalent of equation (C2) for the sea spring scheme relates the residual of (C2) to an integral over the vertical velocity over the boundary to provide an approximation to the vertical velocity required to





ensure that the condition (C2) is satisfied on the next time step, leading to a rapid relaxation in time to the actual solution (see also Berg and Bassis, 2020).

The computational cost of implementing the contact conditions (5) for the viscous flow model proved excessive, mostly due
to the difficulty in identifying the geometry of surface contacts: that is, identifying which surface nodes have come into contact with, or crossed over into another part of the surface and re-entered the mesh. We have therefore defaulted to applying only the stress condition (4) without constraint in the viscous solver. This allows the mesh to overlap itself, which does occur at certain points in the computations as discussed in section 3 below. Although that geometrical overlap always remains small, not imposing the contact conditions (5) does alter the stress field and potentially affects results (see Zarrinderakht et al., 2022,
for an analysis of the equivalent elastic case).

After each time step of the viscous model (whose size is determined by a Courant-Friedrichs-Levy condition, henceforth "CFL condition"), small, vertical seeding cracks of equal length (given by the element size in the boundary element method used) are introduced at positions $x_s(t)$ and $x_b(t)$, and we test whether they propagate. To do so, we solve the linear elastic fracture dynamics component of the model using the boundary element method described in Zarrinderakht et al. (2022). If the
seeding cracks do not propagate, they are immediately removed again to avoid spurious growth of the crack at a rate that is dependent on the choice of time step and seeding crack length. As the crevasses evolve in time, refined mesh areas evolve with them, hence the mesh resolution changes with the evolution of geometry. This remeshing process is achieved through Gmsh package used in Elmer/Ice.

If the cracks propagate, we solve the fracture mechanics model forward in time until a new equilibrium is established,
imposing the contact conditions (B1) along the new crack faces only (since the viscous solver does not provide locations for any pre-existing contact areas as discussed). Subsequently, we immediately re-compute the viscous pre-stress $\sigma_{ij}^{\mathrm{v}}$ using the new domain geometry with the cracks in place (in practice, by taking a single time step of the viscous model with significantly reduced step length $\delta t/10$, $\delta t$ being determined by the CFL condition). Using that updated viscous pre-stress, we recompute the linear elastic fracture mechanics component of the model, iterating between the two solvers until the cracks no longer
propagate.

## 2.6  Prescription of forcing and scaling

The forcing parameters in the model are depth to water level $h_{\mathrm{w}}$ and stretching rate $V_X$. If we view the model as following a small part of an ice shelf, describing local fracture propagation and subsequent deformation around those fractures, then $V_X$ represents the larger-scale stretching of the shelf around the modeled section, in response to the large-scale state of stress of
the ice shelf.

The details of such a large-scale model are beyond the scope of this paper and will be presented elsewhere. To generate the computational results reported below, we choose instead a prescription of $V_X$ and $h_w$ that mimics simple ice-shelf behaviour. First, note that $V_X$ can be used to define an extensional stress parameter through

$$\tilde{R}_{xx} = 2BV_X^{1/n}.\tag{25}$$





This is the non-cryostatic contribution to extensional stress $\sigma_{xx}$ in the special case of a parallel-sided slab $\Omega$, for which $\sigma_{xx} = \rho_{\mathrm{i}} g(s - z) + \tilde{R}_{xx}$ throughout the domain; $\tilde{R}_{xx}$ is then written as plain $R_{xx}$ in the models of van der Veen (1998a,b), Lai et al. (2020), and Zarrinderakht et al. (2022, submitted ). We retain the tilde here to emphasize that $\tilde{R}_{xx}$ is no longer related trivially to the large-scale state of stress once the domain $\Omega$ has become fractured or deformed significantly. In fact, even the question of how to specify the large-scale state of stress then becomes more complicated as extensional stresses typically vary significantly

over a single ice thickness length scale in the presence of substantial crevassing.

A familiar observation from the theory of horizontally one-dimensional, unconfined (and unfractured) ice shelves (van der Veen, 1983) is that extensional stress $R_{xx}$ is proportional to ice thickness, as $R_{xx} = (1 - \rho_{\mathrm{i}}/\rho_{\mathrm{w}})\rho_{\mathrm{i}} g \bar{H}/2$, $\bar{H}$ being ice thickness (which is well-defined in the absence of local crevasse-induced surface or basal topography on the ice shelf). Here we adapt that simple result to impose $\tilde{R}_{xx}$ in the form

$$\tilde{R}_{xx}(t) = \tau^* \rho_{\mathrm{i}} g \bar{H}(t), \tag{26}$$

where $\tau^*$ is a constant, and we treat $\bar{H}$ as the mean ice thickness over the domain, defined as

$$\bar{H}(t) = \frac{|\Omega(t)|}{W(t)}, \tag{27}$$

and $|\Omega| = \int_\Omega 1 \mathrm{d}\Omega$ is the size of the domain as previously defined. Mean thickness $\bar{H}(t)$ is trivial to compute if its initial value $\bar{H}(0)$ as well as the initial and current domain width $W(t)$ are known: by conservation of mass, $|\Omega(t)| = |\Omega(0)| = \bar{H}(0)W(0)$

(appendix A) and hence $\bar{H}(t) = \bar{H}(0)W(0)/W(t)$. The parameter $\tau^*$ can be thought of loosely as measuring the degree of buttressing in the ice shelf: $\tau^* = (1 - \rho_{\mathrm{i}}/\rho_{\mathrm{w}})/2 = 0.05$ is an unbuttressed ice shelf in the absence of crevassing, while $\tau^* < 0.05$ is buttressed. Note that equations (25) and (26) combined imply that the stretching rate $V_X$ decreases significantly as mean ice thickness $\bar{H}$ does, as is also the case in an unconfined ice shelf.

If a prescription of $V_X$ should really be provided by a model for the large-scale mechanics of the ice shelf, then depth to

the water table $h_w$ should be computed from a surface hydrology model. Once more, we choose a simple formulation instead. Motivated by the definition (27), we define $\bar{s}$ in equation (10) by

$$\bar{s} = (1 - \rho_{\mathrm{i}}/\rho_{\mathrm{w}})\bar{H}; \tag{28}$$

from equation (C2) it can be shown that this is the mean surface elevation (defined by the area of the part of the domain above $z = 0$, divided by domain width) if there is no net loading due to surface water, and if no part of the lower boundary is above

sea level. Given that mean surface elevation, we prescribe depth to the water table as proportional to ice thickness, so

$$h_w(t) = \eta^* \bar{H}(t), \tag{29}$$

treating $\eta^*$ as constant.

Note that the parameters $\tau^*$ and $\eta^*$ are the equivalent of the parameters $\tau$ and $\eta$ in the purely elastic models of Zarrinderakht et al. (2022, submitted ), who use a simple parallel-sided slab geometry prior to crack propagation. In fact, as in Zarrinderakht

et al. (2022, submitted ), the model in this paper can be non-dimensionalized such that the only remaining parameters are the



forcing parameters $\tau^*$ and $\eta^*$, in addition to the density ratio $r = \rho_i/\rho_w$ and the dimensionless material constants $n$ and $\nu$. (For completeness, note that the value of Poisson's ratio only affects the computation of the value of elastic surface displacements, but not the sign of the gap opening $[u_i n_i]^+_-$ that determines whether the faces of a crack are in contact with each other or not (condition (B1)), and is therefore irrelevant to our results.)

For the linear elastic component of the model, equations (16)–(21), (23), and (24), the scales defined in Zarrinderakht et al. (2022) can be used for that non-dimensionalization, defining the thickness scale used as $H = \bar{H}(0)$, and scaling time $t - t_c$ rather than time $t$ itself with the crack propagation time scale defined in equation (22) of Zarrinderakht et al. (2022). Note that there is no equivalent of the parameter $\kappa$ in Zarrinderakht et al. (2022, submitted) here since we assume vanishing fracture toughness. For the viscous part of the model, equations (1)-(10), we define scales as

$$[\sigma] = \rho_i g H, \qquad [D] = B^{-n}(\rho_i g)^n H^n, \qquad [v] = B^{-n}(\rho_i g)^n H^{n+1}, \qquad [t_v] = 1/[D] = B^n(\rho_i g)^{-n} H^{-n}, \tag{30}$$

and corresponding dimensionless variables (indicated by an asterisk) as

$$\sigma^*_{ij} = \frac{\sigma_{ij}}{[\sigma]}, \qquad p^*_f = \frac{p_f}{[\sigma]}, \qquad D^*_{ij} = \frac{D_{ij}}{[D]}, \qquad v^* = \frac{v}{[v]}, \qquad (x^*, z^*) = \frac{(x,z)}{H}, \qquad W^* = \frac{W}{H}, \qquad t^* = \frac{t}{[t_v]}. \tag{31}$$

Beyond the reduction in the number of free parameters, the main insight of non-dimensionalizing the model is that the model is scale-invariant to changes in thickness $\bar{H}(0)$ if we keep constant values of $\tau^*$ and $\eta^*$: a simultaneous doubling in initial 380    thickness $\bar{H}(0)$ and width $W(0)$ while retaining the same initial domain shape (that is, the same initial domain shape if plotted in the $(x^*, z^*)$-plane) will lead to a doubling in stresses and a $2^{n+1}$-fold increase in velocity, but relative to the initial state, the same fractional changes in velocity, stress and domain shape, only $2^n$ times as fast.

In order to separate that scaling effect from the effect of different levels of forcing $\tau^*$ and $\eta^*$, we display our results in terms of the dimensionless variables defined above. For stress, translating from $\sigma^*_{ij}$ to the dimensional $\sigma_{ij}$ is straightforward, since it 385    requires multiplication by $\rho_i g H$: for $H = 500$ m, the unit of scale used below is then $[\sigma] = 4.5 \times 10^6$ Pa. Assuming Glen's law parameters $n = 3$ and $A = B^{-n} = 10^{-24}$ Pa$^{-3}$ $s^{-1}$ appropriate for temperatures around $-10$ C, the corresponding velocity unit is $[v] = 1400$ km yr$^{-1}$ and the unit of time is $[t_v] = 3.1$ hours. These velocity and time units are clearly not comparable to actual ice velocities and times over which ice shelves evolve viscously, and the numerical results below bear this out, with small dimensionless velocities and usually large dimensionless times at calving (the latter typically being $\sim 10^5$, corresponding 390    to about 30 years for the 500 m scale thickness above). The disparity between scale and realistic value comes about because actual deviatoric stresses in the ice scale as $(1 - \rho_i/\rho_w) \times [\sigma] \approx 0.1 \times [\sigma]$, and the factor $(1 - \rho_i/\rho_w) = 0.1$ is exponentiated by $n = 3$ to obtain velocity and time scale. We have kept the choice of scale $[\sigma] = \rho_i g H$ and the remaining choices in equations (30) here in order to remain consistent with Lai et al. (2020), Zarrinderakht et al. (2022) and Zarrinderakht et al. (submitted).





## 3 Results

### 3.1 Crevasse evolution

Figure 2 gives one example of progressive crevasse growth through the interplay of viscous deformation and elastic fracture propagation. Here, and in the remainder of this paper, we use the dimensionless variables of equation (31) with the asterisk decoration omitted for clarity. We also set $n = 3$ and $r = 0.89$, and report the parameter values $\tau^*$ and $\eta^*$ separately for each set of results.

Panel a shows the shape of the domain at several points in the time immediately after one of the crevasses has grown. We see a pattern of episodic lengthening of the cracks part-way across the remaining ice thickness at the crack location, followed by periods of widening of the crack through viscous deformation, and a corresponding stretching and thinning of the domain. In the example shown, the basal crevasse undergoes the most pronounced evolution through repeated crack propagation and subsequent widening. That process eventually leads to the complete propagation of the basal crevasse through the ice thickness, which we interpret as calving.

The growth of crevasses is visualized in greater detail in Figure 3. The main challenge in tracking crevasse growth is that the crack length variable $d$ of equation (24) is defined only during such fracture propagation episodes, since the viscous flow will subsequently move the crack faces apart from each other, meaning they become part of the wider ice surface and no longer define a crack. To address that issue, we define two slightly more complicated cumulative crack length variables, $d_\mathrm{b}^\mathrm{tot}$ and $d_\mathrm{t}^\mathrm{tot}$, as follows: $d_\mathrm{b}^\mathrm{tot}$ and $d_\mathrm{t}^\mathrm{tot}$ are constant between fracture propagation episodes and have initial values of zero. Consider then the $i$th fracture propagation episode, and let $H_\mathrm{b}$ and $H_\mathrm{t}$ be the ice thickness above the bottom and below the top cracks at the start of the episode, respectively, while $d_\mathrm{b}^{\mathrm{tot}-}$ and $d_\mathrm{t}^{\mathrm{tot}-}$ are the values of $d_\mathrm{b}^\mathrm{tot}$ and $d_\mathrm{t}^\mathrm{tot}$ immediately prior to the episode. During the fracture propagation episode, define the increase in $d_\mathrm{b}^\mathrm{tot}$ and $d_\mathrm{t}^\mathrm{tot}$ through

$$d_\mathrm{b}^\mathrm{tot} = d_\mathrm{b}^{\mathrm{tot}-} + \left(1 - d_\mathrm{b}^{\mathrm{tot}-}\right) \frac{d_\mathrm{b}}{H_\mathrm{b}}, \qquad d_\mathrm{t}^\mathrm{tot} = d_\mathrm{t}^{\mathrm{tot}-} + \left(1 - d_\mathrm{t}^{\mathrm{tot}-}\right) \frac{d_\mathrm{t}}{H_\mathrm{t}}, \tag{32}$$

where $d_\mathrm{b}$ and $d_\mathrm{t}$ are the lengths of the new bottom and top cracks during the episode as computed using equation (24). The final value of $d_\mathrm{b}^\mathrm{tot}$ and $d_\mathrm{b}^\mathrm{tot}$ then provide the values of $d_\mathrm{b}^{\mathrm{tot}-}$ and $d_\mathrm{b}^{\mathrm{tot}-}$ at the start of the next fracture propagation episode. The normalization by $\left(1 - d_\mathrm{b}^{\mathrm{tot}-}\right)/h_\mathrm{b}$ and $\left(1 - d_\mathrm{t}^{\mathrm{tot}-}\right)/h_\mathrm{t}$ is chosen so that $d_\mathrm{b}^\mathrm{tot}$ and $d_\mathrm{t}^\mathrm{tot}$ reach a value of unity when a crack finally propagates fully across the ice.

In Figure 3, an empty marker shows $d_\mathrm{b}^\mathrm{tot}$ and $d_\mathrm{t}^\mathrm{tot}$ at the end of fracture propagation events if the crack is immediately reactivated by recomputing the viscous pre-stress, while a solid marker indicates the end of a fracture propagation episode. We see that there are many more fracture propagation episodes than are shown in Figure 2a. Only the initial fracture propagation episode, starting from the same unfractured geometry as Zarrinderakht et al. (submitted), involves multiple individual events combined with immediate fracture reactivation (as indicated by the empty marker), with the bottom crack penetrating over halfway across the ice in that initial episode. The remaining fracture propagation episodes involve finite but generally moderate lengthening of bottom or surface cracks, either separately or simultaneously, in a single event. These events are separated in time by substantial intervals in which neither crevasse propagates, but in which viscous deformation continues.

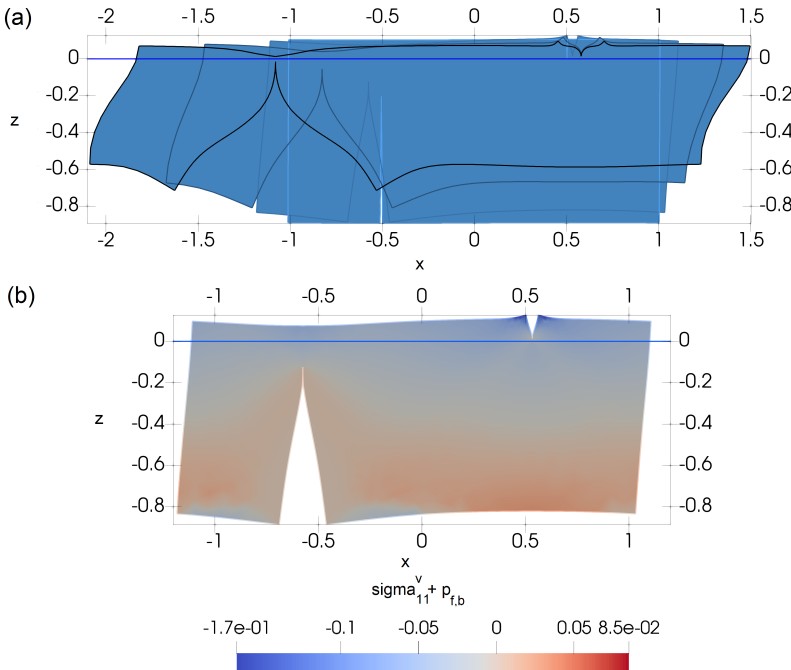

**Figure 2.** Domain evolution for $\tau^* = 0.06$, $\eta^* = 0.1$. a) The domain shape at $t = 74$, $t = 0.67 \times 10^4$, $t = 3.16 \times 10^4$ and $t = 4.7 \times 10^4$. The horizontal blue line indicates the sea level. Each profile is plotted immediately after the termination of a fracture propagation episode, with the profile at $t = 4.7 \times 10^4$ showing the last time step of the linear elastic fracture propagation solver before the tip of the bottom crack reaches the upper surface. Note that $t = 74$ marks the end of the initial fracture propagation episode that starts with the initial, rectangular domain shape. The crack has opened discernibly because we use a time step of $\delta t / 10$ to recompute the pre-stress, which allows for slight domain deformation during a fracture propagation episode, and accounts for the episode terminating for a small but non-zero time. b) The effective pre-stress $\sigma_{xx}^{\mathrm{eff}} = \sigma_{xx}^{\mathrm{v}} + p_{\mathrm{f,b}}$, immediately after the fracture propagation episode at time $t = 0.67 \times 10^4$.

Recall that propagation of a seed crack depends on the effective pre-stress being tensile as discussed in section 2.4. With the vertical cracks considered here (with normals parallel to the $x$-axis), compressive effective pre-stress corresponds to $\sigma_{nn}^{\mathrm{eff}} = \sigma_{xx}^{\mathrm{v}} + p_{\mathrm{f}} < 0$, suppressing crack growth, while a tensile effective pre-stress $\sigma_{xx}^{\mathrm{v}} + p_{\mathrm{f}} > 0$ facilitates the growth of the seeding

crack. Crack re-activation, therefore, corresponds to the updated pre-stress immediately after crack propagation satisfying $\sigma_{xx}^{\mathrm{v}} + p_{\mathrm{f}} > 0$ near the crack tip, and as described above, we only observe this during the initial fracture propagation episode.

Ultimately, all fracture propagation episodes in Figure 3 end with a compressive effective pre-stress near the crack tip: see for instance Figure 2b, where the blue shading above the bottom crack tip indicates a compressive effective pre-stress. A finite amount of time has to elapse before the effective pre-stress becomes tensile near the crevasse tip again. That change in the

sign of the effective pre-stress results from the changes in domain shape that occur due to viscous flow in the meantime. Only when the pre-stress has changed sign can a new fracture propagation episode can begin. The change in effective pre-stress $\sigma_{xx}^{\mathrm{v}} + p_{\mathrm{f}}$ around the tip of the basal crevasse are shown in Figure 4a. There we plot $\sigma_{xx}^{\mathrm{v}} + p_{\mathrm{f}}$ immediately after the termination



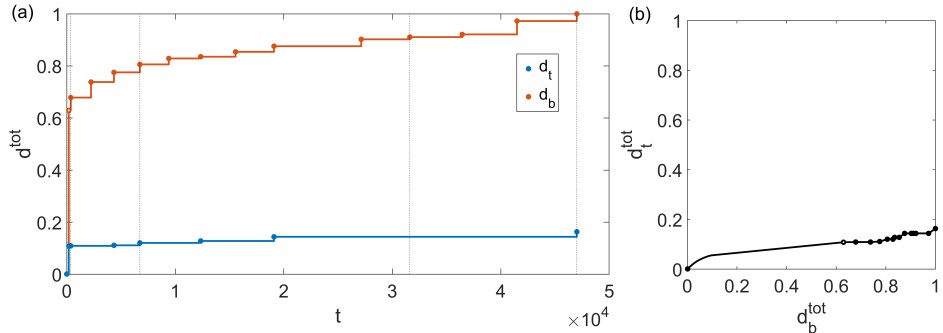

**Figure 3.** Cumulative fracture growth for $\tau^* = 0.06$, $\eta^* = 0.1$. a) The cumulative crack length variables $d_{\mathrm{b}}^{\mathrm{tot}}$ (red) and $d_{\mathrm{t}}^{\mathrm{tot}}$ (blue) plotted against time $t$. The times at which the domain shape is plotted in Figure 2a are indicated by vertical dashed lines. Note that for the 500 m initial ice thickness discussed at the end of section 2.6, $t = 10^4$ corresponds to 3.08 years, so calving in this example occurs within 15.5 years of starting with a parallel-sided slab geometry. Panel b) $d_{\mathrm{t}}^{\mathrm{tot}}$ plotted against $d_{\mathrm{b}}^{\mathrm{tot}}$. The curve resolves the growth of cracks *during* fracture propagation events when basal and surface cracks typically grow at different, variable rates. An empty circle indicates the termination of a fracture propagation event followed by immediate crack reactivation, and solid circles indicate the termination of a fracture propagation episode.

of the fracture propagation episode at $t = 0.67 \times 10^4$ in panel a1, and immediately before the subsequent fracture propagation episode at $t = 0.94 \times 10^4$ in panel a2. The change in sign of the effective pre-stress immediately above the crack tip between

the termination of the previous episode and the initiation of the next should be evident.

We see that the elastic fracture propagation mechanism generally "overshoots" the position at which the effective pre-stress remains on the cusp of being tensile, resulting in the finite time interval required for the change in sign of the effective pre-stress as described, and therefore in episodic crack propagation as described above. A corollary of compressive effective pre-stress near the crack tip following the termination of fracture propagation is however that the velocity field associated with that stress field points into the crack (Figure 4b1). The viscous flow solver used here does not prevent such inward velocities even where

the crack walls are in contact, as we have only been able to implement the contact conditions (5) for the simpler case of the elastic solver. The result in Figure 2b1 indicates that at least temporarily following the termination of fracture propagation, the viscous flow may keep the crevasse walls in contact, and a desirable addition for future versions of the model is to properly implement the contact conditions for the Stokes flow solver. In our solution, the inward velocity causes the crack walls to

overlap spuriously instead. By the time effective pre-stress has become tensile again, the velocity field has reverted to opening the crack once more prior to renewed fracture propagation (Figure 2b2).

### 3.2 Calving by surface crevasse propagation

The example illustrated in Figures 2–4 is one of calving by complete propagation of the basal crevasse, which occurs in discrete but relatively small increments until the crevasse tip reaches the upper ice surface. This result is loosely consistent with those

in Zarrinderakht et al. (2022, submitted ), where the growth of basal crevasses under continuous changes in forcing parameters



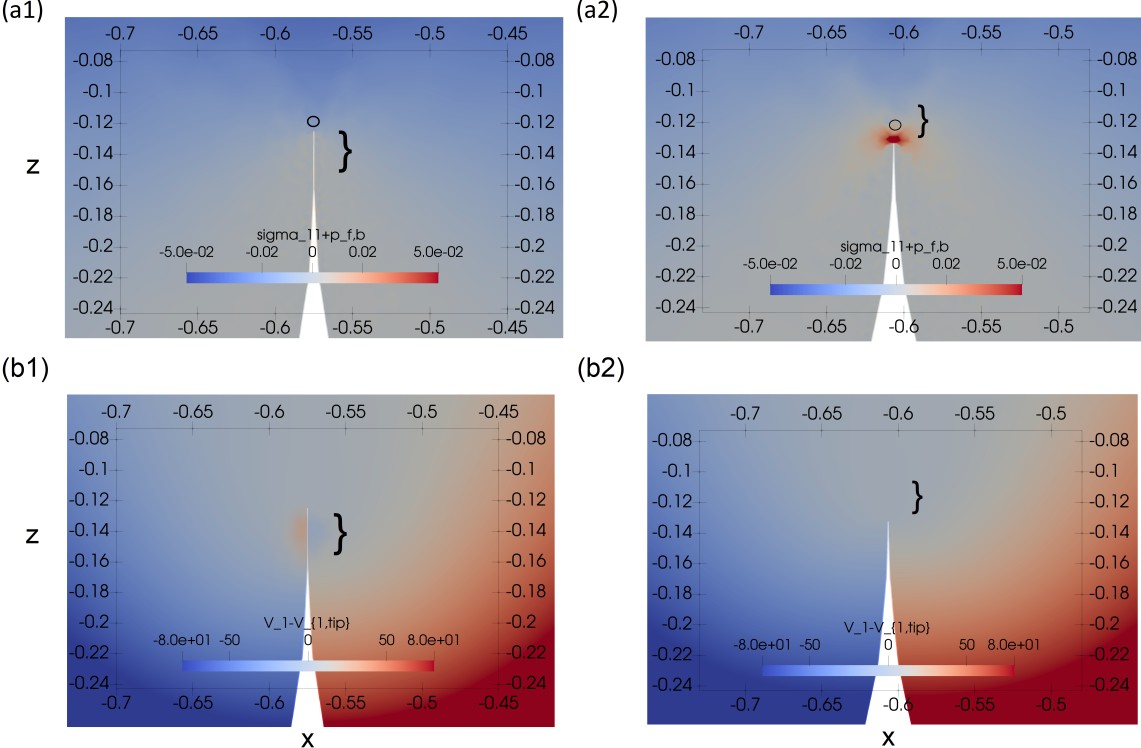

**Figure 4.** Viscous velocity and stress fields immediately after and before a crack propagation episode for $\tau^* = 0.06$ and $\eta^* = 0.1$. The effective pre-stress $\sigma_{xx}^{\mathrm{v}} + p_{\mathrm{f,b}}$ (row a) and horizontal velocity relative to the velocity at the crack tip, $v_1(x,z) - v_1(x_{\mathrm{tip}}, z_{\mathrm{tip}})$ (row b) immediately after the fracture propagation episode at $t = 0.27 \times 10^4$ (column 1) and immediately before the fracture propagation episode at $t = 0.94 \times 10^4$ (column 2). The vertical braces indicate the length of the crack that has just formed (column 1) or is about to form (column 2). The circles in row a indicate the centre of the first boundary element in the new seed crack introduced to test for further crack propagation. Note that a positive relative velocity to the left of the crack and a negative relative velocity to the right of the crack corresponds to a velocity field closing the crack as in panel b1.

is itself continuous. Note that instead of gradual changes in forcing parameters, the example in Figures 2–4 involves a gradual change in ice geometry, and growth of the basal crevasse is not continuous as such, but occurs in small steps, with the effect of geometric changes being mediated by changes in the viscous pre-stress.

A counterexample involving calving by surface crevasse growth is provided in Figures 5–7, using the same plotting scheme.
Most of the evolution of the crevasses here mirrors that in Figures 2–4: again, crack re-activation only occurs during the initial fracture propagation episode. Subsequent episodes involve a single fracture propagation event each and are again separated by finite time intervals during which the effective pre-stress changes from compressive near the crack tip at the end of the previous episode to tensile at the start of the next, as illustrated for the surface crevasse in Figure 7.



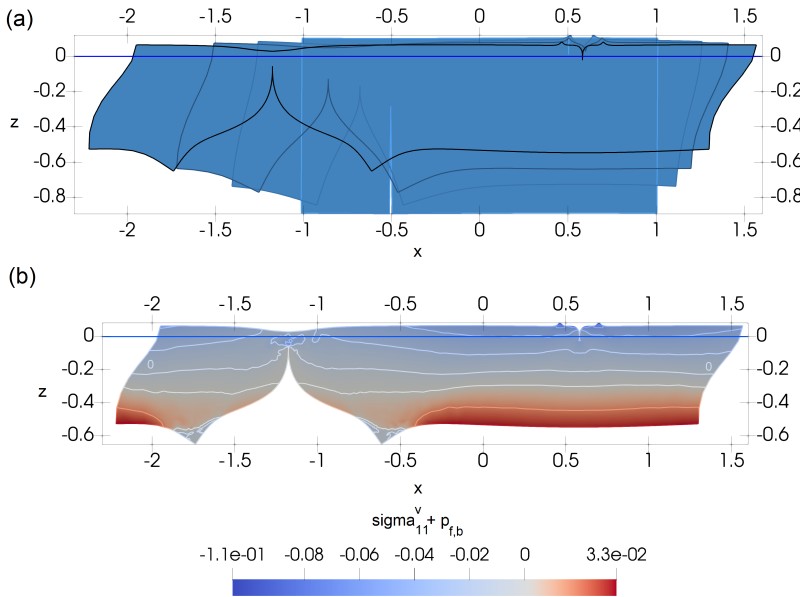

**Figure 5.** Domain evolution for $\tau^* = 0.05$, $\eta^* = 0.08$, same plotting scheme as Figure 2. a) The domain shape at $t = 1.1 \times 10^2$, $t = 0.269 \times 10^5$, $t = 0.545 \times 10^5$ and $t = 1.13 \times 10^5$. Each profile is plotted immediately after the termination of a fracture propagation episode. b) The effective pre-stress $\sigma_{xx}^{\text{eff}} = \sigma_{xx}^{\text{v}} + p_{\text{f,b}}$, immediately before the fracture propagation episode at time $t = 1.13 \times 10^5$. The zero contour is indicated by the numeral 0 at the left- and right-hand edges of the domain.

The key difference is that, despite the bottom crevasse growing substantially for most of the modelled time interval and the

surface crevasse only undergoing limited growth, calving eventually occurs due to a single fracture propagation event in which a surface crack grows all the way to the bottom of the ice at $t = 1.13 \times 10^5$. The abrupt growth of the surface crevasse to cause calving is again loosely consistent with the results in Zarrinderakht et al. (2022, submitted), who find that calving by surface cracks involves abrupt crevasse growth across a significant fraction of the total ice thickness $\bar{H}$.

### 3.3   Finite resolution

One caveat to the conclusion that crevasse growth is episodic is that we use significant finite element mesh refinement when we solve the viscous flow problem (and hence to determine the pre-stress), while the boundary element solver employed here uses a fixed element size on the crack. As a result, the finite element resolution near the crevasse tip is typically greater than the resolution of the boundary element method.

The conclusion that effective pre-stress needs to be tensile for a seed crack to grow in discrete terms corresponds in discrete

terms to the effective pre-stress at the center of the first boundary element in the seed crack being tensile. The location of that first element center is shown by an empty circle in Figures 4a and 7. Typically, the effective pre-stress between that boundary element centre and the crevasse tip becomes tensile before it does so at the boundary element centre.



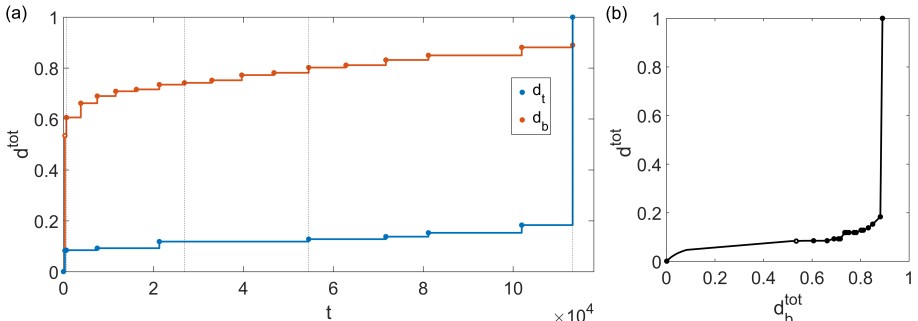

**Figure 6.** Cumulative fracture growth for $\tau^* = 0.05$, $\eta^* = 0.08$, using the same plotting scheme as Figure 6. a) The cumulative crack length variables $d_{\mathrm{b}}^{\mathrm{tot}}$ (red) and $d_{\mathrm{t}}^{\mathrm{tot}}$ (blue) plotted against time $t$. The times at which the domain shape is plotted in Figure 5a are indicated by vertical dashed lines. Panel b) $d_{\mathrm{t}}^{\mathrm{tot}}$ plotted against $d_{\mathrm{b}}^{\mathrm{tot}}$.

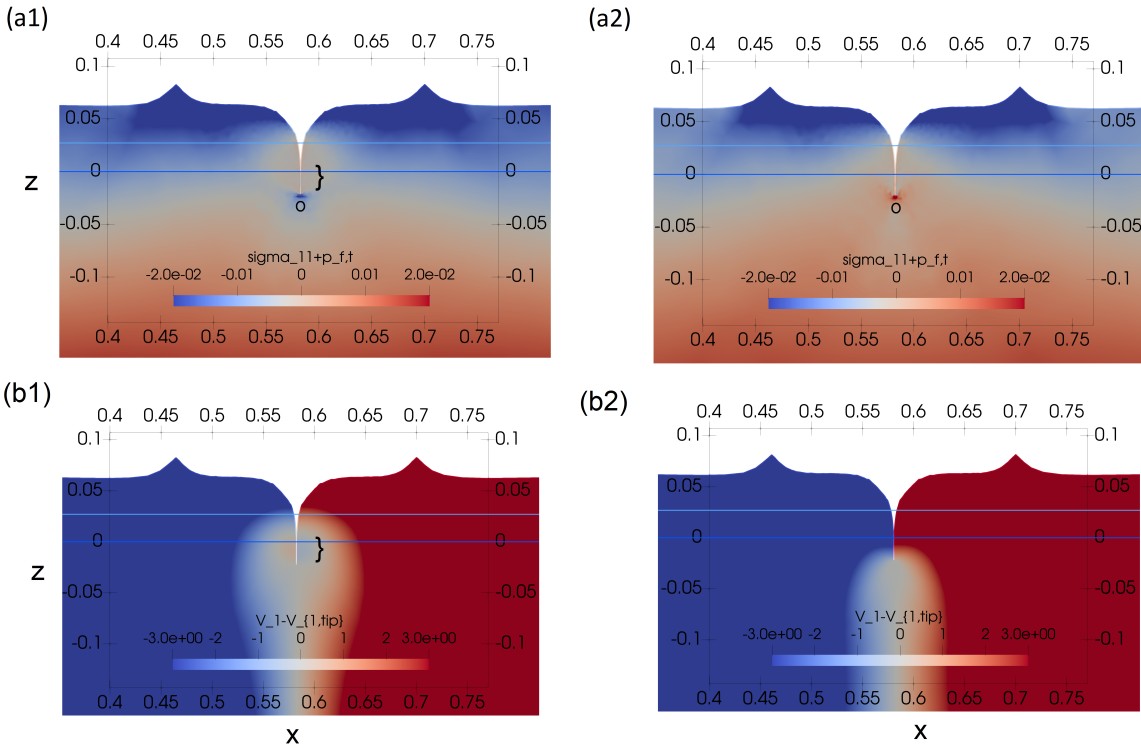

**Figure 7.** Viscous velocity and stress fields immediately after and before a crack propagation episode for $\tau^* = 0.05$ and $\eta^* = 0.08$. The effective pre-stress $\sigma_{xx}^{\mathrm{v}} + p_{\mathrm{f,s}}$ (row a) and horizontal velocity relative to the velocity at the crack tip, $v_1(x,z) - v_1(x_{\mathrm{tip}}, z_{\mathrm{tip}})$ (row b) immediately after the fracture propagation episode at $t = 0.918 \times 10^5$ (column 1) and immediately before the final fracture propagation episode at $t = 1.13 \times 10^5$ (column 2). The horizontal light blue line indicates the surface water level. The vertical braces indicate the length of the crack that has just formed (column 1).



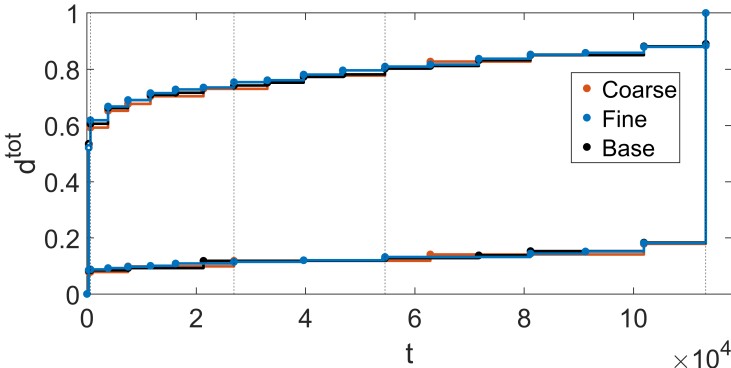

**Figure 8.** Recomputation of panel a) of Figure 6 using different element sizes for the boundary element solver used to compute fracture propagation. The "base" solution (black curve) uses an element size of 0.005, while the "coarse" and "fine" solutions use four times and a quarter of the base element size. The fine and base solutions agree well on the timing and extent of fracture propagation events.

To test for possible resolution-dependent effects that may result, we have recomputed the solution shown in Figure 5 with different boundary element sizes as shown in Figure 8. While there are some differences in the precise timing and magnitude of fracture propagation episodes, the results shown are robust under refinement in element size. In particular, the conclusion of episodic crack growth appears to be robust, since we see the same number of propagation events, occurring at approximately the same times, with crack re-activation only occurring during the initial episode.

### 3.4 The role of forcing parameters

Here we investigate more systematically how the choice of forcing parameters affects calving styles. Specifically, we consider parameter values ranging from $\tau^* = 0.03$ to $\tau^* = 0.1$, and $\eta^* = 0.05$ to $\eta^* = 0.12$. All solutions presented below were started with an initial domain width of $W = 2$ (and parallel-sided slab geometry of unit thickness), and terminated at $W = 4$, as viscous flow slows significantly once the domain stretches and stresses are reduced further, making longer runs very time-consuming.

In addition to solutions exhibiting calving as the result of intermittent fracture propagation combined with viscous deformation of the domain, we find solutions in which the initial fracture propagation episode itself leads to calving. These cases are illustrated in Figure 9a using $(d_{\mathrm{b}}^{\mathrm{tot}}, d_{\mathrm{t}}^{\mathrm{tot}})$ "phase planes" analogous to Figures 3b and 6b. In all the cases shown in Figure 9a, the "orbits" in phase space involve a single solid dot (meaning termination of a fracture propagation episode) at one of the calving boundaries $d_{\mathrm{b}}^{\mathrm{tot}} = 1$ or $d_{\mathrm{t}}^{\mathrm{tot}} = 1$.

We can group these examples of "instant" calving (with no viscous deformation) into two broad groups. First, there are those for which the first fracture propagation *event* leads to calving: these are the "orbits" without an empty circle (which would indicate the end of a fracture propagation event that is followed by immediate re-activation). An example is the green curve in Figure 9a. For the parameter values we have used here, this form of instant calving without crack reactivation occurs only by propagation of the top crack across the full ice thickness. For the corresponding parameter values, the purely elastic



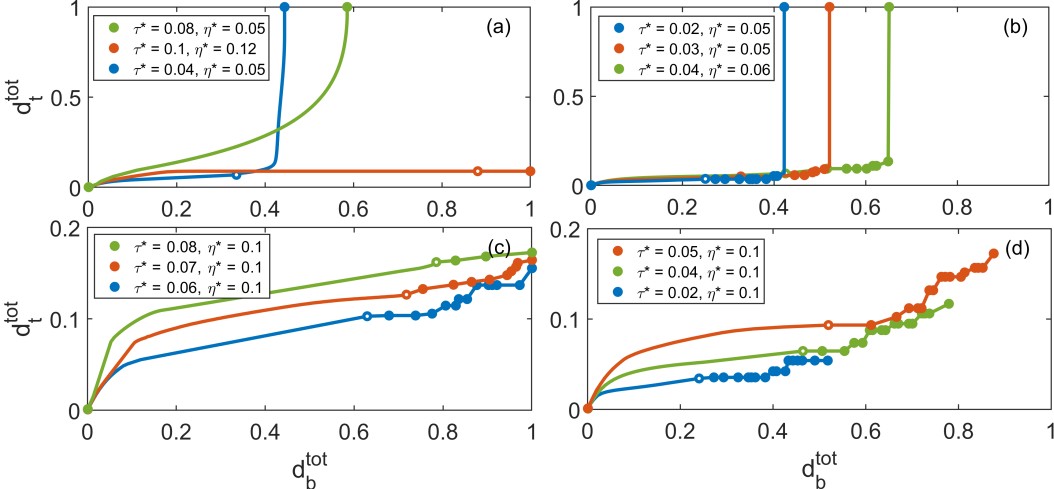

**Figure 9.** The equivalent of Figures 3b and 6b for different forcing parameter combinations, grouped by calving style. a) Instant calving, occurring during the first fracture propagation episode. b) Calving by crack propagation combined with viscous deformation, with the top crack penetrating fully through the ice. c) Same as panel b, but the bottom crack penetrates fully. d) No calving by the time $W = 4$ is reached.

model of Zarrinderakht et al. (submitted ) also predicts calving for these parameter combinations, as the pre-stress assumed in Zarrinderakht et al. (submitted ) corresponds to the viscous stress field in a parallel-sided slab before the introduction of any

cracks.

Second, there are the cases in which calving occurs in the first fracture propagation *episode*, but not the first *event*: in other words, the updated viscous pre-stress after the first fracture propagation event is essential in driving crevasse propagation across the entire ice thickness. These cases are identifiable through the empty circle along the "orbit" marking $(d_b^{tot}, d_t^{tot})$ at crack re-activation. For these parameter combinations, the purely elastic model of Zarrinderakht et al. (submitted ) would *not* predict

calving, precisely because it does not recompute the pre-stress after crack propagation.

Calving with crack reactivation can occur either through the propagation of the top (blue curve in Figure 9a) or bottom crack (red curve in Figure 9a). Even though there is no viscous deformation of the domain, the evolving viscous pre-stress plays a key role. Purely elastic fracture propagation models coupled with simplistic pre-stresses underestimate the range of parameter values that lead to calving even for as simple a geometry as a parallel-sided slab.

As discussed in sections 3.1–3.3, calving also occurs through the combination of episodic crack propagation combined with viscous deformation and can do so by surface and bottom crack propagation. We show further examples in Figures 9b and 9c. These share the general characteristics of the examples discussed in sections 3.1 and 3.2: calving by bottom crevasse growth (Figure 9c) is the result of multiple episodes of moderate fracture propagation growing the basal crevasse until it reaches the upper surface as in Figure 3, while calving by surface crevasse growth (Figure 9b) typically involves a sequence of similar

episodes of modest crack propagation followed by sudden propagation of a surface crack across most of the mean ice thickness $\bar{H}(t)$ at the time of calving, as previously shown in Figure 6.





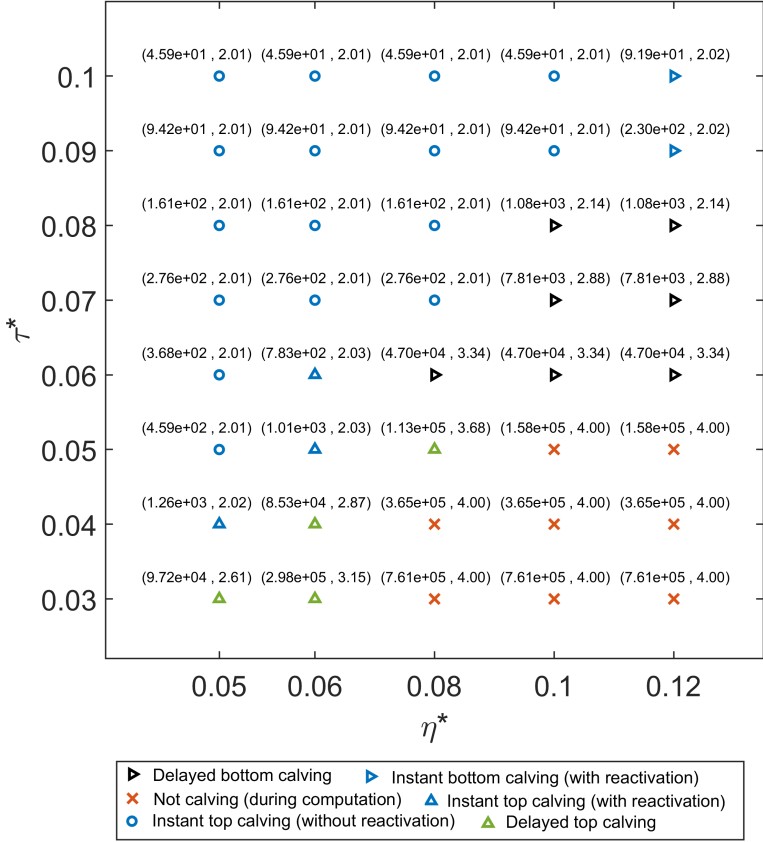

**Figure 10.** Calving styles for different parameter combinations $(\eta^*, \tau^*)$. The different marker types are identified in the Figure legend: "delayed" calving signifies calving after a combination of fracture propagation and viscous domain deformation, while "instant" calving implies calving during the fracture propagation episode that occurs starting with the initial rectangular domain shape. With and without reactivation refers to whether instant calving occurs as a result of multiple fracture propagation episodes or a single fracture propagation episode, respectively. "Not calving" indicates that calving did not occur before the domain had doubled in width (and ice thickness has halved). For each marker, we have given the time elapsed before calving and the domain width at the instant of calving in brackets above the marker; note that most instances of "instant" calving involve a non-zero time and a domain width of 2.02; this reflects that a single shortened time step is taken to compute the pre-stress before the elastic solver is run.

Lastly, cases that have not calved when $W = 4$ is reached are grouped into Figure 9d. Note that we are far from being able to say that any of these cases will never calve: the two cases with larger "stress" values $\tau^* = 0.04$ and $\tau^* = 0.05$ are terminated while there is clearly still continuing crevasse growth, and both seem likely to predict calving at some later time. The other
case involving smaller "stress" value of $\tau^* = 0.02$ is not particularly close to either crack having propagated across the full ice thickness, but we also cannot be sure that this case does not eventually lead to calving as well.





More systematically, we have aggregated most of our results in the plot of parameter space shown in Figure 10, with different markers indicating the style of calving that occurs. The results are perhaps unsurprising: given a water level $\eta^*$, larger "stress" $\tau^*$ favors instant calving, with intermediate values of $\tau^*$ corresponding to "delayed" calving through an interplay between crack propagation and viscous deformation, and no calving before $W = 4$ is reached for the lowest $\tau^*$. Where instant calving occurs with crack reactivation, this occurs at lower values of $\tau^*$ than instant calving without reactivation, but larger values of $\tau^*$ than delayed calving. Larger values of $\eta^*$ (larger depths to the water table below the upper surface, or equivalently, lower water) favor calving through bottom crack propagation or no calving at all, while smaller values of $\eta^*$ favor calving through surface crevasse penetration. Note that a water level of $\eta^* > 0.1$ corresponds to a surface water table below sea level, which may not be realistic (see also Krug et al., 2014).

## 3.5   The failure of simple models

Previous work in Krug et al. (2014) and Yu et al. (2017) has represented the linear elastic fracture mechanics portion of the problem using tabulated Green's functions for a simple parallel-sided slab geometry, rather than the actual, more complicated domain geometry. Replacing that model component with an actual elastic stress solver has been the major step in the present work. However, motivated by the desire for a simpler solution method inherent in the prior work in Krug et al. (2014) and Yu et al. (2017), we test here whether a simplified representation of geometry *and* of the stress field can still adequately predict calving in our model. We do so for an example where calving only occurs after there has been significant viscous deformation.

We consider crack propagation in the same parallel-sided, periodic slab geometry and offset crack configuration as the elastic fracture propagation model of Zarrinderakht et al. (submitted ), prescribing the same "shallow shelf" pre-stress as used in Zarrinderakht et al. (submitted ) as well as in the models of van der Veen (1998a,b), Lai et al. (2020), Zarrinderakht et al. (2022) and in one version of the model in Yu et al. (2017). We test whether a simple choice of slab thickness, domain width and stress $R_{xx}$ in that model can predict the actual calving behaviour seen in Figure 5–6.

Specifically, we set the slab thickness $H_s$ (where the subscript s stands for "simple" model) in the simple model of Zarrinderakht et al. (submitted ) equal to the (dimensional) mean ice thickness $\bar{H}(t_c)$ at the moment of calving in Figures 5–6, and correspondingly put the domain width $W_s$ in the simple model to the domain width $W(t_c)$ at the moment of calving in Figures 5–6. That choice of ice thickness is analogous to the choice in Krug et al. (2014). Similarly, we set the extensional stress parameter in the simple model to $R_{xx,s} = \tau^* \rho_i g \bar{H}(t_c)$ and the water level parameter $h_{w,s} = \eta^* \bar{H}(t_c)$, where $\tau^* = 0.05$ and $\eta^* = 0.08$ are the parameter value used in Figures 5–6. We also estimate the initial length of the surface and bottom cracks that we introduce into the parallel-sided slab geometry of the simple model of Zarrinderakht et al. (submitted ) to represent the pre-existing crevasses juts prior to calving as $d_b = \bar{H}(t_c) - H_b, d_t = \bar{H}(t_c) - H_t$, where $H_b$ and $H_t$ are the remaining ice thicknesses above the bottom and top crevasses just before the final fracture propagation episode in Figures 5–6.

In dimensionless terms (reintroducing asterisks for definiteness), this implies setting the parameter values $(\tau, \eta, W)$ of (Zarrinderakht et al., submitted ) to $(\tau^*, \eta^*, W(t_c)/\bar{H}(t_c))$, $\tau^*$ and $\eta^*$ being defined in section 2.6 (and given in the caption of Figure 2), and $W/\bar{H}$ being the aspect ratio of the domain (which is what $W$ in Zarrinderakht et al. (submitted ) represents) evaluated at the moment of calving. In Figure 11, we plot the corresponding (dimensionless) $(d_b, d_t)$ phase diagram of





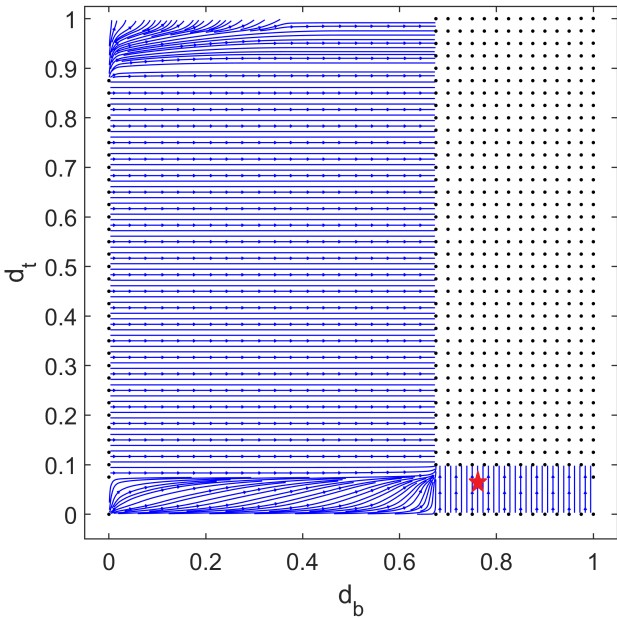

**Figure 11.** Phase plane plot for $(d_b, d_t)$ for a simplified, dimensionless model of linear elastic fracture propagation in a parallel-sided slab of ice with a unit thickness and width $W(t_c)/\bar{H}(t_c) = 5.53$ subject to dimensionless parameters $\tau = 0.05$, $\eta = 0.08$ and $\kappa = 0$ as defined in Zarrinderakht et al. (submitted). The blue curves show "orbits" of $(d_b(t), d_t(t))$ during elastic crack propagation for different initial conditions, with black dots indicating crack lengths for which there is no further crack growth (see Zarrinderakht et al., submitted). The red star indicates the position $(1 - H_b/\bar{H}(t_c), 1 - H_t/\bar{H}(t_c))$ in the phase plane, corresponding (in dimensionless terms) to our estimate of crack length based on crevasse size just prior to calving in Figures 5–6.

Zarrinderakht et al. (submitted), the red star representing the initial crack lengths inferred from the crevasse sizes in Figure 5–6 immediately prior to calving.

Calving by surface crack propagation (as in Figure 5) from the initial condition given by the red star would require the presence of an orbit from the star to the upper boundary of the plot box; that is clearly not the case. In fact, the only orbits that reach the upper boundary of the plot box start with relatively small bottom cracks and very substantial surface cracks. As shown in Zarrinderakht et al. (submitted) the phase plane shown in Figure 11 is quite distant in parameter space from the spontaneous calving by surface crack propagation; the simplified model considered here, therefore, does not seem to be a good candidate for predicting calving.

Note that Figure 11 is not an entirely fair test of the modelling procedure in Krug et al. (2014) or Yu et al. (2017): for instance, while these authors do use a simplified domain geometry analogous to the one used in Figure 11 to solve their elastic fracture propagation problem, they do use the correct viscous pre-stress, solved for using a Stokes flow solver and the actual domain shape with a crevasse that has partially penetrated (for Yu et al. (2017), this is true for at least one version of their model). As



we have seen in Figure 10, simply using the correct pre-stress as opposed to the "shallow shelf" pre-stress of Zarrinderakht et al. (submitted) can make the difference between predicting calving and not predicting calving (as is the case for parameter

combinations that predict instant calving with crack reactivation). We cannot, therefore, be certain that using the correct domain geometry when solving the elastic fracture propagation problem is key to predicting calving; a better representation of viscous pre-stress may suffice.

That said, if a model goes to the trouble of extracting the correct pre-stress from a Stokes flow solver to apply on the faces of a newly-propagating crack, there seems little to be gained by not posing the fracture mechanics problem in the correct

domain geometry since the boundary element solver that we use is relatively inexpensive computationally. A simplified model for calving really makes sense if it can be run in a largely decoupled fashion, using only a few parameters such as mean ice thickness and stretching rate from the viscous flow solver to predict calving. On that count, the results in Figure 11 indicate that such a simplified approach is unlikely to work.

## 4   Discussion

As anticipated in the introduction, our results confirm that purely elastic models for calving (that is, with a viscous pre-stress that is kept fixed and usually of simple "shallow shelf" form, see e.g. van der Veen, 1998a,b; Lai et al., 2020; Zarrinderakht et al., 2022, submitted) are not robust tools for predicting calving. There are two primary reasons. First, it is possible for calving to occur as the result of multiple successive crack propagation events, and second, the viscous deformation of partially incised crevasses may precondition the ice shelf to calve.

When calving occurs due to multiple successive crack propagation events, a crack propagates part-way through the ice and reaches a new equilibrium, with normal stresses (or more precisely, normal components of effective stress) being compressive near the crack tip in its new equilibrium position. However, after crack propagation has stopped, the stress field relaxes viscoelastically, and the (effective) stress field may become tensile near the crack tip. The crack then reactivates, which leads directly to calving in some of our computations, without any significant viscous deformation. Despite that lack of deformation,

a purely elastic model will still not account for the viscoelastic stress relaxation that leads to calving.

It is important to state that our model actually does not fully describe the effect of viscoelastic relaxation since it computes a fully viscous stress field immediately after crack propagation has run to completion, rather than resolving stress relaxation over the Maxwell time scale. Given that we find crack reactivation once the stress has fully relaxed to its viscous form, it is likely that crack reactivation actually occurs sooner, after only partial relaxation of the stress field. It is unclear whether this

has a significant effect on the total length of the fracture formed in a single fracture propagation episode (by which we mean a series of successive fracture propagation events between which the pre-stress relaxes viscoelastically, but the domain remains unchanged by viscous deformation): it is conceivable that crack reactivation after partial stress relaxation could lead to the crack propagating less far after each reactivation, but that there could also be more such propagation events in the propagation episode, leading to a comparable final crack length.





Figure 10 indicates cases where a purely elastic model correctly predicts calving with a blue circle in the parameter plane.
Cases in which the viscoelastic relaxation of stress leads to immediate calving by viscoelastic crack reactivation are shown with
a blue triangle. The latter cases are actually relatively rare. A much more common style of calving instead involves not only
the viscoelastic relaxation of stress following a calving event but the substantial viscous deformation of the ice (in the sense
of significant strains) before a new fracture again propagates. A purely elastic model obviously cannot capture that viscous
deformation.

     Our results in Figures 2–3 and 5–6 illustrate how the feedback between crack propagation and viscous deformation works:
cracks episodically propagate, generally by a moderate amount, with individual crack propagation episodes separated by long
periods of viscous deformation. Between individual fracture propagation episodes, the ice stretches and thins viscously, pulling
apart the crack faces that were previously formed to create fairly wide crevasses that significantly alter the shape of the domain.
Note that this form of episodic crack propagation contrasts with the conclusions of the purely elastic model in Zarrinderakht
et al. (2022, submitted ): for the latter, a slow change in forcing parameters leads to a continuous change in crevasse size, which
would equate here to a small amount of crack growth on each time step. That is not, however, what the coupled elastic-viscous
model predicts, and the relaxation of the viscous pre-stress after fracture propagation (omitted in purely elastic models) is
clearly key since it generates the compressive effective pre-stress that keeps the crack from reactivating for some time after
each fracture propagation episode.

     The change in domain shape that occurs between fracture propagation episodes appears to be key to repeated crack propa-
gation and eventual calving, through the associated change in viscous pre-stress, from a compressive effective pre-stress near
the crevasse tip immediately after a fracture propagation episode to tensile at the start of the next episode. It is worth asking
whether the change in the shape of the crevasses as a result of viscous deformation is key to calving, or whether the simple
widening of the domain is more important: as shown in Zarrinderakht et al. (submitted ), even a purely elastic model with
a periodic domain will permit calving at lower extensional stress if the domain is wider, due to the effect of elastic torques
generated by one crack on the opening of neighbouring cracks. A widening of the domain is also in-built into the model we
use here, and so a form of "delayed" calving (with a finite amount of viscous deformation prior to calving) is perhaps to be
expected purely due to the widening of the domain.

Figure 11 shows that widening alone is not sufficient to explain all the delayed calving cases in figure 10: the phase plane
plot shows how top and bottom cracks $(d_{\mathrm{b}}, d_{\mathrm{t}})$ would evolve if the viscous stress field was that which applies to a shallow ice
shelf, but the domain width had evolved to the same width as in the solution shown in 5–6 at the moment of calving. As Figure
11 shows, merely widening the domain by the same amount but not accounting for the effect of crevasses on the pre-stress
does not lead to calving.

If coupling viscous flow and fracture propagation is key to calving, then purely stress-based calving criteria for large-scale
ice sheet dynamics simulations (e.g. Pollard et al., 2015; Morlighem et al., 2016; Lai et al., 2020; Zarrinderakht et al., 2022)
are likely to miss much of the physics involved: they do not capture the formation and subsequent evolution of crevasses that
eventually lead to calving.



A better analogy exists between the physics in our model and damage mechanics (e.g. Pralong and Funk, 2005; Krug et al.,
2014), albeit a loose one. A damage mechanics model evolves a damage variable (as a function of $x$, $z$, and $t$ in a domain
like ours) and predicts the formation of an open crevasse when a critical level of damage has accumulated. In our model, we
instead evolve the surface geometry of the ice (which can be thought of as being encoded in the elevations of the upper and
lower ice surfaces as functions of $x$ and $t$), and calving occurs when the combination of stress, surface water level, and domain
geometry reaches some critical state. The key is that, like the damage variable of a damage mechanics model, the surface
geometry evolves over time (effectively as an analogue to damage), and calving does not happen instantly when a critical stress
is reached.

An important observation is that the time scale of surface geometry evolution *between* fracture propagation events is intrin-
sically the same as the advection time scale for the flow of ice in the ice shelf: unlike a damage mechanics model, where the
time scale of damage evolution is given by an independent model parameter ("$D$" in Krug et al. (2014) and Pralong and Funk
(2005), referred to as the "damage enhancement factor" in the latter paper), the evolution of surface and crevasse geometry due
to viscous flow in our model occurs on the same time scale over which the ice thins appreciably (or equally, on which $O(1)$
strains develop in the ice). That time scale is also the same as the time scale on which a piece of ice travels the length of the ice
shelf.

The most significant open challenge in making these results useful is in representing the fine detail that our model is able to
capture at the scale of an ice sheet. Clearly, modelling the detail of individual crevasses when running an ice sheet simulation
is not feasible or even desirable: a low-dimensional representation of the local crevasse geometry that can be modelled ade-
quately using a closed set of equations is necessary, in such a way that calving can be predicted robustly: that low-dimensional
representation can be forced by variables representing the large-scale state of stress, surface hydrology and mean thickness
of the ice shelf, but must not require additional information about local crevasse geometry, outside of the low-dimensional
representation. Such a low-dimensional representation would ideally be trained using the output of a highly resolved model
such as ours.

There are however a number of improvements to our model that should be made before attempting to coarse-grain it. One
obvious deficiency already discussed above is that we use a periodic domain of finite length $W$, and yet we know from the
results in Zarrinderakht et al. (submitted ) that fracture propagation is sensitive to the periodicity of the domain. As discussed
there and in Zarrinderakht et al. (2022), the model for elastic stress $\sigma_{ij}^{e}$ used here likely needs to be amended by taking account
of buoyancy effects in equations (19) and (B1) to remove the dependence on domain width, by making widely spaced cracks
less likely to propagate preferentially since vertical displacements between them can be larger (leading to bigger changes
in fluid pressure) even at small strains. (For completeness, note that this is not an issue for the viscous component of the
model, which correctly accounts for the buoyancy effect; the elastic model does not at leading order due to the assumption
of small elastic surface displacements, which can however become large if the domain is taken wide enough as explained in
Zarrinderakht et al. (2022, submitted ).)

A closely related issue is our insistence that there can be only two cracks, one on each ice surface. That choice allows a
relatively simple model set-up, with cracks in known locations propagating vertically. The plot of effective pre-stress $\sigma_{xx}^{\text{eff}}$ in





Figures 2b and 5b, however, suggests that additional seed cracks would grow (and would have grown prior to the domain shape
shown having been attained) if inserted in a large range of locations along the basal surface. $\sigma_{xx}^{\text{eff}}$ is the effective pre-stress
stress acting on a vertical seed crack, which is the likely favored direction in which new cracks should grow on a horizontal
surface. As the two plots show, $\sigma_{xx}^{\text{eff}}$ is tensile along most of the lower boundary, and in particular, where that lower boundary
is approximately horizontal, suggesting that seed cracks inserted there should grow. It is plausible that seed cracks at the upper
surface would also grow: Figures 2b and 5b show effective stress as defined in terms of the basal water pressure, and are
therefore not relevant to the formation of surface cracks.

This suggests a future improvement of the model should incorporate not only buoyancy effects on stresses at the boundary,
accounting for the effect of elastic surface displacements on the fluid pressure there, but also the possibility of multiple inter-
acting cracks that can have arbitrary orientation, in the expectation that a preferred crack spacing and orientation will emerge
spontaneously, rather than being imposed by the choice of initial domain width, and through the assumption of vertical, laterally
offset cracks.

There are additional problems to deal with in the future. As described, the model at present is able to impose the contact
conditions (5) only in the elastic component of the model, while the Stokes solver imposes a prescribed fluid pressure ev-
erywhere on the boundary as per equation (4). Closure of newly grown cracks due to viscous flow is, however, a common
observation (column 1 of Figures 4 and 7). With the Stokes flow solver as currently implemented, the crack walls can then
overlap (though they never do so by any significant amount), and compressive normal stress on the contact areas is lower than
would be the case if the contact conditions (5) were implemented as stated, though the effect on crevasse growth is currently
unclear. Implementation of the contact conditions in the viscous flow solver is another important model improvement for future
work.

A related issue is whether such contact areas should in fact be allowed to persist, or whether the ice on both sides should be
assumed to merge, healing the crack. Such healing is implicitly assumed in other free surface Stokes flow models for glaciers
(e.g. Jouvet et al., 2008), though these usually do not consider crack formation. Note that viscous flow here is relatively slow,
with many of the calving examples here taking years if we assume a plausible initial ice thickness of 500 m (see section 2.6).
Crack wall contact can therefore be maintained for significant lengths of time, and it seems likely that the ocean or surface water
that is typically in contact with the crack walls should freeze, healing the crack. This is likely to lead to qualitatively different
crack propagation patterns compared to those shown in Figures 3, 6 and 11, with crack propagation reversed potentially quite
quickly by healing, and relatively small net advances of the crevasse tip.

## 5  Conclusions

We have developed a simplified version of a viscoelastic model for the fracturing and local deformation of an ice shelf. A
separation of time scales between crack propagation time scale, Maxwell time, and the time scale for significant viscous
deformation allows us to model crevasse growth and calving through the coupling of a purely viscous model for large-scale
deformation with a linear elastic fracture mechanics model for crack propagation. As previously explored (though arguably



less explicitly) in Yu et al. (2017) and Krug et al. (2014), the coupling between viscous and elastic deformation occurs through a viscous pre-stress that acts on new crack faces (or rather, through an "effective pre-stress" that additionally accounts for the role of water pressure).

The main novelty in our paper is that we actually compute the elastic response to that imposed pre-stress by using a solver that uses the domain geometry at the time of crack propagation, rather than the idealized proxy shapes of Yu et al. (2017) and Krug et al. (2014). As in Yu et al. (2017), we track the subsequent viscous deformation of any crack that propagates only partially across the ice. That viscous deformation appears to be key in allowing crevasses to grow episodically until they reach the opposite side of the ice, specifically by changing the effective pre-stress around the tip of existing crevasses from

compressive to tensile.

We interpret a crevasse propagating through the full ice thickness as calving, in common with other two-dimensional models (Lai et al., 2020; Zarrinderakht et al., 2022). In three dimensions, that is not clear-cut, since we are not guaranteed that the crack is stable to variations in depth with respect to the transverse coordinate $y$, which is suppressed in our model. It is possible that the model really describes the complete opening of a rift in an ice shelf, and that this rift will then widen sideways,

presumably along any weakness presented by pre-existing crevasses that extend laterally from the location of the open rift (Scambos et al., 2000; Rignot et al., 2011). A model for subsequent rift propagation would then need to extend the present work to three dimensions (Lipovsky, 2020), ideally making use of the same separation of time scales to separate viscous and elastic model components. Even assuming to the contrary that the present model can be used to describe calving, making it useful in large-scale ice sheet models will likely require a simplified representation of the evolving ice surface that does not

capture all the detail of local ice geometry that our model does, but still accurately predicts growth of crevasses across the full thickness of the ice.

In closing, we note that a similar procedure to ours is used in coupled viscous flow-discrete element simulations such as Crawford et al. (2021), which use an ice geometry predicted by a viscous flow solver to then compute rapid fracture propagation using a brittle-elastic or viscoelastic discrete element model (Aström et al., 2013). The analogy is not perfect, as the viscous

pre-stress is not directly imposed on the discrete element model, despite the key role played by that pre-stress in our model. The discrete element model conversely has the advantage of automatically being able to predict arbitrary crack geometries, albeit at far more computational effort than our boundary element approach, but the complicated geometry that results is not used directly to restart the viscous flow solver, while our model does exactly that. Future work on the modelling approach used here will need to bridge the gap with discrete element models by allowing for a much larger number of arbitrarily located and

oriented cracks.

## Appendix A: Global mass conservation

Consider the evolution of mass inside the domain

$$\frac{\mathrm{d}}{\mathrm{d}t} \int_{\Omega} \rho_i \, \mathrm{d}t = \int_{\Omega} \frac{\partial \rho_i}{\partial t} \, \mathrm{d}\Omega + \int_{\partial\Omega} \rho_i \mathbf{V} \cdot \mathbf{n} \, \mathrm{d}\Gamma, \tag{A1}$$



where $\mathbf{V}$ is the velocity at which the domain boundary moves, and $\mathbf{n}$ is the outward-pointing unit normal as in the remainder
of the paper. On account of the surface being a material surface, $\mathbf{V} = \mathbf{v}$, so using the divergence theorem,

$$\frac{\mathrm{d}}{\mathrm{d}t} \int_\Omega \rho_\mathrm{i} \mathrm{d}\Omega = \int_\Omega \frac{\partial \rho_\mathrm{i}}{\partial t} + \nabla \cdot \mathbf{v} \mathrm{d}\Omega = 0, \tag{A2}$$

on account of local mass conservation, equation (3): this is nothing more than a standard conversion of local to global mass
conservation. The relevant insight is that, if ice density is constant, then $\int_\Omega \rho_\mathrm{i} \mathrm{d}\Omega = \rho_\mathrm{i} |\Omega|$, and hence

$$\frac{\mathrm{d}|\Omega|}{\mathrm{d}t} = 0, \tag{A3}$$

so domain area is conserved. We use this fact in section 2.6.

## Appendix B: Elastic contact conditions

Consider a contact area that existed immediately prior to the growth of a new crack, or the surfaces of the crack itself. Using
$[\cdot]_-^+$ to denote the jump between one side of the contact surface or crack as before, $-[u_i n_i]_-^+ = 0$ indicates that the two sides
of the crack or pre-existing contact area continue to be in contact since they are in contact at $t = t_\mathrm{c}$, and the displacement $u_i$ is
measured relative to initial position at $t = t_c$. Consequently, $-[u_i n_i]_-^+ > 0$ indicates that an infinitesimal gap has opened, and
$-[u_i n_i]_-^+ < 0$ is impossible. Where contact remains, the boundary conditions (5) imply that $[\sigma_{ij} n_j]_-^+ = 0$, $\sigma_{ij} n_i n_j \geq -p_\mathrm{f}$ and
$(\delta_{ij} - n_i n_j)\sigma_{jk} n_k = 0$; the constraint that $-[v_i n_i]_-^+ \geq 0$ merely states that the gap $-[u_i n_i]_-^+$ cannot become negative in the
future but has no implications for the solution of the elastostatic problem determining the displacements $u_i$. By contrast, where
contact has been lost, (4) ensures that $\sigma_{ij} n_i n_j = -p_\mathrm{f}$ and $(\delta_{ij} - n_i n_j)\sigma_{jk} n_k = 0$. Substituting the decomposition of stress (14)
and using the fact that $[\sigma_{ij}^\mathrm{v} n_j]_-^+ = 0$ on pre-existing contact areas and on the crack faces, it follows that the boundary conditions
satisfied by $\sigma_{ij}^\mathrm{e}$ on pre-existing contact areas and on the crack are in fact analogous to (5), but with $u_i$ taking the role of $v_i$, and
the elastic stresses modified by the pre-stress:

$$\left. \begin{array}{llll} \text{either} & \left(-[u_i n_i]_-^+ > 0 & \text{and} & -\sigma_{ij}^\mathrm{e} n_i n_j = \sigma_{ij}^\mathrm{v} n_i n_j + p_\mathrm{f}\right) \\ \text{or} & \left(-[u_i n_i]_-^+ = 0, & [\sigma_{ij}^\mathrm{v} n_j]_-^+ = 0, & \text{and} & -\sigma_{ij}^\mathrm{e} n_i n_j \geq \sigma_{ij}^\mathrm{v} n_i n_j + p_\mathrm{f}\right) \end{array} \right\}, \tag{B1a}$$

$$(\delta_{ij} - n_i n_j)\sigma_{jk}^\mathrm{e} n_k = -(\delta_{ij} - n_i n_j)\sigma_{jk}^\mathrm{v} n_k, \tag{B1b}$$

The right-hand sides in (B1) can also be rewritten using effective pre-stress as

$$\sigma_{ij}^\mathrm{v} n_i n_j + p_\mathrm{f} = \sigma_{ij}^\mathrm{eff} n_i n_j, \qquad (\delta_{ij} - n_i n_j)\sigma_{jk}^\mathrm{v} n_k = (\delta_{ij} - n_i n_j)\sigma_{jk}^\mathrm{eff} n_k, \tag{B2}$$

as described in section 2.3.





## Appendix C: Solvability conditions for the viscous and elastic problems

By integrating the $i = 2$ component of the local force balance relation (2) over the domain $\Omega$, we obtain on applying the divergence theorem

$$\int_\Omega \left( \frac{\partial \sigma_{2j}}{\partial x_j} - \rho_i g \right) \mathrm{d}\Omega = \int_{\partial\Omega} \sigma_{2j} n_j \mathrm{d}\Gamma - \int_\Omega \rho_i g \mathrm{d}\Omega = 0. \tag{C1}$$

Next, we can apply boundary conditions. The periodic conditions (6) ensure that the contributions to the surface integral from the two lateral boundaries cancel. On the upper and lower boundaries $\partial\Omega_\mathrm{b}$ and $\partial\Omega_\mathrm{s}$, the contributions to the surface integral from any contact areas vanish due to the condition $[\sigma_{ij}n_j]^+_- = 0$, and that cancellation remains the case if we replace $\sigma_{2j}n_j$ by $-p_\mathrm{f} n_2$ on contact areas; elsewhere $\sigma_{2j}n_j = -p_\mathrm{f} n_2$ by (4). Hence the constraint (C1) can be rewritten as

$$\int_{\partial\Omega_\mathrm{b}} p_\mathrm{fb} n_2 \mathrm{d}\Gamma + \int_{\partial\Omega_\mathrm{s}} p_\mathrm{fs} n_2 \mathrm{d}\Gamma + \int_\Omega \rho_i g \mathrm{d}\Omega = 0. \tag{C2}$$

Equation (C2) is simply an elaborate statement of Archimedes' principle: the last term is the weight of the ice, the second term is the weight of any surface water pressing down on the ice, while the first term is the buoyancy force that results from submersion of the ice in the ocean.

Associated with the solvability condition is an indeterminacy in the velocity solution $\mathbf{v}$ of the Stokes flow problem (2)–(13): adding a constant to the velocity field leads to an unchanged stress field (and an unchanged jump in normal velocity $[v_i n_i]^+_-$ across any contact areas) so that all the imposed stress boundary conditions continue to be satisfied. In addition, the periodic velocity boundary conditions (7) continue to hold. This indeterminacy is a variant of the problem considered in Berg and Bassis (2020). The vertical velocity component $v_2$ is in fact unique as we have to insist that the solvability condition (C2) continues to hold after any finite length of time, by ensuring that the ice continues to sit at exactly the right vertical level in the water.

More formally, differentiate (C2) with respect to time, using the kinematic boundary conditions. It is then easy to show that the addition of a constant to $v_2$ affects only the derivative of the surface integrals. For any trial solution $v_2$ that satisfies(2)–(13) but does not necessarily guarantee that (C2) will continue to hold at a future time $t$, the value of that constant can be determined by demanding that the derivative of the left-hand side of equation (C2) remains zero. This construction is however unnecessarily complicated, as we discuss in section 2.5: the "sea spring" regularization algorithm of Durand et al. (2009) is far simpler. For the horizontal velocity component $v_1$, we pick a unique solution by insisting that the velocity component $v_1$ at the point $(0, -r\bar{H}(0)/2)$ vanishes, where $\bar{H}(t)$ is the mean ice thickness at time $t$ as defined in equation (27). With an initially rectangular domain symmetric about $x = 0$, this location is the centroid of that initial domain shape.

Note that the symmetry condition $[\sigma_{ij}^\mathrm{v} n_j]^+_- = 0$, which arises from (B1a) on pre-existing contact areas as well as the new crack faces, combined with the traction-free condition (19) ensures that $\int_{\partial\Omega} \sigma_{ij}^\mathrm{e} n_j \mathrm{d}\Gamma = \int_\Omega \partial\sigma_{ij}^\mathrm{e}/\partial x_j \mathrm{d}\Omega = 0$ identically, and there is no additional solvability condition analogous to equation (C2) for the elastic problem consisting of equations (16)–(21) combined with the conditions (5b). It suffices to impose global force balance as a constraint on the viscous Stokes flow problem.



*Code availability.* The code for calculations is available from the corresponding author upon request.

790 *Author contributions.* MZ executed the research. MZ and TZ developed the code. MZ and CS wrote the paper, which all authors jointly edited. CS conceived the project.

*Competing interests.* The authors declare no competing interests.

*Acknowledgements.* MZ was supported by the ArcTrain NSERC CREATE graduate training program and by NSERC Discovery Grant funding to CS. CS acknowledges NSERC Discovery Grant RGPIN-2018- 04665. TZ was supported by the Finnish Academy COLD consortium
795 grant 322978. MZ and TZ would like to thank Peter Råback for advice regarding Elmer/Ice code implementation. Computations on CSC's platforms were supported by the HPC-Europa3 program, part of the European Union's Horizon 2020 research and innovation programme under grant agreement No.730897.



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
