# Peer review of "A leading-order viscoelastic model for crevasse propagation and calving in ice shelves"

_EGUsphere, 2023_

## Referee Comment (RC2)

**Comments to "A leading-order viscoelastic model for crevasse propagation and calving in ice shelves" by Zarrinderakht et al.**

**1 General comments**

In this paper, the authors coupled a boundary element method with the viscous ice-flow model, in order to combine the cracks propagation process with the viscous ice-flow model. The authors improved previous elastic models by using the real geometry at the time of crack propagation in their calculation. This work is potentially valuable to the cryosphere community, where the fracture and calving models are poorly developed.

However, I find the manuscript is hard to follow. This is partly because it's heavily citing other papers, some hasn't been published (Zarrinderakht et al., submitted), and some are not well known in glaciology. Furthermore, some of the key reference, which is used to describe the numerical solution, is wrongly cited. I hope the authors could improve the writing by being accurate, and bearing in mind that fracture mechanics is not widely implemented in ice-flow models, and some concepts are not well known (not as good as Stokes equations, for example). For example, when introducing equations, not only cite the original publication but also put the essential equations in the paper; also describe the physical meaning of the variables and equations in more details. I suggest a major revision to this manuscript. I hope the authors can put some effort in the writting style. There are some specific examples in the following comments.

Abstract: The authors mention they solved the fracture mechanics problem on the actual domain geometry. I think here actual domain geometry here doesn't mean real glacier/ice shelf, but solve the cracks boundary. This is slightly misleading. Nevertheless, can we use observational datasets to validate the model?

The key novelty of this work is the implementation of the boundary element method. A general description of boundary element method and why it's a good solution for the crack propagation problem (advantage) should be necessary?

**2 Specific comments**

L31: Unit of extensional stress is missing

L110-L114: Give the physical description of equations (5a).

Equation (7), extra comma

Citation of Figure 1 is missing. It should be somewhere in section 2.1. Furthermore, the first figure citation in the main text is Figure 5a, which is also unusual.

Equation (10), consider indicating  $h_w$  and s in Figure 1 sketch.

L156: a d $\rightarrow$ and

L192: where... the sentence is not finished (?)

L194:  $t_i \rightarrow t_c$

L205:  $\partial \Omega_b^+$  should be  $\partial \Omega_s^-$ ?

Equation (21): delete the negative sign before 0.

Line 237: Again, try to cite figures in order, e.g. Figure 2a?

L250: The authors are using stress and displacement matching method to estimate the static stress intensity factor. The stress matching method requires high degree of mesh resolution to obtain accurate value. Did the authors implement convergence studies on this problem? What would be the relative efficiency compare to the J integral approach?

L261: "We assume that such short cracks are readily available as material flaws in the ice shelf...". Does this sentence indicate cracks can potentially develop everywhere (with tensile effective pre-stress) with the rate defined by equation (24), although only at the predefined locations in this study?

L299: "sea spring" scheme is not a well known scheme in glaciology (at lease to me). Furthermore, the citation Durand et al., 2009 does not has section 3.4 and is not about handling the normal stress condition. Therefore, this part and the rest of that paragraph is quite unreadable to me.

section 2.5: How sensitive is the model to temporal  $(\delta t/10)$  and spatial mesh resolution?

L335: Describe the physical meaning of Rxx and Rxx rather than cite the variable from other references.

L363-: Again, these variables (same with  $\kappa$  mentioned a few times) are cited from other papers (especially unpublished) without explanation. Very hard (if possible) for the readers.

L386: correct the unit of temperature

L406: Are  $d_b^{tot}$  and  $d_t^{tot}$  crack lengths at the bottom and top, correspondingly?

L416: variables are repetitive

L417:  $h_b, h_t \rightarrow H_b, H_t$ ?

Figure 2: no units for t?

L436: 'can begin', delete 'can'

L447-L451: Figure 2b1 and Figure 2b2 should be Figure 4b1 and Figure 4b2

L461: involve-¿ involving

Figure 4: are there some plotting issues such that the axes are smaller than the domain?

Figure 7: same problem as Figure 4, the axis is offset, and there are two blue lines in the panels.

section 3.3: Could you present a figure with the mesh on top, so we can see the finite element mesh in the calculation domain as well as the boundary element?

L478-479: What are the different element sizes tested here? I think a proper mesh convergence study should be conducted and presented.

Section 3.4: In L346 and L362,  $\tau *$  and  $\eta *$  are described as 'a constant', while these are actually the two essential forcing parameters tested in section 3.4. For these important parameters, the physical meaning should be clear, and the chosen of the ranges should be justified.

---

## Author Comment (AC1)

**RC1: 'Comment on egusphere-2023-807', Jeremy Bassis, 08 Sep 2023**

**February 24, 2024**

**Reviewer Comment:** This manuscript is the second (or third?) paper in a trilogy of papers by the same authors focused on a boundary element model that is used to simulate the vertical propagation of surface and bottom crevasses in ice shelves and ice tongues. This manuscript focuses on a viscoelastic implementation of the model in which the (elastic) boundary element model is coupled with a full Stokes "flowlline" model. In this approach, the viscous stress is used to drive fracture propagation assuming that fractures propagate on a time scale that is rapid compared to that of viscous flow. This separation of scales allows the large scale viscous deformation of the ice to alter the stress near pre-existing crevasses providing a means to simulate the viscous widening and vertical propagation of crevasses.

The authors use this model to map out conditions where a combination of surface water filling surface crevasses and longitudinal stretching eventually permit crevasses to penetrate the entire ice thickness. This ductile deformation driven models result in a version of what is often called a "necking" instability evident in ductile failure of metals and other materials that exhibit creep on a time scale comparable to fracture.

There is an impressive amount of work that has gone into developing the boundary element model and then coupling it into a viscous Stokes flow model, which, as the authors clearly show, is not a trivial exercise and involves both nuance and significant numerical hurdles. Overall, I think this is an interesting manuscript that makes novel contributions to the field. The amount of details and technical exposition is, however, challenging. I had to read it several times and I'm still not entirely sure I understand some of the details. Below I give several suggestions that might ease the exposition for a broad readership. My overarching comment is that, as much as possible, it would be helpful for readers to be provided with a physical picture associated with model results. I also think some of the choices for non-dimensionalization are opaque making it harder to interpret the results. For example, the authors provide thorough descriptions of conditions when surface crevasses penetrate the entire ice thickness, but it isn't clear **why** these conditions lead to full thickness penetration. Is that once the crevasse penetrates deep enough, there is a critical height of water that allows the crevasse to rapidly penetrate the rest of the ice shelf?

Before I delve into some of my more detailed comments, I do want to point out that the mechanism for calving that the authors envision is one in which "starter cracks" in the middle of an ice shelf or ice tongue gradually propagate through the ice thickness is not the way calving occurs for most ice shelves and ice tongues. What we see most often is that fractures initiate along the margins and pinning points where stress is concentrated and then propagate horizontally across the ice shelf. This is not to say that the mechanism that the authors simulate doesn't happen (or isn't related), but it might be useful to signpost the fact that the two-dimensional model may not be adequate for many interesting situations that aren't well represented by flow lines. **Response:** Most of these points are reiterated in greater detail under "Major comments", and we have (hopefully!) responded to them there. There are two points that perhaps deserve attention here, however.

1) The point about rifting is well-put. In an ideal world, we would produce some version of the model in the paper for three-dimensional fracture. At the moment, that is probably best described as futuristic, though worth working towards. As requested, we have flagged the essential difference between our model geometry and more typical rifts in the updated introduction, stating in the re-written seventh paragraph of the introduction

Our focus is on the evolution of a periodic array of vertical crevasses in an ice shelf, spaced at distances comparable to the thickness of the ice. Note that this is an attractive in that it permits a two dimensional formulation that naturally extends prior work on purely elastic fracture propagation (Weertman, 1973, 1980; van der Veen 1998a, 1998b; Jiménez et al, 2018, Lai et al, 2020; Zarrinderakht et al, 2022, 2023), as well as being computationally tractable.

Full propagation of a crevasse across the whole ice thickness in our two-dimensional model need not correspond to calving, but may simply mark the initiation of an ice shelf rift that has to extend laterally in order to calve a tabular iceberg. It is however not the most common geometry for the initiation of big through-cutting rifts that are responsible for large calving events from ice shelves (e.g., Borstad et al 2013, Larour et al, 2021),: these rifts often start near prominent ice shelf pinning points, where there are high stress concentrations but the stress field is also likely to be intrinsically three-dimensional, which is computationally much harder to capture in a elastic or viscoelastic framework (see also Lipovsky, 2020; Forbes, 2023) Regardless of how a rift is initiated, its subsequent propagation is likely to be favoured by the presence of weaknesses in the ice shelf generated by partially propagated crevasses of the kind described by our model. In recent efforts to model rifting, such weaknesses have been described by simpler, depth-integrated damage mechanics approaches (Huth et al, 2021, 2023), and our model may be a stepping stone towards unifying them with a process-scale appropach to viscoelastic fracture mechanics.

2) Viscous necking: this is an interesting issue. We're hesitant to describe our results in terms of necking as the latter usually results from an imposed far-field tension, and strain rate increasing at that fixed tension when a viscous thread is narrowed locally, as discussed in Bassis and Ma (2015, *EPSL*) and the references therein (as well as various other texts in wider fluid dynamics like Wilson, 1988 *J. Fluid Mech.* vol 190, pp 561–570) The set up here at least superficially is different — as the hopefully improved description of boundary hopefully makes clear, we impose what is effectively a mean strain rate on the system, and it is not immediately obvious that necking as such will still occur under the boundary conditions used in the present set of numerical solutions (once again, see the discussion of those boundary conditions below).

The key connection with necking is likely the result visualized in Figure 4 of Bassis and Ma (2015): broadening of an incision enhances tensile stresses around the neck of ice, and it seems likely that the same effect is involved in reactivating cracks as, for instance, in figure 4 of the present paper. We have incorporated that in the updated fifth paragraph of section 4, where we now state

Between individual fracture propagation episodes, the ice stretches and thins viscously, pulling apart the crack faces that were previously formed to create fairly wide crevasses that significantly alter the shape of the domain. That change in shape leads to tenisle pre-stresses near the original crack tip. Note that what appears to be the same effect was previously described in Bassis and Ma (2015), who found that widening and blunting of an incision into a block of ice can lead to more tensile stresses at the tip of that incision (see Figure 4 in the latter paper).

**Reviewer comment:** Non-dimensional numbers. There are two main non-dimensional numbers in the manuscript  $\tau^*$  and  $\eta^*$  and these two numbers play a crucial role in the exposition. I had to

constantly flip back-and-forth to remind myself what the numerical values correspond to. As far as I understand it, both of these are defined with respect to some mean ice thickness. This is perhaps my own issue, but I am especially interested in certain limits that correspond to physical situations. Examining  $\tau^*$  first, my suspicion is that the relevant quantity is whether the deviator stress (or resistive stress) is larger or smaller than it would be for a freely-floating shelf/tongue that is spreading solely under its own weight. Intuitively, it would be a little bit surprising if fractures could penetrate the entire thickness of a freely floating ice tongue spreading solely under its own weight without \*some\* additional forcing. At the very least, it seems as though this should be marginally stable since we see freely floating ice exist stably in many environments.

I think it would be more intuitive to define a non-dimensional number that measures the ratio of the imposed longitudinal extension stresses relative to that of a freely spreading ice shelf such that a value of unity (or something!) corresponds to a freely spreading ice shelf and we can more easily assess whether it is being pulled apart with more or less vigor. I apologize if I am misinterpreting or imposing my own framework on the authors.

One aspect of the non-dimensionalization used by the authors that isn't intuitive to me is that because of the way  $\tau^*$  is defined, it seems like assuming a reference density of ice that is 800 kg/m3 (some firn/marine ice) vs 900 kg/m3 (mostly ice) will result in distinct values of  $\tau^*$  even for a freely floating ice tongue that is solely spreading under its own weight. Is that intentional? Or am I hung up on own mental rigidity.

**Response:** We have attempted to clarify this by moving the description of the forcing that the shelf is subjected to section 2.1, where the corresponding boundary conditions are stated.

The first thing to say is that  $\tau^*$  is strictly speaking a parameter that defines *stretching rate* rather than stress. The reason for this is discussed in more detail in the response to the next point. That said, for an unfractured domain,  $\tau^* = R_{xx}/[\rho_i g \bar{H}(t)]$  does measure the ratio of extensional stress to cryostatic normal stress at its base:  $\bar{H}(t)$  is *not* a fixed reference thickness, but the *mean ice thickness over the domain at time t*, as the updated text hopefully clarifies.

The extensional stress for an unbuttressed shelf is simply (1 - r)/2 times that cryostatic stress, where r is the density of "ice" to water density. As a result,  $\tau^*$  is then the product of 2/(1-r) with ratio of actual extensional stress to extensional stress in an equally thick unbuttressed ice shelf, (In other words,  $\tau^*$  is the desired stress ratio, but multiplied by a factor that might be seen as irrelevant; we have kept the definition as was to ensure consistency with the published papers by Zarrinderakht et al (2022) and Lai et al (2020).)

A value of  $\tau^* = (1 - r)/2$  then corresponds to an unbuttressed shelf, while  $\tau^* < (1 - r)/2$  is a buttressed ice shelf, and  $\tau^* > (1 - r)/2$  is an ... "underbuttressed" (?) ice shelf, where extensional stress has become more concentrated than for an unconfined ice shelf.

The critical value of (1 - r)/2 does depend on the density chosen for "ice", and indeed for water. We have chosen to use r = 0.89, corresponding to  $\rho_i = 916$  kg m-3 and  $\rho_w = 1030$  kg m-3, the latter representing ocean waters with a non-negligible salt content. Note that the dimensionless version of the model contains r explicitly as a parameter (in addition to  $\tau^*$ ,  $\eta^*$  and n), As a result, the critical value of (1 - r)/2 = 0.055 for an unbuttressed ice shelf is fixed throughout all of the numerical results. Consulting figure 10, it would appear that unbuttressed ice shelves *can* persist providing surface water level is sufficiently low (that is  $\eta^*$  is large enough, see below). There *is* an important caveat, which is that we are starting the domain size  $W^*$ ; the companion papers alluded to in the introduction above show that calving is sensitive to domain size for a periodic domain, and how the effect of vertical elastic displacements on fluid pressure at the lower ice surface (that is, "flexure" in the sense of elastic beam theory) must likely be included in the elastic model to remove that domain size dependence (at least for sufficiently wide domains). That is, however, beyond the scope of what we were able to do for this paper.

(We should add that even if an unbuttressed ice shelf was predicted to calve by the theory in this paper, that wouldn't necessarily be a contradiction of the fact that such ice masses exist: in many of our results, calving occurs by progressive fracture propagation and viscous deformation, and takes a finite amount of time, during which the domain thins. This represents the piece of ice in the domain being advected along an ice shelf as fracturing and deformation occur, unitl eventually the piece of ice breaks off after having travelled a finite distance from the point where it enters the unbuttressed ice shelf. That, in a sense, is the point of the model. Calving is need not be instantaneous, but can be the result of progressive deteriation of the ice shelf in the downstream direction.)

The relevant changes to the wording of section 2.1 (hopefully clarifying the definition of  $\tau^*$ ) are the following (paragraph 11 of the section onwards):

If we view the model domain as following a small, locally periodic part of an ice shelf (meaning that the geometry of the domain actually changes over many individual "wavelengths" W as in a typical multiscale setting, see e.g. Holmes (1995)), then  $V_X$  represents the larger-scale stretching of the shelf around the modeled section, in response to the large-scale state of stress of the ice shelf. The details of such a large-scale model are beyond the scope of this paper and will be presented elsewhere. Here, we simply prescribe  $V_X(t)$  at any given time as a function of mean ice thickness  $\overline{H}(t)$  across the domain at the same time, as follows.

If B and n are the usual parameter's in Glen's law (Cuffey and Paterson, 2010), then we can define a proxy for  $V_X$  with units of stress through

$$\tilde{R}_{xx}(t) = 2BV_X(t)^{1/n}.$$
(1)

The quantity  $R_{xx}$  equals the non-cryostatic extensional stress in the ice if the domain remains an unfractured rectangle (in which case  $\sigma_{xx} = \rho_i g(\bar{s} - z) + \tilde{R}_{xx}$ ), and corresponds to the viscous extensional stress parameter  $R_{xx}$  in previous models for elastic fracture propagation (e.g. van der Veen 1998a, 1998b; Zarrinderakht et al, 2022).

For such an unfractured rectangular domain, the standard theory of unbuttressed ice shelves (e.g. MacAyeal and Barcilon 1988) predicts that  $\tilde{R}_{xx}(t) = (1 - \rho_i/\rho_w))\rho_i g\bar{H}(t)/2$ . To account somewhat crudely for buttressing effects, we define  $V_X(t)$  by putting

$$\tilde{R}_{xx}(t) = \tau^* \rho_{\rm i} g \bar{H}(t), \qquad (2)$$

with  $\tau^*$  held at a constant value that represents the degree of buttressing;  $\tau^*$  is then the direct equivalent of the dimensionless extensional stress parameter  $\tau$  used in Zarrinderakht et al (2022, 2023). For a rectangular domain,  $\tau^* = (1 - \rho_i/\rho_w)/2$  then corresponds to an unbuttressed ice shelf, while smaller values  $\tau^*$  correspond to buttressing and larger values to extensional stress concentrations beyond those found in an unconfined ice shelf. In the present paper, we use a density ratio of  $\rho_i/\rho_w = 0.89$  throughout, meaning the critical value for an unbuttressed shelf is then  $\tau^* = 0.055$ . Importantly, we retain the definition of  $V_X$  through equations (12)–(13) regardless of whether the ice has been fractured and deformed into an irregular shape. For the complicated geometries that result,  $\tilde{R}_{xx}$  can no longer be interpreted directly as an extensional stress (if nothing else, the non-cryostatic contribution to extensional stress,  $\sigma_{xx} - \rho_i g(\bar{s} - z)$  depends on both, x and z in that case).

Note that the mean ice thickness H used in the prescriptions for stretching rate  $V_X$  and surface water level  $h_w$  can be defined simply as

$$\bar{H}(t) = \frac{|\Omega(t)|}{W(t)},\tag{3}$$

where  $|\Omega(t)| = \int_{\Omega(t)} 1 d\Omega$  is the size of the domain. Conservation of mass (appendix A) then ensures that  $\bar{H}(t)$  is simply inversely related to W(t) as  $\bar{H}(t) = \bar{H}(0)W(0)/W(t)$ .

We also remind the reader of this at the start of the revised section 3.4, stating

Recall that  $\tau^*$  defines the lateral stretching rate of the domain through equations (12)–(13), but is intended to minic the effect of extensional stresses, with  $\tau^* = (1 - r)/2 = 0.055$  being the value appropriate to an unbuttressed ice shelf.  $\eta^*$  measures the depth of the surface water table below the upper surface, scaled to mean ice thickness:  $\eta^* = 0$  represent completely full surface crevasses,  $\eta^* = (1 - r) = 0.11$  represents surface water level at sea level, and  $\eta^* = 1$  represents dry surface crevasses regardless of depth.

**Reviewer comment:** Similarly, it is hard to ferret out the physical significance of the  $\eta^*$  parameter. What I would really like to know for the  $\eta^*$  parameter is when are surface crevasses water free? Does this correspond to  $\eta^* = 0$  (was this not considered)? Are all of the simulations done for surface crevasses with some amount of water (provided they propagate deep enough)? I think  $\eta^* = 0.05$ corresponds to a water table at the mean surface height (but this, again depends on the ratio of the density of ice to water, right?). I think in the past the water table was defined relative to crevasse depth so that some fracture of the crevasse was water filled. I may just be very inflexible on this. Another possibility would be to annotate the graphs more to show where on graphs crevasses are water free and where crevasses are "brimful" and also to show where the ice is being pulled apart with a stress larger than or smaller than the stress associated with a freely floating ice shelf.

**Response:** We have also tried to clarify this by providing all the relevant information in section 2.1. Water level is at a depth  $h_w$  below the mean surface elevation  $\bar{s}$  (by which we mean a spatial mean at any given time t!), and we express that depth as a fraction of mean ice thickness  $\bar{H}$  (again, the spatial mean):  $h_w = \eta^* \bar{H}$

 $\bar{s}$  is related to  $\bar{H}$  as

$$\bar{s} = (1 - r)\bar{H},$$

which we now derive explicitly in appendix C, although it is just a restatement of Archimedes' principle.

In other words, the water level relative to sea level is at

$$z = \bar{s} - h_w = \bar{s} - \eta^* H = (1 - r - \eta^*) H,$$

z = 0 being defined by sea level. As a consequence, a surface water table at sea level corresponds to

 $\eta^* = 1 - r.$

A "brimful" water level is approximately

$$\eta^* = 0;$$

we say "approximately" because  $\bar{s}$  is the mean over the upper surface elevation, and unless the upper surface is flat, there will be higher and lower spots than  $z = \bar{s}$  on the surface; these however depend on the specifics of the solution and the distance between water level and those high spots can only be determined a posteriori. For such an uneven upper surface,  $\eta^* = 0$  will mean standing water on some parts of the upper surface, with other parts of the surface expposed above the surface water. Lastly, dry surface crevasses are essentially assured if we put

$$\eta^* = 1$$

in which case surface water level is at the elevation of the mean lower boundary of the ice.

The revised wording of setion 2.1, paragraphs 8 onwards, tries to make the interpretation of  $\eta^*$  clearer:

To specify the fluid pressure  $p_f$ , we assume a hydrostatic increase in water pressure below sea level on any part of the bottom surface  $\partial \Omega_b$ ,

$$p_{\rm f} = p_{\rm f,b}(z) = \rho_{\rm w} g \max(-z,0),$$
(4)

where  $\rho_{\rm w}$  is the density of water. On the upper surface  $\partial \Omega_{\rm s}$ , we assume that there is a prescribed water level below which water pressure increases hydrostatically. To remain consistant with previous work in Zarrinderakht et al (2022), we express that water as a depth  $h_{\rm w}(t)$  below the spatial mean upper of surface elevation  $\bar{s}(t)$

$$p_{\rm f} = p_{\rm f,s}(z,t) = \max(\rho_{\rm w}g(\bar{s}(t) - h_{\rm w} - z), 0).$$
(5)

Equation (C2) of appendix C allows us to relate  $\bar{s}(t)$  to mean ice thickness  $\bar{H}(t)$  across the domain through a simple flotation criterion as  $\bar{s} = (1 - \rho_i / \rho_w) \bar{H}$ .

The lateral boundary conditions described immediately below ensure that the ice stretches horizontally over time, and therefore thins correspondingly. In a real ice shelf, the surface water level is determed by a near-surface hydrological system, which si beyond the scope of the present paper. We assume instead that, as the ice stretches and thins, the surface water level  $h_w$  remains equal to a constant fraction  $\eta^*$  of the mean ice thickness  $\bar{H}(t)$  over the domain at time t,

$$h_{\rm w}(t) = \eta^* \bar{H}(t). \tag{6}$$

 $\eta^*$  is then the direct equivalent of the dimensionless surface water level parameter  $\eta$  used in Zarrinderakht et al (2022,2023). As in those papers, we assume a density ratio  $\rho_w/i = 0.89$ . In practice, a value of  $\eta = 0$  then corresponds to a water level at the mean upper surface  $z = \bar{s}$ , a value of  $\eta = (1 - \rho_i/w) = 0.11$  corresponds to a surface water level at sea level z = 0, and a value of  $\eta = 1$ corresponds to a surface water level equal to the mean lower surface elevation (and therefore, for almost all practical purposes, a permanently dry upper surface).

**Reviewer comment:** Horizontal boundary conditions. I'm still a little bit confused about the horizontal boundary conditions imposed on the full Stokes model. In fracture mechanics we tend to distinguish between displacement and stress controlled loading. In displacement controlled loading, you take a sample and pull it apart with a constant velocity or step-wise displacement. In stress controlled loading a stress is applied to the edges of the sample. Stress controlled loading tends to result in catastrophic failure whereas displacement controlled loading tends to result in quasi-stable fracture propagation. I'm still not entirely sure which situation the authors are considering here. I think (?) the authors have imposed a horizontal velocity to the edges of the domain, but is a bit confusing because they talk about applying stretching rates.

This would seem to be a velocity or displacement controlled boundary condition. For ice shelves and ice tongues, stress controlled seems the more appropriate boundary condition and I think that a more physically obvious boundary condition would be to impose a strain rate (velocity gradient) at the boundary. (I apologize if this is exactly what the authors did!). A stress controlled boundary condition would imply (I think) that the elastic strains at the boundary vanish rather than the displacements. Boundary conditions can be really important and I would encourage the authors introduce a separate section to describe the boundary condition, whether they are displacement or stress controlled. I'm not insisting that the authors change the boundary conditions. Just explain what they are. There is text to explain this, it was just scattered in a couple of different places making it harder to follow and the notation wasn't always clear. **Response:** The description of the boundary conditions that are actually applied was indeed unnecessarily scattered in the original submission (mostly because we kept the specification of  $V_X$  and  $h_w$  in the section of scaling).

Rather than introducing a new section, we have moved the relevant material from the original section 2.6 to section 2.1, whre the quasi-periodic boundary conditions on velocity (and hence the stretching rate) are first introduced. The new text in section 2.1 that describes the lateral boundary conditions in fact also defines  $\tau^*$ , and is given in response to second review comment (re dimensionless numbers) above. To save space, we do not repeat it here.

It is correct that we specify a stretching rate  $V_X$  at any time t, in terms of the current mean ice thickness  $\overline{H}(t)$ . That is not necessarily how an actual ice shelf works, but is retained here because the imposition of a stress condition is computationally more complicated (as well as theoretically, see below). Without wishing to argue that a student-led paper should be held to a different standard, a stress constraint simply proved beyond the scope of a single PhD in the context of the sequence of papers identified at the start of the review.

With a prescribed stretching rate, it turns out that the appropriate lateral boundary ocnditions for the elastic problem (which describes the dynamics at a time scale much less than a Maxwell time) are periodic in stress and displacement, neither being prescribed as such. We provide an abbreviatead argument for this (without going through the relevant rescaling of time formally) in section 2.4

Similarly, boundary conditions at the lateral boundaries mirror (9) and (10), now in the purely periodic form

$$[u_i]_1^{\mathbf{r}} = [\sigma_{ij}^e]_j^{\mathbf{r}} n_j = 0.$$

$$\tag{7}$$

Unlike in equation (10), the stretching rate  $V_X$  does not appear in equation (25)1. Assuming that  $V_X$  remains unchanged during crack propagation, this is the case because the viscous strain strain acrued over a time much less than a Maxwell time is small compared with a typical elastic strain. (Formally, if lateral stretching has a significant effect on the Stokes flow problem (4)–(17), then stresses scale as  $BV_X^{1/n}$ , and elastic strains corresponding to the same level of stress scale as  $BV_X^{1/n}/E \sim V_X t_M$ , where we have defined Maxwell time as the ratio of viscosity to Young's modulus,  $t_M = BV_X^{1/n-1}/E$ . Assuming that the crack propagation time scale is much less than the Maxwell time,  $t-t_c \ll t_M$ , then viscous strain  $V_X(t-t_c)$  during crack propagation is indeed much less than elastic strain  $V_X t_M$ .) If we think of the modelled domain as a small part of an actual ice shelf, then  $V_X$  is effectively a

macroscopic strain rate, averaging over the details of what happens in the domain. That macroscopic strain rate should indeed be set by a stress constraint, which is what the following passage of the revised section 2.1 refers to

If we view the model domain as following a small, locally periodic part of an ice shelf (meaning that the geometry of the domain actually changes over many individual "wavelengths" W as in a typical multiscale setting, see e.g. Holmes (1995)), then  $V_X$  represents the larger-scale stretching of the shelf around the modeled section, in response to the large-scale state of stress of the ice shelf. The details of such a large-scale model are beyond the scope of this paper and will be presented elsewhere. Without pre-empting too much of that paper in preparation for "elsewhere", suffice it to say the following: The model as formulated in the present paper would continue to hold for local deformation at the ice thickness scale (as the "inner" model of a multiple scales expansion), with the following two alterations:

1.  $V_X$  would not be defined simply by an algebraic relationship with mean ice thickness at time t, but by current domain geometry and a descriptor of large-scale stress through an additional

integral constraint, of the form

$$\bar{H}\Sigma = \int_{\partial\Omega_r} \sigma_{1j} n_j d\Gamma + \frac{\rho_w g}{2} \left[ \min(b_r, 0)^2 - \max(\bar{s} - h_w - s_r, 0)^2 \right] + \frac{1}{2} \left( 1 - \frac{\rho_i}{\rho_w} \right) \rho_i g \bar{H}^2, \quad (8)$$

where  $b_r$  and  $s_r$  are the lower and upper surface elevations at the ends of the right-lateral boundary  $\partial \Omega_r$ , and **n** is the outward-pointing unit normal to that boundary.

This perhaps overly complicated looking quantity  $\Sigma$  defined by equation (8) above reduces to the more familiar "resistive stress'  $R_{xx} = \int_{b(x,t)}^{s(x,t)} [\sigma_{xx}(x,z,t) - \rho_i g(s(x,t)-z)] dz$ . The reason for using  $\Sigma$  rather than  $R_{xx}$  in the present case is that, for an irregular domain with no "simple" far field,  $R_{xx}$  actually varies by O(1) over the local domain considered here, while  $\Sigma(t)$  above turns out to evaluate to the same quantity no matter which cut I make across the ice from bottom to top in the domain considered here.

2. In addition, a stress constraint means replacing the periodic displacement boundary conditions for the elastic problem,  $[u_i]_1^r$  with

$$[u_i]_1^r = U_X W,$$

where  $U_X$  is a constant "mean strain" that is controlled by another integral constraint on the elastic component of stress  $\sigma_{ii}^{e}$ , of the form

$$0 = \int_{\partial \Omega_r} \sigma_{1j}^{\rm e} n_j \mathrm{d}\Gamma.$$

(Zero because  $\sigma_{ij}^{e}$  gets added to to  $\sigma_{ij}^{v}$  during crack propagation to form the full Cauchy stress  $\sigma_{ij}$ , and  $\sigma_{ij}^{v}$  already satisfies equation (8) above.

In case this is of interest, the releavnt model for larger-scale stress and evolution of the ice shelf alluded to in the excerpt from the paper above takes the form (for a flowline)

$$(\bar{H}\Sigma)_X + (\bar{H}\Sigma_Y)_Y - \left(1 - \frac{\rho_i}{\rho_w}\right)\rho_i g\bar{H}\bar{H}_X = 0, \qquad (9)$$

$$\bar{H}_t + (\bar{H}V)_X = 0,$$
 (10)

assuming no surface mass balance, where subscripts X and t denote differentiation with respect to the "outer" variable X (essentialy, x rescaled with the length of the shelf) and time t, respectively, Y being a corresponding transverse outer coordinate and  $\Sigma_Y$  being lateral shear stress averaged over the inner domain  $\Sigma$ ;  $V_X$  was deliberately chosen notation in the inner model to denote the derivative of the locally averaged velocity V with respect to X.

With this formulation in mind,  $\Sigma = (1 - r)\rho_i g \bar{H}/2$  would indeed be the stress in an unbuttressed ice shelf, and we could apply a  $\tau^*$ -type dimensionless stress scale to crudely represent over- or underbuttressed ice shelves, as in the paper (solution of the outer problem would require the inner problem to be solved in at each mesh node of the outer problem, treating the inner domain  $\Omega$  like a Lagrangian variable. The imposition of a integral constraints does however sufficiently complicate the solution of the partial differential equations used in the paper that we stopped short of this stress-based formulation for the time being; also. The derivation of the outer problem above is also non-trivial, and beyond the scope of a single paper that also seeks to solve the inner problem that is the subject of the present paper.

That said, allow us one word of caution about assuming that imposed stress leads to catastrophic fracturing and imposed strain rates lead to smooth fracture propagation. The point in the review

is well made, but the complications introduced by having singificant overburden effects and a viscoelastic rheology actually means we sometimes see the opposite here. For instance, in a purely elastic setting with imposed stresses, calving by propagation of a basal crack occurs through continuous crack growth as the imposed stress is increased. (At risk of falling foul of the next comment, this is figure 8b of Zarrinderakht et al, 2022). In the controlled-strain-rate set up of the present paper, we get the intermittent crack propagation of figure 3 of the present paper.

**Reviewer comment:** Reference to past and current work. This is a pretty minor point, but I found I had to constantly refer back to the authors prior and current work and this made it a little bit more difficult to follow the manuscript. One example of this is that the the authors refer to the non-dimensionalization done in a previous manuscripts to justify a result \*before\* the non-dimensionalization is introduced in this manuscript. I think it would be helpful to streamline the references to previous/existing work in the introduction to tell summarize what has been done in those manuscripts and how this manuscript differs and then keep the cross-referencing to a minimum in the methods/results sections. The discussion would be an excellent time to come back to any cross-referencing between results.

**Response:** We've tried streamlining this as best as possible, but perhaps haven't made as many choices as the referee (or perhaps other future readers?) would like. In the model set-up, we have kept cross-reference to the other two papers in the sequence to help clarify the relationship between the models being used there for any reader who is already familiar with these other papers.

In terms of essential references that the reader actually may need to refer to to make sense of the model set-up, the only one that really springs to mind is the decomposition of stress into viscous and elastic parts during a fracutre propagation event (start of section 2.2 and equations (18)–(19)); equation (18) in particular from a part of Zarrinderakht et al (2022). We're hesitant to simply reproduce that derivation here. While not that long, it seems bad practice to repeat a basic derivation that already exists in the literature.

We have removed reference to the prior papers in the section on scaling, except (again, for a reader who is familiar with those) to point out that the parameter  $\kappa$  present in those papers is absent here. In terms of the result that we result we justify before non-dimensionalization, we assume that is the omission of the fracture toughness parameter from the model. That is not contingent on knowing the results of Zarrinderakht et al (2022), however, as we think that the argument (stress intensity factor should scale as stress times square root of length scale) really stands on its own. The reference to Zarrinderakht et al (2022) is really just intended to point out, for any reader familiar with that prior work, that one of the parameters in the prior models disappears in the limit being considered. To be fair, we make other scaling arguments, such as the argument about elastic versus viscous stretching during fracture propagation quoted under "horizontal boundary conditions" above, before scaling the model. It is true that a first principles derivation of all the approximations that have gone into the model using a consistent scaling is desirable. We plan to defer that to the paper in preparation mentioned above, which will also deal with the multiple scales expansion — the present manuscript is already rather long!

We have tried to remove all references to prior papers from the results section, deferring discussion of similarities and differences to the discussion, with the exception of section 3.5: the comparison with the elastic model forced by a simple stress prescription in that section seems to belong rightfully under "results" but cannot realistically be done without reference to the Zarrinderakht et al (2023) companion paper. The other exception is that we point out (para 4 of section 3.1) that the initial conditions for the problem are identical to those used in the companion paper (Zarrinderakht et al (2023)

**Reviewer comment:** Model domain size. This a bit of numerical issues, but typically for fracture

studies we take a domain length that is 10 to 20 times the ice thickness to avoid contamination from edge effects. This is a bit of rule of thumb. I think the authors used a much smaller domain in this case (perhaps out of numerical necessity). I would like to encourage the authors to demonstrate that their results are insensitive to domain size. This doesn't need to be included in the manuscript, but a single statement that their results were insensitive to model domain size would be helpful in convincing readers that results aren't sensitive to arbitrary domain choices.

**Response:** This is actually *not* a numerical issue. We are inherently assuming that there is no "nice" shallow shelf-like far field by employing a periodic domain (or nearly periodic, as described above in the context of the model being the inner model of a multiple scales expansion), which is where the consideration of edge effects would be the key consideration.

The results described here *are* inherently dependent on the domain size. This issue is actually one of the main foci of the submitted Zarrinderakht et al (2023) companion paper, where we discuss the purely elastic problem (with a simple rectangular domain shape) under periodic boundary conditions, and find that domain size has a persistent effect even for large domain width W. Since the elastic problem is embedded in the model here, domain size dependence is bound to persist (especially as the initial state of stress is identical to that in te Zarrinderakht et al (2023) companion paper).

The reason for that persistent dependence on W is likely the omission of buoyancy effects (or "flexure effects", depending on your point of view) from the elastic model. By this, we mean the effect of vertical displacements on the hydrostatic ocean pressure at the boundary: this is included in the viscous model, since fluid pressure is evaluated as a function of the current boundary position  $\mathbf{X}(\mu, t)$  in the viscous model. Based on the assumption of small elastic strains, they are self-consistently omitted in the elastic model: a small strain means a displacement that is small relative to ice thickness, and therefore a small effect on fluid pressure for hoirzontal length scales comparable with ice thickness. That ceases to hold true for larger horizontal length scales, when the ice behaves as an buoyantly-suppoerted elastic beam. This issue was already discussed at greater length in Zarrinderakht et al (2022), see the penultimate paragraph of section 2.1 as well as section 6.3 therein.

One of the important next steps in improving the numerical solver is to incorporate these flexue effects in the elastic component, but we have not been able to do that in the context of the present project (and PhD thesis). As a result, we have not attempted to study the dependence of results on W; an additional rationale for that is indeed that computations with larger W (especially once the initial W(0) becomes significantly stretched) are quite costly in terms of computational resources (the project having had no specially funded server time etc.).

This issue is described in some detail in paragraph 14 of the discussion (this text has not changed): There are however a number of improvements to our model that should be made before attempting to coarse-grain it. One obvious deficiency already discussed above is that we use a periodic domain of finite length W, and yet we know from the results in Zarrinderakht et al (2023) that fracture propagation is sensitive to the periodicity of the domain. As discussed there and in Zarrinderkaht et al (2022), the model for elastic stress  $\sigma_{ij}^{e}$  used here likely needs to be amended by taking account of buoyancy effects in equations (23) and (B1) to remove the dependence on domain width, by making widely spaced cracks less likely to propagate preferentially since vertical displacements between them can be larger (leading to bigger changes in fluid pressure) even at small strains. (For completeness, note that this is not an issue for the viscous component of the model, which correctly accounts for the buoyancy effect; the elastic model does not at leading order due to the assumption of small elastic surface displacements, which can however become large if the domain is taken wide enough as explained in Zarrinderakht et al (2022,2023). **Reviewer comment:** Finite element model resolution. Similarly, I would encourage the authors to conduct tests to demonstrate that their results are insensitive to the mesh size of the finite element Stokes solver. My understanding is that the authors did this for the boundary element size, but not the finite element sizes. I apologize if I misunderstood. Again, this doesn't necessarily need to take significant space in the manuscript, but explain any sensitivity to mesh size. Similarly, I'm not sure that the authors told us the number of elements or range of element sizes used for the FEM model domain. This would be helpful information.

**Response:** This is quite, correct, we did not describe the tests we conducted on the effect of finite element size on results. We added the following paragraph to the end of section 3.4: Figure 3.4 focuses on the effect of boundary element size, because of the coarser resolution used in the boundary element method near the crevasse tips compared with the finite element mesh. We also tested for the effect of finite element mesh resolution, by doubling and halving linear element sizes. Doing so was found to have no noticeable effect on results. In order to specify typical element sizes used in the Finite Element part of the model, we have amended the start of section 2.5, to say The viscous component of the model is solved using the Elmer/Ice finite element package (Gagliardini et al, 2013; Todd et al, 2018, Amundson et al, 2022). To handle the high-stress concentrations near the newly-introduced crack tips, we have used local mesh refinement: the smallest (linear) element size near a crevasse tip was typically 0.002 times the initial ice thickness.

**Reviewer comment:** The interplay between viscous and elastic stresses presented here is interesting. Given that one of the requirements for vertical crevasse propagation is that the viscous pre-stress must be tensile, is it possible to determine the vertical crevasse depth solely using the viscous model by assessing if the stress at the tip of crevasses is tensile? A relevant question is whether the elastic part of the problem is even needed to determine if (when?) crevasses will penetrate the entire thickness. It seems entirely plausible that the elastic part of the problem is entirely determined by the viscous pre-stress and it may be possible to ignore the elasticity and visco-elasticity completely. I think that is a result that the wider community would find very interesting. I was left wondering if the boundary element model was even necessary and if a simpler full Stokes model could suffice when paired with some remeshing and a J-integral or some other estimate of energy fluxes.

**Response:** Our original hunch (and hope!) was exactly that it would be possible to use only a viscous solver to model crevasse propagation, at least past the initial incision. We believe, however, that this would have to involve continuous growth of crevasses in the model (that is, each crack advancing by a very short distance on each time step) meaning that the elastic fracture propagation component of the model would have to just keep track of the evolving viscous pre-stress at the crack tip, extending the crack just far enough to keep it at the location where the visocus pre-stress changes from tensile to compressive.

For the n = 3 case studied in the paper (*n* being the Glen's law exponent), that does not appear to be what actually happens: even though individual fracture propagation events typically involve small increments in crevasse length (except for calving by surface crevasses), they do remain episodic and noticeably overshoot the location where viscous pre-stress becomes compressive again. (This was the reason for putting figures 4 and 7 in the paper). It may be worth pursuing this again in the future, to determine how well a purely viscous solver would do, with crevasse propagation determined by the viscous pre-stress.

There are theoretical reasons to suspect that the n = 1 case might be different — that is, it may well conform to the suspicions expressed in the reviewer comment. Specifically, there being no Dirichlet conditions on displacement or velocity, the total stress given a domain shape with the same Neumann conditions on stress should then be the same regardlless of whether we compute it from a Newtonian Stokes flow or a purely elastic model. This can be extended to show that the sum of viscous pre-stress  $\sigma_{ij}^{v}$  and elastic extre stress  $\sigma_{ij}^{e}$  computed as per the model in the paper should then be the same as the purely viscous stress computed at the termination of crack propagation, which suggests that viscous stress should be exactly on the cusp of permitting further crack propagation. We have not pursued this issue further, however, due to time constraints.

(For completeness, the idea of using a *J*-integral to compute stress intensity factors for a viscous flow model is an interesting one; for the  $n \neq 1$  case, the dependence of stress on distance  $r_{\rm tip}$  from the crack tip differs from the  $r_{\rm tip}^{-1/2}$ -type behaviour familiar from linear elasticity. Physically, the energy arguments behind the *J*-integral also work out differently, since there is no elastic stored energy to account for, but instead a dissipation rate that needs to be partitioned between internal energy ("heat") and other free energies; we have not tried to pursue this further for the time being, the reason being that for  $n \neq 1$ , a purely viscous model does not appear to suffice.)

**Reviewer comment:** Section 2.1 This section provides a detailed description of the contact conditions imposed and is fairly elaborate. These are fairly standard conditions, but as the authors found, these conditions can be numerically challenging to impose. Later in the manuscript we learn that the contact conditions are \*not\* actually imposed in the Stokes solver. I will be honest that I was little bit miffed to have gone through and followed all of the details only to find out that these conditions were eventually dismissed because of the numerical difficulty imposing them. Fair enough. But there isn't any detail about how these equations are treated numerically and what the authors end up solving for the viscous part of the problem is a standard Stokes flow problem. I think the manuscript can be simplified by ditching large portions of this section or relegating it to supplementary material. I would be a lot more interested in reading about it in a manuscript that also provided a plausible numerical method/discretization that was implemented.

**Response:** Yes and no. We *do* impose the contact constraints on the elastic solver, and when arguing that the elastic problem with pre-stress is the appropriate short-time-scale limit of the overall viscoelastic problem, it seems worthwhile demonstrating that the contact conditions are consistent with a first principles version based on dynamics rather than static deformations (it;s worth noting section 2.1 is not specific to the viscous problem). You admittedly have to go through appendix B to see that connection being made.1 As a consequence, we've left this as is — in the anticipation that we will fix the numerical in a future iteration of the problem. It seems worthwhile at least setting the stage for pointing out that this is a problem, which figures 4 and 7 are intended to do.

**Review comment:** Setting the critical stress intensity of zero (near Line 255). The argument around setting the critical stress intensity factor to zero isn't obvious to me. Or at least it requires more details. To start, the non-dimensionalization used here isn't introduced until later in the manuscript making it a harder to follow the argument.

As far as I understand it, the authors are introducing a scale for the stresses that is proportional to ice thickness and this results in a scale for the stress intensity factor that is proportional to ice thickness to the 3/2. But the rho\*g\*H term provides the magnitude of the hydrostatic pressure, which is compressive. In my experience, the argument for setting the critical stress intensity to zero relies on an argument centered on starter crack lengths.

For instance, the critical stress necessary to propagate a crack roughly scales like the critical stress intensity factor divided by the square root of the length of the starter crack size. For large starter crack sizes, this quantity becomes small. I would have thought that the authors require an additional

<sup>1This may seem trivial but for problems involving friction, a frequent assumption is that friction opposes displacements on the boundary in elastostatic problems, which neglects the possibility that the direction of friction is set by dynamic movements just prior to the body coming to rest. No such issues here, but it seems worth doing.

length scale here associated with starter crack or boundary element sizes. Are starter cracks assumed to be the same size irrespective of ice shelf thickness? Or are they assumed to scale with ice thickness?

Either way, small cracks are less likely to propagate and large cracks are more likely to propagate. The assumption here seems to be that large cracks are always present and thus the boundary element size is sufficiently large? I think it would be helpful to sketch out the argument in more detail. It actually seems like by setting the critical stress intensity of zero, the situation is analogous to a pre-existing vertical fracture that already exists with zero separation between the surface of the cracks. The question the authors ask is when will that pre-existing crack open.

**Response:** It is true that we use a scaling argument *before* we non-dimensionalize the model, but the non-dimensionalization in section 2.6 is really intended as a vehicle for reducing the size of the parameter space, rather than to argue for simplifications in the model. We have already appealed to various scaling arguments at this point, such as the time scale separation argument that underpins the decomposition of stress into viscous pre-stress and elastic additional stress in section 2.3.

The scaling argument here specifically says the following: assume you have an O(1) crack length, measured relative to ice thickness. Viscous stresses scale as  $(\rho_w - \rho_i)gH$ , and so therefore does the pre-stress on a crack (at least if the pre-stres is tensile, which generally requires some pressurization by water in the crack). If we have a crack that is comparable in length to H, then the stress intensity factor scales as  $(\rho_w - \rho_i)gH$ , which with typical values significantly exceeds fracture toughness  $K_{Ic}$ . Meaning, we can have a crack that is significantly shorter than H to act as a starter crack. We have reworded the existing text in section 2.4, 3rd paragraph onwards, as

With a finite fracture toughness  $K_{Ic}$ , a finite initial crack length is generally required for  $K_{I,stat}$  to exceed  $K_{Ic}$ , leading to the non-trivial question of how that initial crack can be generated (e.g. Krug et al, ,2014, Clayton et al 2022). Here we sidestep that issue by noting that the tpyical stresses due to viscous flow in an ice shelf scale as  $(\rho_w - \rho_i)gH$ , where H is a scale for ice thickness. We expect viscous pre-stress to scale the same way, at least where it is tensile. For a crack with appreciable length (meaning, a length that scales as H), the stress boundary conditions on the elastic problem then lead to a stress intensity factor that scales as s  $K_{I,stat} \sim (\rho_w - \rho_i)gH^{3/2} \approx 10$  MPa  $m^{3/2}$ , if we assume  $H \sim 500$  m,  $\rho_i = 917$  kg  $m^{-3}$ ,  $\rho_w = 1028$  kg  $m^{-3}$  and g = 9.81 m s-2, while estimated fracture toughness values for ice  $K_{Ic} \approx 0.1$  MPa  $m^{1/2}$  (Rist et al, 1996) are typically much smaller. In other words, if we have tensile pre-stresses, seed cracks much smaller than H suffice to lead to fracture propagation.

Motivated by this, we assume that even short initial crack will propagate if the pre-stress near the existing crevasse tip is tensile. Specifically, we assume  $K_{Ic}$  is small enough to be negligible, and replace equation (26) with

$$\dot{d} = \max\left(-\frac{K_{\mathrm{I,stat}}}{K_{\mathrm{Ic}}|K'(0)|}, 0\right).$$

Note that this is equivalent to treating the small fracture toughness parameter  $\kappa$  in Zarrinderakht et al (2022) as zero.

We do not think that this amounts to a pre-existing crack being opened: for a pre-existing crack, you would not allow a tensile stress *anywhere* along the crack, not just at the tip of the already existing crevasse, and you would typically assume either a friction law or free slip even while the crack is closed. None of these apply here. We should also point out that the limit of a vanishing fracture toughness is not that exotic, having been used previously by Tsai and Rice (2010, JGR) in a hydrofracture context. (In addition, although our model prescribes the crack path based on symmetry, that is not interchangeable with the  $\kappa = 0$  limit; even for a vanishingly small fracture toughness, it would still be possible in principle to predict crack paths based on a maximum hoop

stress criterion,)

The question about how long a starter crack needs to be to recommence fracture propagation is entirely legitimate, and all we have done so far as argue that the answer is "much smaller than H". We can do better than that, but it gets complicated. When the first crack incises the ice surface, the question is relatively easy to deal with, and a version of it is in appendix C1 of Zarrinerakht et al (2022). There appears to be a typo at the bottom of page 4510 of that paper; the criterion for a basal crack is that the ratio of starter crack length to ice thickness must be  $\gtrsim \kappa^2/(\tau - 1 + r^{-1})^2$ , and  $\gtrsim \kappa^2/\tau^2$ .  $\kappa$  is a scaled fracture toughness as defined in Zarrinderakht et al (2022),  $K_{Ic}/(\rho_i g H)$ , while  $\tau$  is basically the dimensionless  $\tau^*$  in the present paper, scaling as  $1 - \rho_i/\rho_w = 1 - r$  for an unbuttressed or moderately buttressed ice shelf.2 Translate that to the real world and starter cracks need to be of a length that is comparable to  $K_{1c}^2/[(\rho_w - \rho_i)gH^{3/2}]^2$  times ice thickness, which is why the comparison of  $K_{1c}$  with  $(\rho_w - \rho_i)gH^{3/2}$  that we make in the paper (see italicized text above) seems relevant.

This gets a fair bit more complicated if you have a crevasse already, and that crevasse is still sharply incised near its tip when you add a new small starter crack. The Stokes flow near the sharp crevasse tip is locally self-similar, with viscous pre-stress behaving as  $r_{\rm tip}^{-1/(n+1)}$  for a general Glen's law exponent *n*. A natural scale for the viscous pre-stress along a new starter crack of length *d* extending from the pre-existing crevasse tip is then  $\sigma^{\rm v} \sim (\rho_{\rm w} - \rho_{\rm i})gH \times (d/H)^{-1/(n+1)}$ . Since this is the pre-stress acting on the new starter crack, the elastic stress scales the same way, yielding a stress intensity factor that behaves as

$$K_1 \sim \sigma^{\rm v} d^{1/2} \sim (\rho_{\rm w} - \rho_{\rm i}) g H^{3/2} (d/H)^{\frac{1}{2} - \frac{1}{n+1}}$$

In order for  $K_{1c}$  to be comparable to  $K_{1c}$ , starter cracks need to have a length-to-ice-thickness ratio comparable to

$$\frac{d}{H} = \left(\frac{K_{\rm 1c}}{(\rho_{\rm w} - \rho_{\rm i})gH^{3/2}}\right)^{(n+1)/(n-1)}$$

Again, the ratio  $K_{1c}/[(\rho_w - \rho_i)gH^{3/2}]$  that we estimate numerically in the paper features critically, but now to a more exotic exponent. The n = 1 case is singular, and apparently any starter crack length will do so long as  $(\rho_w - \rho_i)gH^{3/2}$  exceeds fracture toughness  $K_{1c}$  (this goes back to our response regarding the comment whether a purely viscous model might suffice). For the more commonly accepted n = 3, we are back to an exponent (n+1)/(n-1) = 2, the same as is obtained when considering incising a starter crack into a previously unblemished, flat surface as considered in the previous paragraph.

The argument regarding the viscous pre-stress around a sharply incised crevasse tip in the previous paragraph assumes that the stress singularity scales as  $\sigma^{v} \sim (\rho_{w} - \rho_{i})gH \times (d/H)^{-1/(n+1)}$ , which is appropriate if the viscous pre-stress is immediately tensile following the end of a crack propagation event. In our numerical solutions, this occurs only during the first crack propagation episode (e.g. figure 6 in the paper under review). If crack reactivation is delayed because the pre-stress is initially compressive, then the magintude of the pre-stress is initially small once it becomes tensile again, changing the scaling above. In the meantime, viscous flow also blunts the crack tip, and estimating starter length crack gets even more complicated.

We recognize that there are more formal scaling arguments and asymptotic analyses to be presented in this context, as is the case for the multiple scales expansion involved in setting the lateral

<sup>2These results are consistent with the size of the thin regions of steady states near the horizontal axis in e.g. figures 6a–c and figure 8b of Zarrinderakht et al (2022), and should be consistent with the early work by Weertman (1980, *J. Glacio.*) on cracks incised into half-spaces

boundary conditions above. You're probably tired at this point of hearing that we intend to that elsewhere, but the paper as it stands already seems quite long, and we're mindful of the effect of overloading it.

**Reviewer comment:** Line 15: It is true that full-depth propagation is a key requirement for calving. However, even when these fractures are present, fractures need to also propagate horizontally in most cases.

**Response:** It seems likely that line 15 is the wrong place (in the original manuscript), as there seems to be no reference to full-depth propagation there. Hopefully the updated text in the penultimate paragraph of section 1 (see the response to the first reviewer comment above) addresses this adequately.

**Reviewer comment:** Lines 30-35: There is a big chunk of literature missing here from methods associated with damage mechanics (and the associated phase field methods). Damage mechanics can be used to simulate the interaction between multiple crevasses in a continuum model. I see that the authors come back to this in the discussion. I think it might be useful to add damage mechanics and phase field methods into the introduction as another method to contrast with the boundary element. The reality is that damage mechanics has become extremely popular in the fracture community because it avoids or sidesteps many of the issues that they authors have discovered with remeshing and mesh tangling.

**Response:** We have changed the introduction quite a bit. The natural place for an description of this seemed to be immediately after the discussion of Maxwell rheologies. A new fourth paragraph in section 1 states

Viscoelastic phase field models (e.g., Clayton et al 2022, Sondershaus et al, 2023) allow previous work on linear elastic fracture mechanics to be extended to viscoelastic media, while also providing a more sophisticated approach to explaining how cracks are initiated, and automatically handling complex crack geometries that can be more challenging for traditional linear elastic fracture mechanics (e.g., Zehnder 2010, Gordeliy et al 2019). Current formulations of such phase plane models used in glaciology however appear to be limited at least formally to small strains, and therefore unlikely to be able to describe the interplay between fracture propagation and longer-term, large-strain viscous flow. Viscous damage mechanics models (e.g., Pralong et al 2005, Duddu et al 2013, Duddu et al 2020) do not suffer from this limitation, at the cost of a less clear relationship between evolving damage and the usual energetics of crack formation that phase field models encode succinctly.

**Reviewer comment:** Lines 30: It has been demonstrated that for sufficiently small particles, discrete element models do converge to elastic models (without fractures) and can reproduce key features of wing-cracks and other continuum fracture observations given an appropriate parameterization of the failure strength. There is non-uniqueness inherent in discrete element models description of fracture. But this non-uniqueness is related to failure under mixed-mode loading, which is something that also requires additional assumptions to incorporate into LEFM. To me, the bigger issue with discrete element models it that they are fully dynamic and typically limited to time scales of minutes to hours. These models cannot be easily integrated into larger scale and longer time scale simulations.

**Response:** Most of the discussion of discrete element models has now gone. In the fourth paragraph, we reference the stiffness of visocelastic discrete element models (which presumably explain why Åström et al use viscosity parameters that are not close to those for ice, to limit the ratio of time scales:

Taking a conservative estimate of extensional stress of  $2 \times 10^5$  in an ice shelf with Glen's law parameters appropriate for ice at -10° C (Cuffey and Paterson, 2010), we find a viscosity around  $10^{13}$  Pa s. With Young's modulus of  $10^9$  Pa, the Maxwell time is around 7 hours: long compared with the

time scale for crack propagation (Olinger et al 2022, 2024), but short compared with the time scale over which an ice shelf flows and the forcing on any cracks in the ice therefore changes. This separation of time scales suggests that simplifications should be possible to a full large-strain viscoelastic model of ice flow and fracture, permitting a separate representation of short time-scale elastic and long time-scale viscous flow effects (which, if not separated, are also likely to make a model that incorporates both numerically stiff by construction, as is the case also with discrete element models that permit viscous flow, see e.g. (Åström et al, (2013)). This is the basis of the model we state and solve in this paper: "leading order" in the title of this paper refers to our model being valid at leading order in the ratio of crack propagation time scale to Maxwell time, and in the ratio of Maxwell time to vioscus deformation time scale. We also return to this at the end of the paper, in the last paragraph before the appendices, In closing, we note that a similar procedure to ours is used in coupled viscous flow-discrete element simulations such as Crawford et al (2021), which use an ice geometry predicted by a viscous flow solver to then compute rapid fracture propagation using a brittle-elastic or viscoelastic discrete element model (Åström et al. 2013). The analogy is not perfect, as the viscous pre-stress is not imposed on the discrete element model, despite the key role played by that pre-stress in our model. The discrete element model conversely has the advantage of automatically being able to predict arbitrary crack geometries, albeit at far more computational effort than our boundary element approach, but the complicated geometry that results is not used directly to restart the viscous flow solver, while our model does exactly that. Future work on the modelling approach used here will need to bridge the gap with discrete element models by allowing for a much larger number of arbitrarily located and oriented cracks.

**Reviewer comment:** Title: Leading order in what? Is it leading order in the Maxwell viscoelastic time scale? Or the fracture propagation time scale? I don't think this is ever stated. One potential subtlety is that the time scale of fracture propagation has to be short enough to be fully elastic, but also slow enough to avoid dynamic effects.

**Response:** See the first text excerpt in the response to the previous point immediately above. The end of the rewritten fourth paragaph of the introduction explicitly states what "leadaing order" we are interested in.

**Reviewer comment** Line 85, just as a historical point the field of dislocation mechanics has long recognized the non-uniqueness associated with displacement and the compatibility equations in the presence of dislocations.

**Response:** We're struggling a little bit with the line number references; in our copy (recently downloaded from the copernicus website), line 85 (going by the numbers in the left-hand margin) is talks about the introduction of new sections of surface not satisfying a kinematic boundary condition. It's not clear to us whether that is what the comment references.

Revier comment: Line 151: missing word?? "Occurs over a time scale??""

**Response:** Is this really problematic? If we wrote "significant visocus deformation only occurs over a time scale much longer than the Maxwell time", would that be incorrect / unclear?

**Reviewer comment:** Line 216, equation 21. I think I'm confused about the boundary conditions used. I thought that a traction boundary condition was imposed at the edges of the domain, which seems to imply that the gradient in displacements normal to the boundary must vanish and not necessarily the displacement. I think this comes down to whether the authors are considering a displacement controlled or stress controlled boundary condition. The natural choice for this boundary conditions seems to be a traction/stress boundary condition, but I think (?) the authors are using a velocity boundary condition.

**Response:** Hopefully the response to fourth main reviewer comment ("Horizontal boundary conditions") and the associated changes to the text cover this.

**Reviewer comment:** Section 2.4. Why restrict crack growth to the vertical direction. These cracks will generally experience mixed-mode loading and, for sufficiently slow propagation, it is commonly assumed that cracks will propagate in the direction of maximum tensile stress. This should result in a slight curvature of the crack surface. Is this negligible enough to be neglected? Or this just done for convenience?

**Response:** There was a typo in the original text specifying one of the two initial crack locations. The point is that, in a periodic domain, the cracks have been specified to be at locations that ensure that the stress field  $\sigma_{xx}$  remains left-right symmetric and the shear stress  $\sigma_{xz}$  vanishes by symmetry, meaning that the cracks are loaded in a purely mode-I configuration (take figure 2 and extend the results you see periodically to the left and right, and the requisite symmetry should start to be obvious). The set-up also ensures that advection will maintain that symmetry. That contrived choice was made for convenience, as the following updated first paragraph of section 2.4 is hopefully sufficiently honest about.

A description of how cracks actually propagate in response to the stress field around the crack tip is still missing. The formulation up to this point has been quite general. In what follows, we will consider only cracks that grow vertically in two predefined locations. One is at  $x = x_s(t)$  for cracks extending from the upper surface  $\partial\Omega_s$ , and the other at  $x = x_b(t)$  from the lower surface  $\partial\Omega_b$ . We choose an initially rectangluar domain shape between  $x_1 = -W(0)/2$  and  $x_1 = W(0)/2$ , and assume that the crack locations start at  $x_b(0) = -W(0)/4$  and  $x_s(0) = W/4$ . The same geometry is also considered (with a uniform shift in  $x_1$ ) in Zarrinderakht et al (2023). The initial domain geometry is symmetrical about these initial locations when extended periodically, and cracks are subsequently advected at the local ice flow velocity as in Figure 5a. It is then easy to show from the quasi-periodic boundary conditions (9) and (10) that the stress field remains symmetrical about the crack locations and purely mode I vertical crack propagation is self-consistent, if contrived. By not considering arbitrary crack locations and geometries, we are able to take a simpler computational approach that leads to significant qualitative insight. We anticipate that future work will deal with much more general crack geometries.

**Reviewer comment:** Line 235: What is used to initialize cracks here and how is it different from the other submitted study? I got lost in this sentence. I think this might just be a matter of rewriting the clauses in the sentence or separating into two sentences.

**Response:** The confusion is more likely to be the result of a typo that the original text contained here, which mispositioned the basal crack. We have changed the text (and split a couple of sentences). The new text is in the italicized except from the revised paper in the last response immediatly above.

**Reviewer comment:** Lines 302. In Berg and Bassis (2020), we actually found that the sea spring resulted in a time step size dependent solution. This was generally small for most situations, but where it become apparent was when we remeshed or the domain abruptly changed size. I suspect that those effects are more muted here because of the periodic boundary conditions and the relatively flat domain. The sea spring had the most artifacts when we included the horizontal ocean pressure condition at sloped calving calving front. I think doing convergence studies associated with the FEM element size should demonstrate that this isn't an issue here.

**Response:** As indicated in the response to the seventh main reviewer comment ("Finite model resolution"), we have experimented with finite element mesh size (which induces a change in time step size through the CFL criterion) and found no substantial changes to our results.

In our understanding, the time-step-size-dependent results in Berg and Bassis (2020) come about from the removal of an upper portion of the shelf. This leads to a situation where the force balance condition equivalent to equation (C2) in our paper is not satisfied instantaneously, given the current domain shape (and "domain location" relative to sea level, if you will). In that case, the sea spring really compensates for a non-negligible inertial term. The violation of simple force balance (sans sea spring) occurs in Bassis and Berg (2020) because the same force balance condition is satisfied immediately prior to removal of part of the domain, and the left-hand edge of the domain there experiences vanishing shear traction. The removal of part of the shelf then does not change the veritcal component of surface traction, while the integated body force changes abruptly.

In our model, that cannot happen. The relevant integral  $\int_{\Omega} \rho_i g d\Omega$  in (C2) does not change abruptly as we never remove part of the domain (all calculations stop if a crack propagates fully through the ice), and small changes in domain elevation by adjusting vertical velocity are enough to maintain force balance with minimal "sea spring" forces.

**Reviewer comment:** Line 305. Contact conditions. There is a lot of text on contact conditions and imposing the right boundary conditions. But, as far as I can tell, these are largely abandoned and instead a standard Stokes flow problem is solved. This results in mesh tangling, which is a long standing problem in large deformation problems and one of the reasons that damage mechanics has become so much more popular than LEFM for many problems. Can the authors assess if there is any impact on the solutions? The mesh tangling can have subtle effects on the solution because of the unphysical nature of even small node crossing.

**Response:** No doubt this does have an effect while the crevasse walls are in contact. The mesh overlap remains small, but stresses will be computed incorrectly while there is mesh overlap: the stress conditions for contact indicate that compressive normal stress is generally larger (and definitely never less than) the fluid pressure while crevasse walls are in contact, while omitting the contact conditions sets compressive normal stress to that value. This almost certainly affects the computed viscous pre-stress near the crevasse tip, but it seems unlikely that a solution with viscous contact conditions would predict a tensile pre-stress and hence crack reactivation while the walls of the crack are still in contact. It seems likely therefore that crack reactivation with and without the contact conditions, to the entire crevasse wall being subject to a Neumann stress condition, but it is unclear that this occurs at the same point in time. In particular, without the contact conditions, the mesh can still be entangled when pre-stress becomes tensile again: a local analysis of velocity and stress near the crevasse tip would simply say that the velocity must be such that the crevasse walls are moving such as to open the crevasse at that time, but they can still be entangled. As a result, it is difficult to asses in more detail how different results would be with the contact conditions without actually having implemented them.

We're happy to acknowledge that damage mechanics has an advantage here, but we'd note that phase field models (being the closest to replicating LEFM) don't seem to deal with finite visocus strains, and for them at least, the notion of mesh entanglement may as yet not be relevant (since small strains really say that the mesh ought to remains fixed).

**Reviewer comment:** Line 355. The s is defined here based on Archimedes principle using the mean ice thickness, right?

**Response:** Yes. This material has now moved to section 2.1. Just after equation (7), the updated text now says

$$p_{\rm f} = p_{\rm f,s}(z,t) = \max(\rho_{\rm w}g(\bar{s}(t) - h_{\rm w} - z), 0).$$

Equation (C2) of appendix C allows us to relate  $\bar{s}(t)$  to mean ice thickness  $\bar{H}(t)$  across the domain through a simple flotation criterion as  $\bar{s} = (1 - \rho_i / \rho_w) \bar{H}$ .

**Reviewer comment:** Line 359. When portions of the lower boundary are above sea level, this means that crevasse propagate above the water line? Is there any other situations when this occurs? Can this be explained in a simpler fashion?

**Response:** Yes. In general (at least for the initial conditions we are dealing with), a part of the lower boundary above sea level will only result from basal crevasse propagation. That sounds a bit exotic, but is a necessary part of complete basal crevasse propagation. That said, this really has no effect in practice on vertical force balance, because it typically involves a very narrow, almost vertically-sided fracture, and the vertical component of traction on that part of the boundary is nearly nil. The text in question has been removed as part of the reorganization of the description of the boundary conditions, with the nearest equivalent now being the start of the second paragraph of appendix C,

Equation (C2) can be used to find the mean surface elevation  $\bar{s}$  used in equation (7), provided the surface force  $\int_{\partial\Omega_s} p_{\rm fs} n_2 d\Gamma$  is negligible and no part of the basal surface  $\partial\Omega_b$  is above sea level (both of which are typically the case in the solutions presented here, at least until the final crack propagation event).

**Reviewer comment:** Line 410. This might be better described using an illustration. I think this is consistent with the way most define the normalized crack length, but I got lost in the technical description. Maybe this should be moved to the methods section.

**Response:** We have moved the formal definition of cumulative crack length to appendix D, as it felt out of place in both "methods" and "results", and attempted to illustrate this with a new figure in that appendix. We are confident that our definition corresponds to what would be the only sensible definition of cumulative crack length (that is, summing increments in crack lengths whenever a crack lengthens) in the case of vanishing viscous deformation, in which case the length of the remaining neck of ice prior to a crack propagation epsiode,  $H_{\rm b,s}^-$ , is simply equal to the initial ice thickness (unity) minus the total length of cracks that have formed, so  $H_{\rm b,s} = (1 - d_{\rm b,s}^{\rm tot})$ . The moralization factor applied to newly formed cracks is therefore unity. Whether there is a standard definiton for cumulative crack length when changes in domain geometry happen due to episodic crack propagation separated by longer spells of viscous deformation is unclear to us.

**Reviewer comment:** Figure 2. I suspect that these dimensionless numbers will be meaningless for most readers without a strong feeling. Can they be put into a more intuitive scaling or provide physical numbers for a reference ice thickness? I'm having a hard time understanding why the edges of the boundary are not straight given that I thought that a constant velocity was imposed along the edges? Panels a and b didn't initially look like the same shape to me, but I see that different scales are used. Can the authors use the same horizontal scale for both sub panels to help readers translate between the two. Also, can the authors use a different color in panel to illustrate which of these is being shown in panel b?

**Response:** We hope that the updated section 2 as described in the response to the second main reviewer comment ("Non-dimensional numbers") will help with a better intuition. In addition, we have changed the start of the figure caption to the following

Domain evolution for  $\tau^* = 0.06$ ,  $\eta^* = 0.1$ , with an initial domain width W(0) = 2;  $\tau^* = 0.06$  is slightly larger than the value of 0.055 for an unbuttressed shelf, while  $\eta^* = 0.1$  corresponds to a surface water level close to sea level.

We have also fixed the aspect ratio / scaling issue for the plots

**Reviewer comment:** Another minor question that I is whether the episodic propagation nature is inherent to the problem or has that been built in. The authors have assumed that fracture propagation is rapid compared to the viscous time scale and slow and steady propagation would be inconsistent with the approximation. Would omitted terms in the viscoelastic relationship result in more continuous propagation? At the vary least, it is encouraging to see that the results are self-consistent with the approximation.

Response: The fracture propagation time scale should be set either by inertial effects (as is assumed here) or by fluid flow in the fracture (which is actually more likely, and the subject of the cited Olinger et al papers). Assuming as we do that the propagation time scale is short compared with the Maxwell time, then a fracture for which the static stress intensity factor increases at least initially with crack length will propagate with the stress increment above the viscous pre-stress being purely elastic. None of the omitted terms in the reduction to pure elasticity are singular perturbations, even in a non-linear Maxwell model (incorporating the advection and rotation terms are in the appropriate material derivative of stress, over which there is still some measure of debate), so the prediction of elsatic fracture propagation should be robust so long as the pre-stress is computed correctly. (And that pre-stress ultimately dictates, along with domain geometry, whether the elastostatic stress intensity factor increases with crack length).

The point where you might ask whether the model for the pre-stress is correct is in the assumption that "nothing happens at time scales comparable to the Maxwell time", since we do not compute the stress relaxation to a purely vioscus stress that occurs post-fracture-propagation; the time scale for that relaxation is obviously the Maxwell time itself.

It is conceivable that a crack could start propagating again as a result of a partial relaxation of stress, but it is hard to imagine that this would lead to more continuous long-term crevasse growth, as opposed to the episodic behaviour we see. Except during the initial crack propagation episode, we see only single crack propagation events. The effective stress immediately after each event is not tensile near a crack tip, and neither is the purely viscous stress that results after full stress relaxation. In order for partial stress relaxation to result in crack reactivation, that relaxation would somehow have to lead to stress becoming temporarily tensile again near the crack tip. We have not been able to show mathematically that this cannot happen (nor particularly tried yet), but it seems unlikely. As the comment states, our results at least have the advantage of being self-consistent, though they no doubt leave open plenty of avenues for future research!

The relevant portion of the text here (unchanged) would be the third paragraph of the Discussion, It is important to state that the model described in the present paper actually does not fully describe the effect of viscoelastic relaxation since it computes a fully viscous stress field immediately after crack propagation has run to completion, rather than resolving stress relaxation over the Maxwell time scale. Given that we find crack reactivation once the stress has fully relaxed to its viscous form, it is likely that crack reactivation actually occurs sooner, after only partial relaxation of the stress field. It is unclear whether this has a significant effect on the total length of the fracture formed in a single fracture propagation episode (by which we mean a series of successive fracture propagation events between which the pre-stress relaxes viscoelastically, but the domain remains unchanged by viscous deformation): it is conceivable that crack reactivation after partial stress relaxation could lead to the crack propagating less far after each reactivation, but that there could also be more such propagation events in the propagation episode, leading to a comparable final crack length.

**Reviewer comment:** Is it possible to benchmark the model against any of the fully viscoelastic models that have been used for bottom or surface crevasses? This might be interesting and build confidence in the approach used.

**Response:** Maybe. As we point out in the introduction, the viscoelastic models that aim to describe the same fracture physics as is encoded in LEFM owuld be phase field models. The examples we are familiar with however rely on small strains (and therefore, as far as we understand) do not allow for a finite deformation of the domain. So the scope for deformation would be limited to a single fracture propagation episode. (In fact, our suspicion is that the results in Clayton et al (2021) reflect purely elastic behaviour, as the solution runs appear to be quite short due to the need ot resolve inertial effects, certainly shorter than a reasonable estimate of Maxwell time.)

**Reviewer comment:** Line 441. The overshoot is pretty typically of the elastic problem where the

stress concentration allows fractures to penetrate slightly deeper than it would otherwise. Actually, you get the exact same overshoot if you use a linear viscous rheology instead of the power-law viscous.

**Response:** That is interesting as our intuition regarding a Newtonian viscous rheology was different see response to the 8th reviewer comment, "The interplay between viscous and elastic stresses"). References would be useful for us to dig deeper into this; we'd be particularly curious about the extent to which depends on boundary conditions.

**Reviewer comment:** How is the longitudinal extension boundary condition determined? I think I get the impression that  $R_{xx}$  decreases as the ice thickness decreases? Is that true?

**Response:** We're not certain which place in the text this refers to. In general, see our response to the fourth reviwer comment ("Horizontal boundary conditions")

**Reviewer comment:** Line 465: Is there a physical explanation that can be provided? My suspicion is that this is all surface water pressure mediated and then once the water pressure reaches a critical value, the surface crevasse propagates the entire depth, but I think readers would like a little bit more physical description of what is happening this situation to balance the technical description.

**Response:** This is undoubtedly driven by surface water pressure. We have changed the text at the end of section 3.2 to say

This event mirrors similar behaviour in the purely elastic model for surface fracture propagation in (e.g., Zarrinderakht et al, 2002, figure 2): the surface crevasse tip immediately prior to full-depth fracture propagation is below the surface water level, and the crack goes through runaway growth fueled by the faster growth in water pressure with depth, compared with cryostatic overburden.

**Reviewer comment:** Figure 7. There are two blue lines. Which is the water level?

**Response:** The original caption failed to point out that the dark blue line is sea level. The revised version says

he horizontal light blue line at z = 0.025 indicates the surface water level, the dark blue line at z = 0 is sea level.

**Reviewer comment:** Line 480: \*boundary\* element size, right? This isn't done with respect to finite element size, right?

**Response:** Yes. We've changed the text to say ... are robust under refinement in boundary element size.

**Reviewer comment:** Figure 9: Can authors use the same axis for all of the subpanels? Also, can you use a different color than red and green for the lines as this combination can be difficult for color blind folks to distinguish.

**Response:** We're happy to change the colour green. Using the same axis (presumably vertical?) is a bit trickier — you lose a lot of detail in the cases of full propagation of the basal crevasse, where a plot ranging from 0 to 1 on the vertical axis compresses the data a great deal towards the bottom of the plot. We have now advised the reader of the different vertical axis scalings in the caption,

Note panels (c) and (d) scale the vertical axis differently from (a) and (b) in order to avoid excessive white space.

**Reviewer comment:** What does it physically mean when the surface water table is at or beneath sea level? Also, what condition would correspond to water free surface crevasses (the most reasonable condition for most ice tongues in the Antarctic)?

**Response:** It likely means that you have to have impermeable ice at the bottom of the shelf (and its sides, if you're in 3D), sealing the shelf from sea water intrusion. The top presumably still needs to be somewhat porous to support an aquifer with a water table height. That porous firm (?) would have to extend below sea level at the site of interest to support an aquifer at that depth. Contrived, but easy to dial into the model.

Dry crevasses:  $\eta^* = 1$  is the easiest way to ensure that, see the discussion in the updated section 2.1 referenced under the second reviewer comment respose above, "non-dimensional numbers"

**Reviewr comment:** Line 640-650. I think the interpretation of damage mechanics is not entirely accurate. Damage mechanics is a \*method\* that can be used to simulate failure in a continuum model that alleviates the need to remesh necessary in fracture mechanics approaches. The damage production function can be chosen to reproduce results from LEFM, micro-mechanics or be chosen based on a heuristic calibrations.

**Response:** We rather ill-advisedly conflated the depth-integrated damage mechanics models that we actually wanted to reference here with damage mechanics at large. The latter is now discussed in the introduction, see the response to the 12the reviewer comment ("Lines 30–35. There is a big chunk of literature missing here"). We have tried to make clear that here, we are realy only thinking about the specific large scale damage models developed by Borstad et al and Huth et al, for which there presumably can be no question of directly representing LEFM (and which follow the standard "heuristic" recipe for damage mechanics, although ina depth-integrated model:

A better analogy exists between the physics in our model and large-scale, depth-integrated damage mechanics models for ice shelves (e.g. Borstad et al 2013, Huth et al 2021, 2023), albeit a loose one. A damage mechanics model of that type evolves a damage variable through a combination of advection and damage production or healing, and predicts the formation of a through-cutting crevasse when a critical level of damage has accumulated. In our model, we instead evolve the local surface geometry of a piece of ice that is advected with the larger-scale flow of the ice. We can think of that surface geometry as encoding a form of dmaage (see also Bassis and Ma, 2015), and calving occurs when the combination of stress, surface water level, and domain geometry reaches some critical state. The key is that, like the damage variable of a damage mechanics model, the surface geometry evolves over time (as an analogue to damage), and calving does not happen instantly when a critical stress is reached.

An important observation is that the time scale of surface geometry evolution between fracture propagation events is intrinsically the same as the advection time scale for the flow of ice in the ice shelf: unlike a damage mechanics model, where the time scale of damage evolution is given by an independent model parameter ("B\*" in Huth et al (2021, 2023)), the evolution of surface and crevasse geometry due to viscous flow in our model occurs on the same time scale over which the ice thins appreciably (or equally, on which O(1) strains develop in the ice). That time scale is also the same as the time scale on which a piece of ice travels the length of the ice shelf.

**Reviewer comment:** Line 725. A big difference between the work of Crawford et al., and this study is that the DEM in Crawford et al., is initialized with no memory of its previous state. Hence, the model has no memory of basal/surface crevasses that had propagated over part of the domain when re-initialized.

**Response:** It had been our intention to point that out when we said

... but the complicated geometry that results is not used directly to restart the viscous flow solver, while our model does exactly that.

---

## Author Comment (AC2)

**Response to RC2: 'Comments to "A leading-order viscoelastic model for crevasse propagation and calving in ice shelves" by Zarrinderakht et al.**

**February 26, 2024**

**Reviewer Comment:** In this paper, the authors coupled a boundary element method with the viscous ice-flow model, in order to combine the cracks propagation process with the viscous ice-flow model. The authors improved previous elastic models by using the real geometry at the time of crack propagation in their calculation. This work is potentially valuable to the cryosphere community, where the fracture and calving models are poorly developed.

However, I find the manuscript is hard to follow. This is partly because it's heavily citing other papers, some hasn't been published (Zarrinderakht et al., submitted), and some are not well known in glaciology. Furthermore, some of the key reference, which is used to describe the numerical solution, is wrongly cited. I hope the authors could improve the writing by being accu- rate, and bearing in mind that fracture mechanics is not widely implemented in ice-flow models, and some concepts are not well known (not as good as Stokes equations, for example). For example, when introducing equations, not only cite the original publication but also put the essential equations in the paper; also describe the physical meaning of the variables and equations in more details. I suggest a major revision to this manuscript. I hope the authors can put some effort in the writting style. There are some specific examples in the following comments.

**Response:** In our defense, we checked the model definition in sections 2.1–2.4, and could not find any undefined quantities. There is one key spot in which we rely on citation to prior work (Zarrinderakht et al, 2022), namely when we decompose total Cauchy stress into a viscous pre-stress and an elastic additional stress in equation (18). We decided it was not a good use of journal space to rederive that explicitly, since the place to find it is clearly identified (appendix A of Zarrinderakht et al, 2022). We made some changes to section 2.4, for instance to front-load the definition of the static stress intensity factor, which hopefully helps smooth the crack propagation description. We would note, however, that linear elastic fracture mechanics is not that poorly known in glaciology, there being eight papers in the pertinent literature cited in the introduction, and it is likely there are others.

As per the other referee's suggestion, we have also compacted the specification of the lateral stretching rate (previously in section 2.6) with the description of the lateral boundary conditions in section 2.1. That hopefully also makes the text more readable.

**Reviewer comment:** Abstract: The authors mention they solved the fracture mechanics problem on the actual domain geometry. I think here actual domain geometry here doesn't mean real glacier/ice shelf, but solve the cracks boundary. This is slightly misleading. Nevertheless, can we use observational datasets to validate the model?

**Response:** The actual domain geometry is the domain geometry predicted by the viscous flow

solver, the point being that other attempts to solve similar coupled viscous-elastic models have typically stopped short of solving the linear elastic fracture mechanics component properly (in the sense that they have typically used a simplified, rectangular proxy domain). This is expanded upon in section 3.5.

Can the theory be tested with actual data? In principle, of yes. At face value, creating the relevant data set looks like a generational commitment: for the cases where calving does involve the interplay between viscous flow and episodic fracture propagation as described in section 3.4, typical times to full fracture penetration from an initially intact ice shelf typically take a few decades, if we assume an initial ice thickness around 500 m and ice shelf flow that is not strongly buttressed. Ideally, you would want to scan the geometry of a particular piece of ice regularly over that length of time as it transits an ice shelf, while also monitoring average strain rates. The technology undoubtedly exists (geometry can presumably be scanned from UAVs under the ice, and by optical satellite for the surface, though seismic measurements are likely necessary to determine the location of crevasse tips). It's less likely that the funding would be available.

How much can be done using existing satellite data (especially given that basal crevasses are a key part of the picture) is less clear — and unfortunately lies outside our area of expertise. We would note however that we have not seen many process-scale remote sensing papers on calving, presumably because much of the relevant crack propagation is hidden from view. (There is a relatively recent piece by Joughin et al on calving at Sermeq Kujalleq that is an outlier in this regard, but the observations there pertain to grounded ice, so the model would need to be rewritten).

**Reviewer comment:** The key novelty of this work is the implementation of the boundary element method. A general description of boundary element method and why it's a good solution for the crack propagation problem (advantage) should be necessary?

**Response:** We're not quite sure what "advantage" refers to — the existing glaciological literature contains few if any examples of linear elastic fracture mechanics problems being solved numerically, the dominant method apparently being the use of canned kernel functions from Tada et al's (2000) *The Stress Analysis of Cracks Handbook*, while "advantage" suggests comparison with other numerical techniques. The point is that Tada's handbook doesn't have a recipe for arbitrary geometries.

As for describing the method, we have already done so in Zarrinderakht et al (2020), also published in *The Cryosphere*. It would seem a poor use of journal space to repeat that description. We have updated the text in the third paragraph of section 2.5, to reference that description, and point out the reasons for using a boundary element method (these being the ability to deal with arbitrary domain shapes and a small number of degrees of freedom — the BEM code runs rapidly on a single processor, which cannot be said for the FEM code for the Stokes flow problem)

To do so, we solve the linear elastic fracture dynamics component of the model using the boundary element method described in Zarrinderakht et al (2022, appendix B). This method is well-suited to computing stress intensity factors for cracks in domains of arbitrary geometry while using only a small number of numerical degrees of freedom, and is easily adapted to changing crack geometries (since only the domain boundary needs to be discretized, avoiding the need for remeshing in two dimensions) (see also Crouch and Starfield, 1983).

**Reviewer comment:** L31: Unit of extensional stress is missing

**Response:** Indeed. Our apologies. We have added "Pa" to the numerical figure.

**Reviewer comment:** L110-L114: Give the physical description of equations (5a).

**Response:** We have expanded the relevant passage to say the following

$$either \quad -[v_i n_i]^+ > 0 \quad and \quad -\sigma_{ij} n_i n_j = p_{\rm f}, \tag{5a}$$

or
$$-[v_i n_i]^+_{-} = 0$$
,  $[\sigma_{ij} n_j]^+_{-} = 0$ , and  $-\sigma_{ij} n_i n_j \ge p_f$ , (5b)

where  $[f]_{-}^{+} = f^{+} + f^{-}$  is the sum of the limiting values of the bracketed quantity,  $n_i^{\pm}$  being the outward-pointing normal to the side labeled '+' or '-'. Lack of a superscript indicates that the equation holds regardless of which side of the contact the limit is taken from. In addition,

$$(\delta_{ij} - n_i n_j) \sigma_{jk} n_k = 0, \tag{5c}$$

where  $\delta_{ij}$  is the usual Kronecker delta. The conditions (5a) state that normal stress in the contact areas is still given by equation (4) when the surfaces are about to move apart, since the sum of the outward-pointing normal components of velocity  $v_i^+n_i^+ + v_i^-n_i^-$  measures how fast the two sides of the contact area move towards each other. By contrast, if the surfaces are not moving apart as in condition (5b), normal stress is continuous across the interface, and compressive normal stress must equal or exceed the fluid pressure. The third condition (5c) imposes vanishing shear stress. as was done previously in Zarrinderakht et al (2022): in other words, the model ignores the possibility of ice-on-ice friction.

**Reviewer comment:** Equation (7), extra comma

**Response** There are two separate equalities on the same line, separated by a comma, and a trailing comma to link with the text that follows. That use of punctuation seems correct to us.

**Reviewer comment:** Citation of Figure 1 is missing. It should be somewhere in section 2.1. Furthermore, the first figure citation in the main text is Figure 5a, which is also unusual.

**Response:** We have added a reference to figure 1 at the end of the first sentence of section 2.1 ("(see figure 1)"). The reference to figure 5a was a legacy error, resulting from switching the orders of figures 2–4 and 5–7 during the writing process for the original submission, That reference (first para of section 2.4) should now be to figure 2a.

**Reviewer comment:** Equation (10), consider indicating hw and s in Figure 1 sketch.

**Response:** The updated figure now includes these.

**Reviewer comment:** L156: a d  $\mapsto$  and

Response: Corrected.

**Reviewer comment:** L192: where... the sentence is not finished (?)

textbfResponse We have removed "and  $\delta_{ij}$ ; the Kronecker delta is defined at an earlier point in the paper.

**Reviewer comment:** L194:  $t_i \mapsto t_c$

**Response:** Corrected.

**Reviewer comment:** L205:  $\partial \Omega_b^-$  should be  $\partial \Omega_s^-$ ?

**Response** Indeed, this expression is nonsense; it should have said  $\partial \Omega_{\rm b}^- \cup \partial \Omega_{\rm s}^-$ . Corrected.

**Reviewer comment:** Equation (21): delete the negative sign before 0.

Response: Corrected.

Reviewer comment: Line 237: Again, try to cite figures in order, e.g. Figure 2a?

**Response:** Corrected. See response to "Citation of figure 1 is missing" above.

**Reviewer comment:** L250: The authors are using stress and displacement matching method to estimate the static stress intensity factor. The stress matching method requires high degree of mesh resolution to obtain accurate value. Did the authors implement convergence studies on this problem? What would be the relative efficiency compare to the J integral approach?

**Response:** We tested extensively for convergence using a variety of different known solutions (including the solutions in Tada's handbook employed by Lai et al (2020), full citation in the manuscript) when first developing the boundary element code for Zarrinderakht et al (2022). The challenge with using a J-integral approach with a boundary element method while modeling multiple cracks in the same domain is that you cannot use just the outer domain boundary (excluding the matching crack faces) to compute the J-integral, as you would with a single crack: the domain boundary encompasses multiple cracks, so you do not get the stress intensity factor for just one of the crack tips from the calculation. You would have to introduce other contours inside the domain, compute displacement gradients, stress and strain energy density on those contours from the boundary element solution, and then compute the J-integral from these. You would also have to make sure that this interior contour does not intersect any other crack, which makes it a bit of an exercise in computational geometry that looks non-trivial to automate. The BEM solver is quite inexpensive to run even with high resolution so that seemed like a much simpler route to take.

**Reviewer comment:** L261: "We assume that such short cracks are readily available as material flaws in the ice shelf...". Does this sentence indicate cracks can potentially develop everywhere (with tensile effective pre-stress) with the rate defined by equation (24), although only at the predefined locations in this study?

**Response:** For the present version of the code, such flaws are assumed to occur only at the predefined locations, as the qualifier immediately following the cited passage is intended to indicate:  $\dots$  (although we consider them only at the predefined locations  $x_{s}(t)$  and  $x_{b}(t)$  as discussed above) Our next goal in this line of research is to incorporate arbitrary crack geometries (as well as buoyancy effects). That is eminently possible,1 but beyond the scope of the present paper, or the PhD thesis it is based on. We reference this in the discussion A closely related issue is our insistence that there can be only two cracks, one on each ice surface. That choice allows a relatively simple model set-up, with cracks in known locations propagating vertically. The plot of effective pre-stress  $\sigma_{xx}^{\text{eff}}$  in Figures 2b and 5b, however, suggests that additional seed cracks would grow (and would have grown prior to the domain shape shown having been attained) if inserted in a large range of locations along the basal surface.  $\sigma_{xx}^{\text{eff}}$  is the effective pre-stress stress acting on a vertical seed crack, which is the likely favored direction in which new cracks should grow on a horizontal surface. As the two plots show,  $\sigma_{xx}^{\text{eff}}$  is tensile along most of the lower boundary, and in particular, where that lower boundary is approximately horizontal, suggesting that seed cracks inserted there should grow. It is plausible that seed cracks at the upper surface would also grow: Figures 2b and 5b show effective stress as defined in terms of the basal water pressure, and are therefore not relevant to the formation of surface cracks.

This suggests a future improvement of the model should incorporate not only buoyancy effects on stresses at the boundary, accounting for the effect of elastic surface displacements on the fluid pressure there, but also the possibility of multiple interacting cracks that can have arbitrary orientation, in the expectation that a preferred crack spacing and orientation will emerge spontaneously, rather than being imposed by the choice of initial domain width, and through the assumption of vertical, laterally offset cracks.

**Reviewer comment:** L299: "sea spring" scheme is not a well known scheme in glaciology (at lease to me). Furthermore, the citation Durand et al., 2009 does not has section 3.4 and is not about handling the normal stress condition. Therefore, this part and the rest of that paragraph is quite unreadable to me.

**Response:** Our apologies. The Durand et al reference given in the reference list appears to be the

<sup>1see the Gordeliy et al reference in the updated paper, although the method for determining crack orientation described there needs to be applied to a BEM rather than XFEM discretization

result of a bibtex mix-up, and points to the wrong paper entirely. This has been corrected, and now references the appropriate paper:

Durand, G., Gagliardini, O., de Fleurian, B., Zwinger, T., and LeMeur, E.: Marine ice sheet dynamics: Hysteresis and neutral equilibrium, J. Geophys. Res., 114, doi:10.1029/2008JF001 170, 2009.

This paper does have a section 3.4, which describes the regularization method (although the phrase "sea spring" seems to have emerged later; it was in use by the time the Bassis and Berg paper cited in the manuscript was written)

**Reviewer comment:** section 2.5: How sensitive is the model to temporal  $(\delta t/10)$  and spatial mesh resolution?

**Response:** There are two pieces here: the  $\delta t/10$  part is the time step used to update the viscous pre-stess after a crack propagation epsiode. The results are not sensitive to the scale factor 1/10; this is simply what we used.

The temporal step size  $\delta t$  is determined by a CFL condition as described in section 2.5, and changing the finite element mesh automatically changes the time step size. We tested sensitivity of our results to finite element size (as well as boundary element size as previously reported in section 3.3 and found no noticeable effect of double or halving mesh resolution. We state this in the new final paragraph of section 3.3 in the updated manuscript as

Figure 8 focuses on the effect of boundary element size, because of the coarser resolution used in the boundary element method near the crevasse tips compared with the finite element mesh. We also tested for the effect of finite element mesh resolution, by doubling and halving linear element sizes. Doing so was found to have no noticeable effect on results.

**Reviewer comment:** L335: Describe the physical meaning of  $R_{xx}$  and  $\tilde{R}_{xx}$  rather than cite the variable from other references.

**Respose:** We have moved all of this material forward to section 3.1, and substantially re-written it. The probably most relevant part of this the 14th paragraph If B and n are the usual parameter's in Glen's law (Cuffey and Paterson, 2010), then we can define a proxy  $\tilde{R}_{xx}$  for  $V_X$  through

$$\tilde{R}_{xx}(t) = 2BV_X(t)^{1/n}.$$
 (12)

The quantity  $\tilde{R}_{xx}$  has units of stress and equals the non-cryostatic extensional stress in the ice if the domain remains an unfractured rectangle (in which case  $\sigma_{xx} = \rho_i g(\bar{s} - z) + \tilde{R}_{xx}$ ). In that case,  $R_{xx}$  corresponds to the viscous extensional stress parameter  $R_{xx}$  in previous models for elastic fracture propagation (e.g., van der Veen 1998a,b; Lai et al, 2020, Zarrinderakht et al, 2022).

That should unambiguously define both  $\tilde{R}_{xx}$ ; the reference at the end of the paragraph is simply meant to help the reader understand how our  $\tilde{R}_{xx}$  relates to the fairly commonly used  $R_{xx}$  in the relevant prior literature.

**Reviewer comment:** L363-: Again, these variables (same with  $\kappa$  mentioned a few times) are cited from other papers (especially unpublished) without explanation. Very hard (if possible) for the readers.

**Response:** Unpublished material — that is an unfortunate part of The Cryosphere's submission set-up. You can submit "companion papers" but these may not be automatically linked, and obviously the cross-citations in the uploaded pdfs do not automatically update to the doi assigned to the (publically accessible) preprints. Googling the title of the submitted Zarrinderakht et al paper would probably have fixed the issue; in either case, the citation has been updated to give the doi for the preprint of the companion paper, now listed as "Zarrinderakht et al (2023)".

In terms of presentation here, again, this material has moved forward to section 2.1. The reference to parameters in other papers does *not* to define  $\eta^*$  and  $tau^*$ , which have already been defined

fully. Rather, the reference to the notation in other papers is here to help the reader who may also be reading these other papers / manuscripts understand the relationship between parameters used there, and in the present manuscript. That seems highly advisable.

The relevant updated paragraphs in section 2.1 are paragraphs 10 and 15,

... We assume instead that, as the ice stretches and thins, the surface water level  $h_w$  remains equal to a constant fraction  $\eta^*$  (that is, constant in time, but otherwise unconstrained as a forcing parameter) of the mean ice thickness  $\bar{H}(t)$  over the domain at time t,

$$h_{\rm w}(t) = \eta^* \bar{H}(t). \tag{13}$$

 $\eta^*$  is then the direct equivalent of the dimensionless surface water level parameter  $\eta$  used in Zarrinderakht et al (2022,2023).

and

For such an unfractured rectangular domain, the standard theory of unbuttressed ice shelves (e.g., MacAyeal and Barcilon, 1988) predicts that  $\tilde{R}_{xx}(t) = (1 - \rho_i/\rho_w))\rho_i g\bar{H}(t)/2$ . To account somewhat crudely for buttressing effects, we define  $V_X(t)$  by putting

$$\tilde{R}_{xx}(t) = \tau^* \rho_{ig} \bar{H}(t), \qquad (14)$$

with  $\tau^*$  held constant in time at a value that represents the degree of buttressing;  $\tau^*$  is then the direct equivalent of the dimensionless extensional stress parameter  $\tau$  used in Zarrinderakht et al (2022,2023).

The emhpasis here is on "equivalent" as opposed to "definition". You do not need to have read either Zarrinderakht et al paper to understand the definition of  $\eta^*$  or  $\tau^*$ ; the sentences involving "equivalent" could be omitted from the manuscript without impacting the definition of either parameter.

Reviewer comment: L386: correct the unit of temperature

**Response** It now says -10°C

**Reviewer comment:** L406: Are  $d_b^{tot}$  and  $d_t^{tot}$  crack lengths at the bottom and top, correspondingly?

**Response:** We have moved the detailed definitoin of cumulative crack length to a new appendix D in response to a comment by the other reviewer. To calrify, the notation, the running text in section 3.1, paragraph now says ... we define cumulative basal and surface crack length variables  $d_{\rm b}^{\rm tot}$  and  $d_{\rm s}^{\rm tot}$  over multiple fracture propagation episodes ...

Reviewer comment: L416: variables are repetitive

**Response:** We do not see any actual repetition: one set has a superscript -, the other does not. Note that this material has now moved to appendix D.

**Reviewer comment:** L417:  $h_b$ ,  $ht \mapsto Hb$ , Ht?

Response: Yes. Corrected (now in appendix D)

Reviewer comment: Figure 2: no units for t?

**Response:** t in all figures is dimensionless, see section 2.6 / second sentence of section 3.1

Reviewer comment: L436: 'can begin', delete 'can'

**Response:** Corrected.

**Reviewer comment:** L447-L451: Figure 2b1 and Figure 2b2 should be Figure 4b1 and Figure 4b2

**Response:** Indeed. Thank you for spotting that.

**Reviewer comment:** L461: involve  $\mapsto$  involving

**Response** The text says

Subsequent episodes involve a single fracture propagation event each...

This seems correct to us.

**Reviewer comment:** Figure 4: are there some plotting issues such that the axes are smaller than the domain?

**Response:** There were significant issues with getting a zommed-in plot in paraview and subsequently overlaying axes. This led to the solution you see now.

**Reviewer comment:** Figure 7: same problem as Figure 4, the axis is offset, and there are two blue lines in the panels.

**Response:** Same explanation for the axes being where they are. The two blue lines were indeed an issue. We have updated the figure caption to say

... The horizontal light blue line at z = 0.025 indicates the surface water level, the dark blue line at z = 0 is sea level. ...

**Reviewer comment:** section 3.3: Could you present a figure with the mesh on top, so we can see the finite element mesh in the calculation domain as well as the boundary element?

Response: Yes.

**Reviewer comment:** Section 3.4: In L346 and L362, and are described as 'a constant', while these are actually the two essential forcing parameters tested in section 3.4. For these important parameters, the physical meaning should be clear, and the chosen of the ranges should be justified. **Reviewer comment:** L478-479: What are the different element sizes tested here? I think a proper mesh convergence study should be conducted and presented.

**Response:** The element sizes being used are specified in the caption to figure 5. We now point this out explicitly in the revised text

To test for possible resolution-dependent effects that may result, we have recomputed the solution shown in Figure 5 with different boundary element sizes as shown in Figure 8 (see caption for details).

As for a "proper" convergence study, we'd need to know more about exactly what the referee has in mind. From our knowledge of how this works in numerical analysis, you would need a known, exact solution to compute residuals, and typically would refine the mesh over multiple orders of magnitude in element size to see rates of convergence. For the former (an exact solution), that is conspicuously absent for the time-dependent, coupled problem here. For the latter (how much to change resolution by), the time-dependent nature of the problem and the curse of dimensionality rapidly kill our ability to do that with the unfunded computational resources we have available (which is the reason why we tested for robustness to halving or doubling element size instead). We *would* note that the finite element package Elmer has been tested extensively as a widely used open source code. The boundary element code was tested in detail during the preparation of Zarrinderakht et al (2022), where the viscous pre-stress is prescribed analytically. To test the coupling beween Elmer and the boundary element code, we were able to use those analytical pre-stresses, which are exact solutions of the Stokes flow model for a rectangular slab and therefore easily replicated by Elmer; the key here was to test for any indexing issues in coupling the codes.

**Response:** The updated text in paragraphs 10 and 15 of section 2.1 (where we have moved the relevant description of forcing) now states explicitly that  $\eta^*$  and  $\tau^*$  are constant *in time* (but can be changed between different runs of the model):

... We assume instead that, as the ice stretches and thins, the surface water level  $h_w$  remains equal to a constant fraction  $\eta^*$  (that is, constant in time, but otherwise unconstrained as a forcing parameter) of the mean ice thickness  $\bar{H}(t)$  over the domain at time t... and

To account somewhat crudely for buttressing effects, we define  $V_X(t)$  by putting

$$\ddot{R}_{xx}(t) = \tau^* \rho_{ig} \bar{H}(t), \tag{14}$$

with  $\tau^*$  held constant in time at a value that represents the degree of buttressing...